# 2015 and 2016 winter-time air pollution in China: SO₂ emission changes derived from a WRF-Chem/EnKF coupled data assimilation system

Dan Chen[1*], Zhiquan Liu[2*], Junmei Ban[2], Min Chen[1].

[1]Institute of Urban Meteorology, China Meteorological Administration, Beijing, 100089, China
[2]National Center for Atmospheric Research, Boulder, CO, 80301, USA

**Under Review For Atmos. Chem. Phys.**

---

* *Corresponding author: Dr. Zhiquan Liu (liuz@ucar.edu) and Dr. Dan Chen (dchen@ium.cn)*

**Abstract**

Ambient pollutants and emissions in China have changed significantly in recent years due to strict control strategies implemented by the government. It is of great interest to evaluate the emissions reduction and the air quality response using a data assimilation approach. In this study, we updated the WRF-Chem/EnKF (Weather Research and Forecasting (WRF) model coupled with the Chemistry/Ensemble Kalman Filter) system to directly analyze $SO_2$ emissions instead of using emission scaling factors, as in our previous study. Our purpose is to investigate whether the WRF-Chem/EnKF system is capable of detecting the emission deficiencies in the "bottom-up" emission inventory (2010-MEIC, Multi-resolution Emission Inventory for China), dynamically updating the spatial-temporal emission changes (2010 to 2015/2016) and, most importantly, locating the newly added sources. The 2010 January MEIC emission inventory was used as the priori (to generate background emission fields). The 2015 and 2016 January emissions were obtained by assimilating the hourly surface $SO_2$ concentration observations for January 2015 and 2016. The $SO_2$ emission changes for northern, western and southern China from 2010 to 2015 and from 2015 to 2016 (for the month of January) from the EnSRF (Ensemble Square Root Filter) approach were investigated, and the emission control strategies during the corresponding period were discussed. The January 2010-2015 differences showed inhomogeneous change patterns in different regions, including 1) significant emissions reductions in southern China; 2) significant emissions reductions in larger cities with wide increase in the surrounding suburban and rural regions in northern China, which may indicate missing raw coal combustion for winter heating that was not taken into account in the priori emission inventory; and 3) significantly large emissions increases in western China due to the energy expansion strategy. The January 2015-2016 differences showed wide emissions reductions from 2015 to 2016, indicating stricter control strategies having been fully executed nationwide. These derived emissions changes coincided with the period of the energy development national strategy in northwestern China and the regulations for the reduction of $SO_2$ emissions, indicating that the

updated DA system was possibly capable of detecting emissions deficiencies, dynamically updating the spatial-temporal emission changes (2010 to 2015/2016), and locating newly added sources.

Forecast experiments using the priori and updated emissions were conducted. Comparisons showed improvements from using updated emissions. The improvements in southern China were much larger than those in northern and western China. For the Sichuan Basin, Central China, Yangtze River Delta, and Pearl River Delta, the BIAS decreased by 61.8%-78.2% (for different regions), the RMSE decreased by 27.9%-52.2%, and the correlation coefficients increased by 12.5%-47.1%. The limitation of the study is that the analyzed emissions are still model-dependent, as the ensembles are conducted through the WRF-Chem model, and thus, the performances of the ensembles are model-dependent. Our study indicated that the WRF-Chem/EnSRF system is not only capable of improving the emissions and forecasts in the model but can also evaluate realistic emissions changes. Thus, it is possible to apply the system for emissions changes evaluation in the future.

## 1. Introduction

China is one of the fastest growing countries in the world and has produced a significant amount of air pollutant emissions. To control pollution, a series of strict control strategies has been implemented by the government since 2010, including both long-term pollution control strategies and temporary emergency measures activated under different air pollution alerts, which has led to large spatial-temporal changes of emissions (factory mitigation from urban to rural regions, industries staggering peak production, etc.). These spatial-temporal emission changes are difficult to reflect in a timely manner in both bottom-up emissions inventories and air quality models, thus creating large uncertainties. A lot of regional air quality modeling work has been conducted to evaluate the emissions reduction and air quality response by comparing the simulations of a baseline scenario and an emissions reduction scenario. However, there are large uncertainties in those simulations due to the deficiencies of the models, including the meteorological/chemical

initial/boundary conditions, the chemistry process parameterization and, most importantly, the uncertainties of emission inputs from the "bottom-up" emission inventories. On one hand, these three aspects lead to error accumulation and large biases compared to those from observations in the baseline scenario simulations. On the other hand, for emissions reduction scenarios during special control events, the common approach is to assume that the control policies are executed to different extents, and several emissions reduction ratios are calculated and simulated. The ratio producing results best matching the observations is assumed to be the real emissions reduction ratio. This methodology is useful for forward simulations to project the effects of emissions reduction but is not straightforward for evaluating whether the control policies are strictly implemented and whether reductions are actually achieved. The forward approach using these models can neither accurately evaluate the spatial-temporal emissions changes nor locate newly added sources that are missing from the "bottom-up" emission inventory.

Various Data Assimilation (DA) and inversion approaches (e.g., Evensen, 1994; Houtekamer *et al.*, 2005, Hunt *et al.*, 2007; Lee *et al.*, 2011; Pagowski and Grell, 2012, Miyakazi, *et al.*, 2012, 2013, Dai *et al.*, 2014; McLinden *et al.*, 2016) have been conducted to improve the forecast skills and optimize source emissions. The variational data assimilation approach can greatly improve the initial condition by integrating the observational data into the model forecast, but the benefits quickly disappear due to the inaccurate emissions in the model. The inversion approach (also called the "top-down" approach) has been of great interest, as the observations can be directly used to constrain and optimize emissions. There are various methods by which to implement the "top-down" emission constraints, including the adjoint approach (e.g., Guerretta *et al.*, 2017), inverse approach combining satellite/surface observational data with regional/global models, and/or Ensemble Kalman Filter (EnKF). Because the adjoint method involves a huge amount of 4Dvar code development, its application is rather limited. The inverse approach using regional/global models and/or the ensemble Kalman filter method are much more flexible; thus, they are commonly used (Miyazaki *et al.*, 2012; Tang *et al.*, 2011,

2013, 2016). For the inverse approach using satellite data, due to satellite data availability, monthly data are usually used in studies, which can only provide information on the historical trends of the total emissions amount at the regional and national levels. Compared to satellite data, the use of intensive hourly surface observations as constraints can provide more spatial-temporal characteristics of emissions; thus, they can be used to evaluate the spatial-temporal emissions changes and to locate newly added sources that are missing from the "bottom-up" emissions inventory.

In our previous study, Peng *et al.* (2017, 2018) extended the ensemble square root filter algorithm to simultaneously optimize the chemical initial conditions and the emissions input, aiming to improve the forecasting of atmospheric $PM_{2.5}$, $SO_2$, $NO_2$, $O_3$, CO and $PM_{10}$ by using the WRF-Chem/EnSRF (Weather Research and Forecasting (WRF) model coupled with the Chemistry/Ensemble Square Root Filter) system. The surface observational data are used as constraints to update initial condition and relevant emissions (through emissions scaling factors) by minimizing the error variances. The WRF-Chem is used to propagate the initial ensemble forward in time, and the EnSRF is used to assimilate the observations and update the initial chemical conditions and emissions. In air quality models, deficiencies of concentration simulations come from various aspects, including the initial condition, emissions, meteorology, chemistry, transport, etc. Especially for $PM_{2.5}$ and $PM_{10}$ simulations in China, the significant differences between the models and observations possibly come from the deficiency of the chemistry representation in the model, including missing paths of secondary organic aerosols (e.g., Chen *et al.,* 2016) and heterogeneous reactions (e.g., Zheng *et al.,* 2015), in addition to emissions. However, to reduce the error variance, emissions adjustments may compensate for the model error, which leads to unrealistic/excessive emissions adjustment. Because the purpose was to improve the forecasting of chemical species, evaluations of emission changes were not conducted in the previous two studies.

In this study, we introduce two different DA techniques to investigate $SO_2$ emissions changes. First, we

updated the EnSRF system to evaluate $SO_2$ emissions changes, for which the chemistry is better understood and represented in the model. The 2010 January MEIC (Multi-resolution Emission Inventory for China) emissions inventory (Zhang *et al.,* 2009; Lei *et al.,* 2011; He 2012; Li *et al.,* 2014) was used as the priori (to generate emissions background fields), and the 2015 and 2016 January emissions were generated by assimilating hourly surface $SO_2$ concentration observations for January 2015 and 2016. Our purpose is to investigate whether the EnKF algorithm can be capable of detecting emissions deficiencies in the "bottom-up" emissions inventory (2010-MEIC), dynamically updating the spatial-temporal emissions changes (2010 to 2015/2016) and, most importantly, locating newly added sources. Our goal is not only to improve the emissions and forecasting in the model but also to understand to what extent the DA system can accurately evaluate realistic emission changes, thus allowing it to be applied for emissions change evaluations in the future. To better detect new emissions sources, we updated the system to directly analyze $SO_2$ emissions instead of emissions scaling factors, as in Peng *et al.* (2017, 2018). In addition to the EnSRF DA algorithm, we also applied the Gridpoint Statistical Interpolation (GSI) variational DA (3D-var) system to generate the $SO_2$ reanalysis fields, which is helpful in diagnosing the priori emissions deficiency and year-to-year emissions changes in the model. Finally, to fully utilize the DA system, we investigated the combined effects of improved initial conditions (by 3D-var) and dynamically updated emissions (by EnSRF) in the forecast experiments. It has always been challenging to verify optimized "top-down" emissions from the inverse approach due to the uncertainty of the "bottom-up" emissions inventory and the lack of sufficient independent observational data (not used in the DA process). Herein, we designed three groups of comparisons to address this issue, and the details will be discussed in section 2.6.

The paper is organized as follows. In section 2, the DA system, priori emissions, observational data and experimental design are described. The reanalysis $SO_2$ fields obtained using the GSI 3D-var DA system are analyzed in section 3, focusing on the possible indications of priori emissions deficiency and year-to-year

(2015-2016) changes. Section 4 describes the results from the emissions assimilation experiment using the updated WRF-Chem/EnKF system. This section starts with the evaluation of the ensemble performance to verify the DA system capability. Then, the derived emissions changes (2010 to 2015, 2015 to 2016) obtained by the EnSRF approach are given spatially throughout the whole domain and in 8 different regions with inhomogeneous spatial patterns. The temporal factors derived from the assimilation experiment are also given in section 4. To evaluate the accuracy of the analyzed 2015 and 2016 emissions, two sets of forecast experiments with the priori emissions and the analyzed emissions, respectively, were conducted and discussed. The details are given in section 5, and the conclusions follow in section 6.

## 2. Model description, observations and methodology

We applied two different DA techniques. In the first approach, we extended the Gridpoint Statistical Interpolation (GSI) 3D-var DA system originally developed by Liu *et al.* (2011) and recently updated by Chen *et al.* (2018) to assimilate $SO_2$ observations, aiming to generate $SO_2$ reanalysis fields. In the later approach, we updated the EnKF DA system (that used in Peng *et al.,* 2017) to optimize $SO_2$ emissions changes using surface observations as constraints. The WRF-Chem configurations are the same as those in Chen *et al.* (2018), and the update of the GSI 3D-var DA system is also built upon Chen *et al.* (2018); thus, only simple descriptions are given in this section. A further description of the WRF-Chem/EnKF DA system is given, and the priori emissions, observations and experimental design are introduced in detail.

### 2.1 WRF-Chem configuration

WRF-Chem model version 3.6.1 was used in this study (Grell *et al.,* 2005; Fast *et al.,* 2006). The parameterizations were identical to those of Chen *et al.,* (2016), and they are listed in Table 1. The model horizontal resolution is 40 km, and the domain covers most of China and the surrounding regions (Fig. 2). There are 57 vertical levels extending from the surface to 10 hPa. It should be noted that uncertainties might

be produced in the analysis due to neglecting the rapid urbanization/land use changes from 2010 to 2015/2016.

**2.2 Priori emissions**

The Multi-resolution Emission Inventory for China (MEIC) (Zhang et al., 2009; Lei et al., 2011; He 2012; Li et al., 2014) for January 2010 is used as the priori emission input. Unlike in other countries, the national emissions inventories in China (e.g., NEI05-08-11-14-17) are provided in a timely manner and updated for the public. It was previously stated that "there are no official data about how much air pollutants are emitted by China every year. The inventories developed by researchers often lag several years behind the present" (Zheng *et al.,* 2018). MEIC is the only publicly available emissions inventory dataset released by the Tsinghua University research community. In the MEIC, the total amount of sectoral emissions at the national and provincial levels has generally been estimated based on the "bottom-up" approach, which relied on available statistical information concerning activities (energy, industrial production, vehicles, etc.), emissions factors and end-pipe control levels. Due to the large burden of work and the availability of statistical data, the MEIC emissions inventory is not updated annually (e.g., the public versions are MEIC-2010-2012); thus, there are always a few years of time-lag when applying MEIC EI for research studies. In addition, to drive the regional air quality models, the annual/monthly total amounts of emissions at the national and provincial level are allocated spatially and temporally to generate hourly gridded emissions input for the model. Concerning temporal allocation, as many emissions sources have large diurnal and weekly variability that is not fully represented, arbitrary hourly/weekly factors were used in preparing the hourly gridded emissions for the air quality models. Thus, the uncertainties of the statistical information and the spatial-temporal allocation could both cause inaccurate representation of the hourly gridded emissions input and affect the performance of the model application.

The preprocess used to convert the original emissions inventory (in $0.25 \times 0.25$ degree) to match the model grid spacing (40 km) is the same as that used in Chen *et al.* (2018). The spatial distribution of priori $SO_2$ emissions in the simulation domain is shown in Fig. 2. A number of studies have revealed the uncertainties

of the "bottom-up" emissions inventories, including the energy statistics at the national and provincial levels (e.g., Hong *et al.*, 2017) and emissions factors from different industry sectors (e.g., Zhao *et al.*, 2011, 2017). Our purpose is to investigate not only the uncertainty of the priori MEIC emissions but also the capability of the DA system to dynamically update the $SO_2$ emissions using surface observations as constraints. For this purpose, the changes of $SO_2$ emissions from the priori emission year (2010) to our focus years (2015 and 2016) are emphasized.

There are several different driving factors in different regions that may lead to inhomogeneous changing trends during the examined years (especially from 2010 to 2015). As the Chinese government has implemented desulfurization legislation (since 2005-2006 but with stricter control of the actual use of installations since 2008-2009) and strict control strategies to ensure the air quality during winter seasons since 2013, significant $SO_2$ emissions reductions are expected to have occurred since 2010. However, there are converse results for certain regions. For example, Cheng *et al.* (2017) and Zhi *et al.* (2017) conducted a village energy survey for the rural areas in Hebei Province and revealed a huge amount of missing rural raw coal for winter heating. For Beijing and Baoding, rural emissions from raw coal in winter were higher than those from the industrial and urban household sectors in the two cities in 2013. Considering the living habit of residents in northern China, this may imply an extreme underestimation of rural household coal consumptions by the China Energy Statistical Yearbooks. Additionally, multisatellite data (Ozone Monitoring Instrument-OMI, the SCanning Imaging Absorption spectroMeter for Atmospheric CartograpHY-SCIAMACHY) revealed increasing $SO_2$ emissions due to energy industry expansion in northwestern China (the central government of the People's Republic of China, 2012, 2014), especially new power plant installations in Xinjiang and Shaanxi, e.g., in Shen *et al.* (2016) and Koukouli *et al.* (2016). Eight different regions are illustrated to address this issue. Northern China is divided into two regions, the North China Plain (NCP) and Northeastern China (NEC), as the North China Plain is more emissions-intensive and may experience more strict control strategies than those

in Northeastern China during winter haze periods. Northwestern China is also divided into two regions, including the EGT (the Energy Golden Triangle) and XJ (Xinjiang). Southern China is divided into four regions according to its geographic characteristics, including the SB (Sichuan Basin), CC (Central China), YRD (Yangtze River Delta), and PRD (Pearl River Delta). The spatial distributions of the $SO_2$ emissions in the 8 regions are also illustrated in Fig. 2.

In terms of the temporal allocation, we applied predefined functions for the diurnal variations of the priori $SO_2$ emissions, but the hourly factors are the same for all the sectors and all the grids, which is not optimal. Actually, different sectors/regions may have different hourly emissions factors. For example, the transportation sector may emit peak emissions during rush hours, and the industry sector may emit emissions during the production period. Thus, it is not optimal to use the same hourly emissions factors for all sectors. For regions in different time zones, different hourly emissions factors are also expected. The diurnal variability is not publicly released and is highly uncertain, which brings large uncertainty in the simulations. We also want to investigate whether the DA system can capture the diurnal variations of $SO_2$ emissions by using the hourly surface $SO_2$ concentrations as constraints.

**2.3 GSI 3D-var DA system**

Building upon the GSI 3D-var DA system used in Chen *et al.* (2018), we extended the system capability of assimilating surface $SO_2$ observations. The algorithm and methodology for the aerosol DA are described in Chen *et al.* (2018). Here, only the differences implemented for the $SO_2$ DA are addressed.

The $SO_2$ observation operator is rather simple, written as $\prod m = \rho_c M_{SO_2}$. The unit of the model-simulated $M_{SO_2}$ is ppm; thus, multiplication by the unit conversion $\rho_c$ was required to convert the units to µg m$^{-3}$ for consistency with the observations. The observation errors are calculated similarly to the process in Chen *et al.* (2018). In the data quality control process, $SO_2$ observational values larger than 650 µg m$^{-3}$ or observations leading to innovations/deviations (observations minus the model-simulated observations determined from the first-guess fields) exceeding 100 µg m$^{-3}$ were not used. The static BECs (background

error covariance) were computed via the "National Meteorological Center (NMC)" method (Parrish and Derber, 1992) by taking the differences of the 24-hr and 12-hr WRF-Chem forecasts valid at the same time for 60 pairs valid at either 00UTC or 12UTC over January 2015. The standard deviations over the whole domain are shown in supplemental Fig. S1.

**2.4 WRF-Chem/EnKF DA system**

The WRF-Chem/EnKF assimilation system framework (Fig. 1) was used. Similar to other ensemble DA configurations in WRF-Chem, such as WRF-Chem/Dart (Mizzi *et al.,* 2016), WRF-Chem is used to propagate the initial ensemble forward in time. The EnSRF is used to assimilate the observations and update the meteorological conditions, chemical initial conditions and/or emissions. The differences relative to WRF-Chem/Dart are mainly in the assimilation engine (the ensemble adjustment Kalman filter is used in WRF-Chem/Dart, while the ensemble square root filter is used in our study), the structure of the state variables (meteorological and chemical initial/conditions are used in WRF-Chem/Dart, while chemical initial condition/emissions are used in our study), the cycling procedures, etc. The WRF-Chem/EnKF assimilation system framework (Fig. 1) is very similar to that of Peng *et al.* (2017). Peng *et al.* (2017) focused on the joint analysis of both the initial conditions and emissions of $PM_{2.5}$ and addressed the forecasting skill improvement from using the EnKF system. Here, we focus on the estimation of $SO_2$ emissions, aiming to investigate the system capability of reflecting the spatial-temporal emissions changes using observational data as constraints. In addition, instead of setting the emissions scaling factors as control variables and adjusting the emissions by timing the scaling factors, we attempted to set emissions directly as control variables, which allowed adjustments by adding absolute emission values. Through this method, the detection of new emissions sources would become more flexible. As in the previous approach, the emissions scaling factors were extremely large when a new emissions source (e.g., a new power plant) occurred in an originally "clean" model grid (priori emissions close to zero), with the scaling factor being so large that it might be treated as "unrealistic" and be

filtered out in the system. The direct analysis of emissions is expected to be more appropriate for this case.

The Ensemble Square Root Filter (EnSRF, Whitaker and Hamill 2002) algorithm is very similar to that in Peng *et al.* (2017), except for some differences, such as the state variables (changed from aerosols to $SO_2$ concentrations and emissions) and the inflation factor. In addition, the forecast model for emissions is also different from that in Peng *et al.* (2017). More details on the differences from Peng *et al.* (2017) are described below.

### 2.4.1 State variables

A similar ensemble square root filter is used in this study to update a 50-member ensemble to that used in Peng *et al.* (2017). We also applied the state augmentation method (e.g., Miyazaki *et al.*, 2012). The only difference is that model parameter ($SO_2$ emissions) is directly estimated by including it as part of the state vector together with the model forecast variable ($SO_2$ concentration). The background ensemble is defined as below:

$$x_i^b = \begin{bmatrix} C_i^b \\ E_i^b \end{bmatrix} \tag{1}$$

in which $x_i^b$ is the $i^{th}$ member's background vector, consisting of model-simulated $SO_2$ concentrations $C_i^b$ and the $SO_2$ emissions $E_i^b$. For a grid with zero emissions in the priori emissions, the absolute emissions values would be added into the DA analysis to reflect the new emissions sources. Negative emissions estimates were not permitted in the system due to mandatory setting of the minimum values (a small positive value close to zero).

In Miyazaki *et al.* (2012), the state augmentation method was used to estimate $NO_x$ emissions using satellite observations (Ozone Monitoring Instrument-OMI retrieved $NO_2$ column) as constraints with a local ensemble transform Kalman filter (LETKF). The employment of combined state vectors (both $NO_2$ concentrations and $NO_x$ emissions) allowed indirect relationships between $NO_2$ concentrations and $NO_x$

emissions, causing complex chemical and transport processes to be considered through the use of the background error covariance, which was produced by the ensemble Chemical Transport Model-CTM forecast. Building upon Miyazaki *et al.* (2012), we used a similar approach to address the indirect relationships between $SO_2$ concentrations and $SO_2$ emissions caused by chemical and transport processes. The chemical processes include several paths of $SO_2$ oxidation, such as gas-phase reactions with the hydroxyl radical (OH), aqueous-phase reactions with $O_3$ or hydrogen peroxide ($H_2O_2$), and heterogeneous reactions in high-RH environments (e.g., Li *et al.,* 2011; Wang *et al.,* 2012).

To reduce spurious correlations due to sampling errors, covariance localization was applied following Schwartz *et al.* (2014) and Peng *et al.* (2017). EnSRF analysis increments were forced to zero 1280 km from an observation in the horizontal direction and to 1 scale height (in log pressure coordinates) in the vertical direction using a Gaspari and Cohn (1999) polynomial piecewise function.

**2.4.2    Inflation factor in the EnSRF**

During the analysis process, the analyzed emissions of different members converge gradually, and the background emissions (calculated according to Eq. 4) of different members in the DA cycling become similar, thus leading to a small ensemble spread (variance). To maintain the spread level, an artificial inflation process (the original perturbations time the inflation factor larger than 1) was added to increase the perturbations. Further, multiplicative inflation was applied to posterior (after assimilation) perturbations concerning the ensemble mean analyses, following Whitaker and Hamill (2012)'s "relaxing-to-prior spread" approach with an inflation parameter $\alpha$.

$$\delta \mathrm{x}_a^i \leftarrow \delta \mathrm{x}_a^i \left( \alpha \frac{\sigma_b - \sigma_a}{\sigma_a} + 1 \right) \tag{2}$$

In this equation, $\delta \mathrm{x}_a^i$ is the $i^{th}$ member's analyzed perturbation concerning the mean analysis $(\mathrm{x}_a^i - \bar{x})$, $\alpha$ is the inflation factor, and $\sigma_b$ and $\sigma_a$ are the prior (before assimilation) and posterior standard deviations at each model grid point, respectively. Using the definition of the standard deviation, Eq. (2) can be expressed

as

$$\sigma_a \leftarrow \alpha\sigma_b + (1 - \alpha)\sigma_a \tag{3}$$

Assimilating observations reduces the ensemble spread; thus, without inflation, $\sigma_a < \sigma_b$. From Eq. (3), if $\alpha > 1$, the inflated posterior spread is forced to be larger than the prior spread ($\sigma_b$). Conversely, for $\alpha < 1$, the inflated posterior spread must be less than $\sigma_b$. As no prior or additive inflation was employed, $\alpha > 1$ was necessary to maintain the ensemble spread, and we used the inflation factor of $\alpha = 1.12$.

### 2.4.3  Forecast models for emissions

The forecast model is important, as it propagates observation information, inflates the analysis spread and determines the quality of the first guess. In Peng *et al.* (2017), a smoothing operator served as the forecast model for the emissions scaling factors. In this study, direct emissions instead of scaling factors were treated as part of the state variables, thus producing a similar method for the forecasting approach to that used in Miyazaki *et al.* (2012). A linearized forecast model (M) provides a first guess of the state vector for data assimilation based on the background error covariance from the previous analysis time $t_n$ to the new analysis time $t_{n+1}$,

$$P^b_{(t_{n+1})} = 0.75 \times MP^a_{(t_n)}M^T + 0.25 \times P^b_{(t_0)}, \tag{4}$$

in which a persistent forecast model (M=I) is used for $SO_2$ emissions, and the estimated emissions are used in the next step of ensemble forecasting. To prevent parameter covariance magnitude reduction, we added the initial prior ensemble as random noise. The forecast model for direct emissions is weighted 75% toward the results from the previous analysis time and 25% toward the static initial prior ensemble. The initial prior ensemble of $SO_2$ emissions for the first EnSRF analysis was constructed from the priori emissions by taking Gaussian random draws from a standard Gaussian distribution and varied for each ensemble member as in Peng *et al.* (2017). This approach incorporates the useful information from the previous time step and the priori emissions, which propagates the observation information from one step to the next while still keeping

some of the characteristics of the priori features.

### 2.4.4    The initialization and DA cycling procedure

The WRF-Chem/EnKF assimilation system framework is shown in Fig. 1, and the workflow is briefly introduced here. The initialization and spin-up procedures of the 50-member ensemble were conducted using 72-hr ensemble forecasts ahead of the focused period through the same method used in Peng *et al.* (2017, 2018). For the 50 members, the lateral boundary conditions and initial condition of meteorology from GFS were perturbed by adding Gaussian random noise with a zero mean and statistic background error covariances to the meteorological parameters. The emissions of the 50 members were generated by adding random noise to the priori emissions, similar to the method in Schwartz *et al.* (2014) and Peng *et al.* (2017). After the 72-hr forecasts, 50-memble ensemble $SO_2$ forecasts were generated, which were used as part of the background ($C_i^b$ in Eq. 1) in the first EMIS_DA cycle. The other part of the background ($E_i^b$ in Eq. 1) was the perturbed emissions of the last time step. In the EnSRF assimilation step, the state variables, including both the emissions for the last hour and the concentrations at the current hour, were updated. In the new 1-hr cycle, the background field of emissions is forecast through Eq. (4), and the background concentration is from the WRF-Chem 1-hr forecast using the updated chemical fields of the previous assimilation cycle as the ICs. With hourly cycling, the hourly analyzed emissions were obtained.

## 2.5 Observations

Hourly surface $SO_2$ concentrations for January 2015 and 2016 were obtained from the China National Environmental Monitoring Center (CNEMC). There are approximately 1600+ sites in our modeling domain (black dots in Fig. 2). As the 1600+ monitoring sites fall into 531 model grids, observations within the same grid are averaged (by latitude and longitude) for the purpose of statistics and verification. The observation sites span mostly northern, central and eastern China and are relatively sparse in western China. To ensure

1 data quality before use in the DA, $SO_2$ observational values larger than 650 μg m$^{-3}$ were deemed unrealistic

2 and not assimilated in the GSI 3D-var or the EnKF DA system.

**2.6 Experimental design**

To qualitatively evaluate the deficiencies of the priori $SO_2$ emissions, generate updated emissions for

2015 and 2016, and evaluate the improvements from the DA approach, five sets of experiments were

conducted. The corresponding comparisons and purposes are listed in Table 2. The simulated periods were

January of 2015 and 2016. The meteorological initial condition (IC) and boundary condition (BC) were

updated from GFS analysis data every 6 hr (for NO_DA) or 1 hr (for CONC_DA and EMIS_DA) to prevent

meteorology simulation drifting. The same WRF-Chem configurations (Table 1) were used in all the

experiments.

In the NO_DA experiment, a new WRF-Chem forecast was initialized every 6 hr starting at 00UTC, 20

December of the previous year to spin up the chemistry fields and was run through 23UTC, 31 January. The

chemistry fields were simply carried over from cycle to cycle. The 2010-MEIC priori emissions were used,

assuming the same emissions as in 2010. For CONC_DA, the hourly surface observations were assimilated

by GSI 3D-var, and the $SO_2$ concentrations were updated every hour starting from 00UTC, 1 January. The

background of the first time step is from the NO_DA simulation, and those of the later time steps are all from

the 1-hr WRF-Chem forecast using the updated chemical fields of the previous assimilation cycle as the ICs.

As the concentrations from CONC_DA is very close to the observations, the concentration differences

between CONC_DA and NO_DA possibly indicated a model deficiency in reproducing the reality, which was

mainly from the emissions changes from 2010 to 2015-2016. The assumption is that the GFS 6-hr analysis

data provide good meteorological IC/BC values and that the model accurately simulated the meteorology

conditions; thus, the emissions were the major deficiency in the model.

The EMIS_DA experiment with continuous hourly cycling of the WRF-Chem/EnKF was performed for January of 2015 and 2016. The initialization and spin-up procedures of the 50-member ensemble were conducted starting from 00UTC, 29 December of the previous year to 00UTC, 1 January of the next year. Then, the EMIS_DA cycle started at 00UTC, 1 January. Following the procedure in Fig. 1, the EMIS_DA experiment started with conducting EnSRF analysis and generated both the updated $SO_2$ concentration fields for the current time step and the updated analyzed $SO_2$ emissions for the previous time step. In this hourly cycling approach, 1-hour WRF-Chem/EnKF cycling was conducted for January of 2015 and 2016, and hourly analyzed $SO_2$ emissions were then obtained. We compared the 2015 emissions from EMIS_DA versus the 2010 priori emissions, as the emissions differences between the 2015 analyzed emissions and the 2010 priori emissions not only reflected the changes from 2010 to 2015 but also included the deficiencies in the 2010 priori emission. We also compared the updated emissions between 2015 and 2016, as the differences between the 2016 analyzed emissions and the 2015 analyzed emissions reflected the pure emissions changes from 2015 to 2016, since the deficiencies of the 2010 priori emissions were offset in the subtraction. The emissions control policies are discussed to investigate whether the emissions changes are reasonable.

To investigate the impact of using analyzed emissions from the EnKF DA system, two forecast experiments (NO_DA_forecast and EMIS_DA_forecast) were conducted for the same period. Twenty-four-hour forecasts were performed at 00UTC of each day from 1-31 January for 2015 and 2016. The original priori emissions and the updated analyzed emissions were used, respectively, in the NO_DA_forecast and EMIS_DA_forecast experiments. The chemistry initial conditions for each forecast in the two forecast experiments were from the 1-hour cycling GSI 3D-var (COND_DA) experiment. The meteorological IC and BC were all from GFS analysis and forecast data. The concentration differences between the two sets of 24-hour forecasts reflects the effects of the updated emissions.

## 3. Changes in ambient concentrations

This section presents the simulated $SO_2$ concentration results of NO_DA and CONC_DA. As shown in Chen *et al.* (2018), 1-hr cycling of the GSI 3D-var DA system produces reliable $PM_{2.5}$ reanalysis fields. As the methodology and procedure are the same for $SO_2$, we can expect that the improvement of $SO_2$ assimilation will be as good as that of $PM_{2.5}$, as evidenced by the basic statistics including the mean bias (MEAN/BIAS), standard deviation (STDV), and root-mean-square errors (RMSE) between NO_DA/CONC_DA and observations shown in Supplemental Fig. S2. Therefore, the purpose of this section is not to verify the performance of the GSI 3D-var assimilation experiment but to investigate the differences between NO_DA and CONC_DA. As NO_DA is the simulation with the 2010 emission inventory, while the results of CONC_DA can serve as gridded reanalysis data from real observations, the differences between the two runs actually reflect the possible deficiencies in the model. As the meteorology data are from the 6-hr GFS reanalysis data, we assume that most of the deficiencies come from using the 2010 priori emissions for the years 2015 and 2016 in the model, and the comparisons also provide an idea of the changing trends of the emissions.

### 3.1 Spatial distribution

Figure 3 shows the observed and modeled monthly average of surface $SO_2$ for January in 2015 and 2016. The observations show great differences between northern and southern China, reflecting the dominating role of heating-related emissions in northern China during the winter season. The high values in northern China also show localized characteristics (no smooth transitions from the high-value region to the surroundings) that reveal the localization of $SO_2$ emissions and transport. The NO_DA experiment significantly overestimates the surface $SO_2$ in the Sichuan Basin and Central China but underestimates it at several locations in northern China and Xinjiang. After GSI 3D-var hourly cycling of the DA, the CONC_DA experiment results are very close to observations because they corrected most of the biases in NO_DA except for the very high values at

some of the "hot spots" in northern China. The reason why the improvements at those locations are not significant may be the data filtering process, in which $SO_2$ data with either the observed values larger than 650 μg m$^{-3}$ or innovations/deviations (observations minus the model-simulated values determined from the first-guess fields) exceeding 100 μg m$^{-3}$ were rejected. The differences between CONC_DA and NO_DA more clearly revealed the inhomogeneous emissions changes in different regions. For 2015, a great $SO_2$ decrease from NO_DA to CONC_DA in most of the eastern and southern regions but increases in Northeastern China, the Energy Golden Triangle and Xinjiang are found, indicating that the 2010 January priori emissions should be adjusted accordingly (following decreasing/increasing trends respectively) to reflect the 2015 January status. The negative discrepancies in the eastern and southern regions are even larger for 2016, indicating continuous emissions decreases there.

To further investigate the deficiencies of the priori emissions, the spatial distributions of the statistics (MEAN BIAS, RMSE and CORR) at each observational site (with more than 2/3 valid data in the month) in January of 2015 and 2016 for the two experiments are shown in Fig. 4. We start from the 2015 statistics and then address the differences relative to 2016. In NO_DA, consistent with Fig. 3, the surface $SO_2$ in southern China (the Sichuan Basin, Central China, the Pearl River Delta and the Yangtze River Delta) is generally overestimated by 20-50 μg m$^{-3}$, but it is underestimated in Northeastern China and the Energy Golden Triangle. The BIAS also showed localized characteristics with positive biases in mega-cities (e.g., Beijing), along with negative biases in the surroundings, indicating overestimated/underestimated emissions, respectively. There are also high RMSEs in Northeastern China, Northern China Plain and the Energy Golden Triangle, indicating a wide spread of differences between observational data and NO_DA simulations, which may also indicate a model deficiency in reproducing strong temporal fluctuations (with the same daily emissions and fixed hourly factors as in the priori emissions). The poor correlations (less than 0.5) at most of the sites also indicate the assumption. From year to year, the biases in 2016 are even more prominent. With GSI 3D-var hourly cycling,

the BIAS, RMSE and CORR are greatly improved, as expected, while the improvements in northern China are smaller than those in southern China.

## 3.2 Changes from 2015 to 2016

The differences between the January values of the two years (2015-2016) are shown in Fig. 5. Observations (Fig. 5a) mostly show decreases from 2015 to 2016 for most sites, especially in the Northern China Plain and southern China. In the NO_DA experiment (with the same emissions and different meteorology), some decreases are shown that reflect the meteorology condition differences between the two years, but the observed significant decreases in the Northern China Plain and southern China are not captured. CONC_DA (Fig. 5b) did reproduce the large decreases in the Northern China Plain and southern China from 2015 to 2016. From the difference between Fig. 5b and 5c, it can be assumed that factors other than meteorology (e.g., emissions control measures) did play important roles in causing the decreasing changes. CONC_DA failed to reproduce the large positive changes at 3 locations in the Energy Golden Triangle region, as CONC_DA failed to reproduce the high $SO_2$ concentrations in both years due to the data filtering processes.

## 4. Changes in emissions

Before the emissions trend analysis, the ensemble performance was evaluated. For comparison with priori emissions, the analyzed hourly emissions were averaged monthly. Analysis of the total amounts and spatial changes were conducted for the aforementioned 8 regions. We focus on the emissions trends for two periods, 2010-2015 and 2015-2016. Additionally, the hourly factors (diurnal cycle) of the optimized emissions were given to reflect the values of the hourly DA.

## 4.1 Ensemble performance

In a well-calibrated system, when compared to the observations, the prior ensemble mean root mean square error (RMSE) would equal the prior "total spread", defined as the square root of the sum of the

observation error variance and the ensemble variance of the simulated observations (Houtekamer *et al.*, 2005). The time series of the hourly prior ensemble mean RMSE and the total spread of surface $SO_2$ in the 8 regions are shown in Figure 6. The time series of the two months (Jan. 2015 and Jan. 2016) are given separately. Typically, the statistics at a single site for a certain period reflect the model biases and variances of errors at that site for the whole period. Differently, herein, the statistics for the 8 regions were determined for all sites within the region at a 1-hr frequency, which means that the statistics actually reflect the biases and error variances of the model simulations for those sites at every hour. As the emissions and meteorology conditions could be very different at sites in the same region, the RMSE for that region could be large. Due to the spatial-temporal inhomogeneity of emissions and meteorological conditions in different regions, the model shows different performances in terms of the differences in the RMSE. The "total-spread" reflects the ensemble variances of the model-simulated values.

The magnitudes of the total spread and the RMSE are influenced by the diurnal cycle and the pollution events (driven by meteorology patterns and emissions). As expected, all of the total spreads in the 8 regions are smaller than the RMSE for almost the whole period, except those for the first few days in 2015. Because in the spin-up procedure, the lateral boundary and initial conditions of meteorology were also perturbed in addition to emissions perturbation, larger spreads were obtained in the first DA cycle, which may have remained for a short period. For all other periods without meteorology perturbations, insufficient spreads of $SO_2$ ensemble forecasts were shown, with the effects in the northern and western regions (North China Plain, Northeastern China, the Energy Golden Triangle, Xinjiang) being worse than those in southern regions (the Sichuan Basin, Central China, the Yangtze River Delta, and the Pearl River Delta), as the prior ensemble mean RMSE values in the northern regions were much larger, but the total spreads were fairly constant. For the northern regions, the total spread (Fig. 6) was relatively small compared to the RMSE. This might indicate that the analyzed emissions converged gradually and that the background emissions (calculated according to

Eq. 4) of different members in the DA cycling were similar, thus leading to the small spread. As the spread is small, some observations might be rejected in the DA outlier check, which may impact the DA performance. The distinction of the comparisons among different regions (the North China Plain vs. the Yangtze River Delta/Pearl River Delta) indicated the deficiencies of the perturbation procedure in the DA system when applied to northern regions. Further investigations should be conducted to generate larger spreads for northern regions in future studies.

**4.2 Analyzed 2015 and 2016 emissions**

The optimized $SO_2$ emissions obtained from the assimilations for Jan. of 2015 and 2016 are shown in Fig. 7. To address the changes from 2010 to 2015 and those from 2015 to 2016, the differences and ratios between the two groups (2010 vs. 2015 and 2015 vs. 2016) are given. For the comparison of 2015 analyzed emissions with 2010 priori emissions, as real observations were used to constrain the 2015 emissions, the differences between the two sets of emissions actually reflect the adjustments based on the 2010 priori emissions, which are needed to better capture observations; thus, the comparison not only reflects the changing trends from 2010 to 2015 but may also indicate the deficiencies of the 2010 priori emissions. It should be noted that the two aspects are mixed in interpreting the results. While the comparison of 2015 analyzed emissions with 2016 analyzed emissions is more straightforward, as they are both produced using observation constraints, the differences between the two reflect the annual changes between the two years, and the impacts from priori emissions deficiencies are removed in the subtraction.

Compared to the 2010 priori emissions, the analyzed emissions for 2015 show different spatial changes (northern, western, southern China). Large emissions decreases in southern China (the Sichuan Basin, Central China, the Yangtze River Delta and the Pearl River Delta) are shown, but there are also some small emissions increases in scattered regions. These increases are relatively small in absolute value (shown in a light-yellow color in Fig. 7c), but the 2015/2010 ratios can reach large numbers (shown in orange to red colors in Fig. 7d),

as the priori emissions in those regions are very small (Fig. 2); thus, minor changes lead to large ratios. For northern China (North China Plain, Northeastern China), the change pattern is somewhat opposite. Emissions increases are shown for most of the regions, with decreases only at scattered points. Large 2015/2010 increasing ratios are also shown in western China (Xinjiang and The Energy Golden Triangle), while the priori emissions are very sparse in those two region; thus, the emissions increases are more significant, which may indicate new emissions sources. For the changes from 2015 to 2016, the pattern is rather homogenous across the whole domain, with decreases in almost all regions.

**4.3 Changes in different regions**

To further illustrate the changes in different regions, the details of 8 regions are given in Fig. 8 (2015 vs. 2010) and Fig. 9 (2016 vs. 2015). Similar to Fig. 7, the emissions changes in terms of absolute values (left) and ratios (right) are given for each region. To better understand the geographic changes, the center locations of some large cities (capital cities of provinces and municipal centers at the city level) in those regions are labeled. According to Fig. 7, the change patterns are different in northern, western and southern China for 2015; thus, discussions are given based on this classification. We start from the comparison of the 2015 analyzed emissions with the 2010 priori emissions, as there is a five-year time lag between the two sets of emissions, along with large uncertainties, and thus, large changes are expected.

It is interesting to see that for northern China (North China Plain and Northeastern China), the most significant decreases occur in or around large cities (city center locations are labeled as black dots). The phenomenon is very prominent in the North China Plain, as we can see some "cold" spots (grids with cold colors) in Fig. 8a, which are either overlapped with city center locations (Beijing, Tianjin, Xingtai, Handan in Beijing-Tianjin-Hebei Region, and Dongying, Jinan, Zibo, Jining in Shandong Province) or adjacent to the center locations (Shijiazhuang, Linyi, Zaozhuang). As the center locations are represented as latitudes/longitudes that do not cover entire city areas, there might be some shifting produced from interpreting

the results when the city areas are too large (e.g., Shijiazhuang, Changchun, Shenyang) and have been split into two or more grids in the model. While the results still indicate that from 2010 to 2015, the emissions in these larger cities decreased due to the strict control strategies (factory migration from urban regions to remote regions, desulfurized equipment in factories/vehicles, low-sulfur energy, etc.), there are some emissions increases in the suburban and rural regions surrounding these larger cities, either due to emissions migration from urban regions or new emissions sources added due to urbanization development. The results might also indicate that the control strategies were executed at different levels in urban (more strict) and suburban-rural regions during 2010 to 2015. In Northeastern China, significant "cold spots" also occur in the three larger cities, including Ha'erbing, Changchun, and Shenyang, but increases mostly occur in other areas, indicating a similar trend to that in the North China Plain, in which large emissions decreases occurred in bigger cities and mild emissions increases occurred in suburban-to-rural regions from 2010 to 2015. In addition to the possible aforementioned reasons accounting for the different changes of urban and suburban-rural regions from 2010 to 2015, it should be noted that the month of January is during the heating season for the North China Plain and Northeastern China, and the large areas of emission increase might also indicate some heating emissions (from energy that has not been well statistically recorded, e.g., crop combustion and residential coal combustion) that are missing from the priori emissions.

In western China, where the emissions intensities are not so high and the emissions sources are relatively sparse, the emissions changing trends from 2010 to 2015 are more obvious and are more meaningful for distinguishing new emissions sources/regions. As some studies revealed increasing $SO_2$ emissions due to energy industry expansion and relocation in northwestern China from OMI measurements (Ling *et al.*, 2017), our 2015 analyzed emissions also show large emissions increases in the whole area of the Energy Golden Triangle and Xinjiang, except for in very few larger cities (Yinchuan, Wuhai, Lanzhou; Kelamayi). The emissions in other areas of the Energy Golden Triangle and Xinjiang are almost all increasing from 2010 to

2015. Especially for Xinjiang, the increases of emissions are all attributed to the rapidly developing cities, including U'rumqi, A'kesu, Ku'erle, Yecheng, Manasi, Tacheng, Huocheng, Bachu, A'tushi, Shanshan, Shache, etc. Koukouli *et al.* (2016) and Ling et al. (2017) used multisatellite data to investigate the SO$_2$ load changes from 2004-2014/2005-2015 and identified locations with increases (including U'rumqi in Xinjiang and cities in northwestern China). They reported that "These belong to provinces with emerging economies which are in haste to install power plants and are possibly viewed leniently by the authorities, in favor of growth." Our findings are also consistent with those of these two studies.

In southern China, decreasing changes are shown for large areas, especially in the Yangtze River Delta and Pearl River Delta, in which decreasing trends in larger cities are clearly shown, e.g., in Shanghai, Nanjing, and Hangzhou in the Yangtze River Delta and in Guangzhou, Shenzhen and Foshan in the Pearl River Delta, with relatively larger decreasing ratios in more well-developed cities. In the Sichuan Basin and Central China, the decreases in larger cities are also significant, and different extents are achieved in cities of different levels. For Chengdu, Chongqing, Zunyi, Guiyang, and Yunyang in the Sichuan Basin and Wuhan and Changsha in Central China, approximately 40-50% reductions are shown from 2010 to 2015. For other larger cities (municipal centers of cities), 20-30% reductions are shown.

As previously mentioned, the comparisons between the 2015 and 2016 analyzed emissions (Fig. 9) are more straightforward and reflect the necessary emissions changes from 2015 to 2016, since the uncertainties in the priori emissions are subtracted. As expected, decreasing trends are shown for almost all the labeled cities, indicating the continuing strict execution of control strategies. However, there are still some grids with emissions increases (approximately 10-30%) in surrounding regions, especially in the North China Plain, which might reflect the emissions increase from January 2015 to January 2016. As shown in Fig. 10 of Chen *et al.* (2018), the temperature in January 2016 was much colder than that in 2015, and the emissions increases at those points may indicate heating-related emissions. Compared with Fig. 8 (2015 vs. 2010), the changes

from 2015 to 2016 (both increases and decreases) in Fig. 9 are much milder.

The regional averages of the 2015 and 2016 January emissions are summarized in Table 3. In northern China (North China Plain, Northeastern China) and western China (the Energy Golden Triangle and Xinjiang), the 2015 analyzed emissions are all larger than the 2010 prior emissions. The increase percentages are 12.7%, 49.4%, 25.6% and 72% for the North China Plain, Northeastern China, the Energy Golden Triangle and Xinjiang, respectively, indicating an increasing trend from 2010 January to 2015 January, due either to emissions increases in reality (possibly in the Energy Golden Triangle and Xinjiang) or uncertainties in the 2010 priori emissions (possibly in the North China Plain and Northeastern China). The largest increase occurred in Xinjiang, reaching 72%, which is consistent with the previous findings of newly added emissions sources in that region. In southern China, the 2015 analyzed emissions are all smaller, and the decreasing ratios are -10.5%, -9.9%, -13.8% and -22.9% for the Sichuan Basin, Central China, the Yangtze River Delta and the Pearl River Delta, respectively. For the changes from 2015 to 2016, decreasing trends are shown for all regions, with the ratios ranging from -5.3% to -16.1%.

In the recent study by Zheng *et al.* (2018), the 2010-2017 trends of anthropogenic emissions in China were investigated. According to the "bottom-up" approach, the annual total amounts of $SO_2$ emissions were calculated to be 27.8, 16.9 and 13.4 Tg for the years 2010, 2015 and 2016, respectively. The 2010 to 2015-2016 decreases were mostly attributed to the power and industry sectors due to the strict pollution control measures implemented for these two sectors. The sectoral distribution of emissions changed significantly during the recent years, and emissions other than those from power and industry have occupied larger portions, especially for the residential sector, as the current control policies have limited effects on reducing emissions from the residential sector. According to Zheng et al. (2018), the national total $SO_2$ emissions decreased by 20.8% from 2015 to 2016. Our derived changes ratios for the month of January in most of the regions (NEC, XJ, SB, CC, PRD) are comparable (13.9%, 16.1%, 12.4%, 15.5%, 12.6%, respectively, see Table 3), but the

change ratios for NCP, ETR and YRD are relatively smaller. As discussed in Zheng *et al.* (2018), "bottom-up" emissions estimates are uncertain due to incomplete knowledge of the underlying data, and uncertainties are larger when emissions are contributed by scattered emissions sources. Especially for the residential sector, the effectiveness of the measures (e.g., phasing out of small high-emissions stoves, banning of coal heating) is difficult to validate due to the lack of inspections; thus, higher uncertainties may arise for regions in which residential emissions are relatively important.

**4.4 Hourly factors**

As hourly observations were used to constrain the emissions, analyzed emissions at an hourly frequency were obtained, which provided us an opportunity to investigate the hourly emissions factors from observations. To retrieve the hourly factors, the emissions during each hour (24 hr) are averaged based on the EMIS_DA experiment for the whole period (Jan. 2-31). The retrieved hourly factors for 2015 and 2016 and the 2010 priori emissions are shown in Fig. 10. The priori hourly factors are given arbitrarily, with two peaks during the day at 01UTC (09 Beijing-BJ time) and 09UTC (17 BJ time) to reflect the emissions during rush hours. The retrieved hourly factors in northern and western China showed two peaks at approximately 02TC (10 BJ time) and 12UTC (20 BJ time), but the second peak is obscure in southern regions. In addition, the second peak of the hourly factors in northern and western regions is are much lower than the first, which was different from the predefined curve. In Xinjiang, the peaks occurred later than in the other regions, indicating that the time zone differences caused a different energy consumption/emissions pattern. It should be noted that the hourly factors were derived from the analyzed emissions constrained from ambient concentration observations; thus, the response times from emissions to ambient concentrations were simplified in the assimilation system. Although the background emissions contain the information from the previous cycles, and thus may help to pass the response information, there might still be some time-lag in the retrieved hourly factors, which should be further verified.

**5.   Forecast improvements**

As there are large uncertainties in the "bottom-up" 2010 priori emission inventory and in the assimilation process itself, it is difficult to verify the accuracy of the 2015 and 2016 January analyzed emissions. The "bottom-up" emission inventory for the two years is not yet available for comparison. Thus, two sets of forecast experiments using the priori emissions and the analyzed emissions were conducted (NO_DA_forecast vs. EMIS_DA_forecast, see details in section 2.5). The forecast differences between the two experiments can reflect, to some extent, the performance/improvement of the analyzed emissions. To show the differences spatially, the statistics at single observational sites in the two forecast experiments are given and compared. In addition, the improvement from the hourly forecast is more meaningful in showing the system capability of hourly emission optimization. Thus, the time series of the regional means in 8 regions are also given to show the performance temporally.

**5.1 Changes of spatial statistics**

Figure 11 and 12 show the performances of the NO_DA_forecast and EMIS_DA_forecast experiments for January of 2015 and 2016, respectively. Statistics, including the BIAS (bias, equal to the difference between the modeled value and the observational value, representing the overall model tendency), RMSE (root mean square error/root mean square deviation, equal to the square root of the second moment of the differences between the model values and the observational values, reflecting both model biases and error variances) and CORR (correlation coefficient, equal to the linear relationship between the modeled values and the observational values), were chosen to evaluate the two forecast experiments with priori emissions and analyzed emissions, respectively. For a single site, the three statistics (BIAS, RMSE and CORR) may change in two directions—for example, the BIAS (bias of the absolute emission amount) may get worse, but the RMSE (error variance) and CORR (in terms of the diurnal or day-to-day emission changes) may get better. To fairly evaluate and show the overall changes, the 531 lumped sites were classified into five different groups

to reflect the differences of the statistics. The classification and performance are listed in Table 4. The spatial

distributions of the NO_DA_forecast statistics for each site are given in Fig. 11a and Fig. 12a. To better

illustrate the changes of the statistics after applying the analyzed emissions, the differences

(EMIS_DA_forecast − NO_DA_forecast), instead of the absolute values, are shown for the five defined

groups in Fig. 11b-11f and Fig. 12b-12f. Specifically, the absolute values of the BIAS were used in the

difference calculation.

Concerning single statistics, the BIAS, RMSE and CORR are improved at 383, 444 and 426 sites

respectively for the year 2015 (Table 4), while the total valid sites are 524 in the whole domain. That is to say

that the ratios of sites improved are 73%, 85% and 81%, respectively, as determined using BIAS, RMSE and

CORR as the single criterion. When considering the overall performances using the three statistics, 300 sites

(57%) are fully improved (BIAS/RMSE decrease and CORR increase), 138 sites (26%) are partially improved

(either the BIAS and RMSE improved or the RMSE and CORR improved), only 16 sites (3%) are overall

worse and the remaining approximately 13% of sites could not be justified. The performance in 2016 is even

better than that in 2015, with the fully improved/overall worse sites being more/less, respectively, compared

with the 2015 case.

Figure 11b shows that overall improvements are achieved in the whole domain, with the largest BIAS

corrections occurring at the sites in the Sichuan Basin, Central China, Yangtze River Delta and Pearl River

Delta (reaching 60-70% reductions) and the largest CORR improvement occurring in Xinjiang (reaching 0.35).

The sites that are partially improved (Fig. 11c, d) and unclassified (Fig. 11e) are not in specific regions but are

scattered through the whole domain. The sites that became overall worse (Fig. 11f) are very few, and the

variances are relatively small. Consistent with Table 4, the performance in 2016 (Fig. 12) is even better than

that in 2015 (Fig. 11), with the bias corrections being more significant, especially in the Sichuan Basin, Central

China, the Yangtze River Delta and the Pearl River Delta, and the CORR improvements are even larger in

Xinjiang.

## 5.2 Time series of the regional mean

Figure 13 shows the time series of the regional mean forecasts (NO_DA_forecast and EMIS_DA_forecast) and the observed $SO_2$ concentrations in 8 regions for 2015 and 2016. From the aspect of the regional mean, forecasts with priori emissions are severely overestimated in southern China (the Sichuan Basin, Central China, the Yangtze River Delta, and the Pearl River Delta), and the overestimations are largely corrected in the forecasts with analyzed emissions. For Northeastern China, the Energy Golden Triangle and Xinjiang, forecasts with priori emissions are underestimated, and forecasts with analyzed emissions helped to correct the biases. It is surprising to see that the regional averages in the North China Plain match well with the observations, although the site-to-site comparisons (Fig. 11 and 12) show large biases at single sites. As the sites in one region are averaged, the positive/negative biases among different sites might be offset in the averaged time series. For this reason, the RMSE and CORR of all the hourly data in one region are also calculated for verification.

The statistics of the BIAS, RMSE and CORR in the 8 regions are given in Table 5. From the aspect of the regional mean, the improvements obtained after applying the analyzed emissions are more significant in southern China than in northern China, with the RMSE decreased by 27.9-39.3%, the BIAS decreased by 63.3%-78.2%, and the CORR increased by 16.7%-45.0% for the year 2015. For northern China, although the improvements are not so large, the BIAS still decreased (except in the North China Plain), and the decreasing ratio ranged from 6.3% to 22.9%, while the RMSE decreased by 4.2-8.8% and the CORR increased by 7.7% to 366.7%. The largest CORR increase occurred in Xinjiang, changing from 0.06 to 0.28, indicating that the newly added emission sources in the analyzed emissions are necessary. Compared to 2015, the improvements in 2016 are also larger, which is consistent with previous discussions.

**6. Conclusions**

Based upon our previous study (Peng *et al.* 2017), we further updated the WRF-Chem/EnKF DA system to quantitatively estimate gridded hourly $SO_2$ emissions using hourly surface observations as constraints. Different from Peng *et al.* (2017), direct emissions instead of emissions scaling factors were used as the analysis variables, which allows for the detection of new emissions sources.

The 2010 January MEIC priori emissions were used to generate 2015 and 2016 January analyzed emissions, applying the hourly surface $SO_2$ observations as constraints. Compared with the 2010 priori emissions, the analyzed emissions in January 2015 showed inhomogeneous change patterns in different regions. 1) Significant emissions reductions were found in southern China, including in the Sichuan Basin, Central China, the Yangtze River Delta and the Pearl River Delta; however, there were still some grids with slight emissions increases surrounding larger cities, indicating the emission transition due to urbanization development. The reduction ratios of the total January emissions for the aforementioned four regions were -10.5%, -9.9%, -13.8% and -22.9%, respectively. 2) For northern China (the Northern China Plain and Northeastern China), the situation is more complicated during the winter heating season. Comparisons show large emissions reductions in larger cities but wide increases in surrounding suburban and rural regions, which may indicate missing raw coal combustions not taken into account in the priori emission inventory. The increase ratios of the total January emissions for the Northern China Plain and Northeastern China were 12.7% and 49.4%, respectively. 3) Significantly large emissions increases were found in western China (the Energy Golden Triangle and Xinjiang) due to the energy expansion strategy, which is consistent with satellite observations (e.g., Ling *et al.*, 2017). The increase ratio of the total January emissions for the Energy Golden Triangle and Xinjiang were 25.6%, and 72.0%, respectively. It should be noted that the comparisons between the 2010 priori emissions and the 2015 analyzed emissions not only reflect the changes during the five years but also include the uncertainties in the priori emissions (either due to uncertainties in the total annual/monthly

emissions or to the allocation process from the provincial emissions to the gridded data). Comparisons of the 2015 and 2016 analyzed emissions show wide emissions reductions from 2015 to 2016, which is consistent with a recent study on emissions changing trends using the "bottom-up" approach (Zheng *et al.,* 2018), indicating that stricter control strategies have been fully executed nationwide. These changes coincided with the period of the energy development national strategy in northwestern China and regulations for the reduction of $SO_2$ emissions, indicating that the updated DA system was possibly capable of detecting the emissions deficiencies, dynamically updating the spatial-temporal emission changes (2010 to 2015/2016), and locating the newly added sources. The detection of emissions changes by the DA system can be localized to the city level, benefitting from the intensive observations and the model grid resolution.

It is difficult to verify the accuracy of the analyzed emissions, as the "bottom-up" emissions inventories for 2015 and 2016 are not yet available for comparison. Two sets of forecast experiments using the priori emissions and the analyzed emissions were conducted to show the differences and improvements. Among the lumped 531 sites, 300 sites were fully improved (BIAS and RMSE reduced and CORR increased), and only 16 sites were entirely worse for the year 2015. The other 138 sites were partially improved (two statistics became better). The improvements were much larger in southern China than in northern and western China. Upon using the analyzed emissions, the BIAS and RMSE were reduced by 61.8%-78.2% and 27.9%-52.2%, respectively, and the correlation coefficient increased by 12.5% - 47.1% for southern China regions (the Sichuan Basin, Central China, the Yangtze River Delta, and the Pearl River Delta). However, for northern and western China, where the original BIAS and RMSE values were larger, the decreases were relatively smaller. Nevertheless, the correlations were indeed improved, especially for Xinjiang, as new emissions were captured in the analyzed emissions. The distinction of the comparisons among different regions (northern/western regions vs. southern regions) indicated the deficiencies of the perturbation procedure in the DA system when applied to the northern/western regions. Further investigations should be conducted to generate larger spreads

for those regions in future studies.

Our study serves as an example indicating that the ensemble Kalman filter algorithm combinned with the WRF-Chem regional model can be used to optimize model-ready gridded hourly emissions inputs by using hourly surface observations as constraints. This approach is useful for assessing emissions control strategies and can also improve forecasting skills. The limitation of this study is that the analyzed emissions are still model-dependent, as the ensembles are conducted through the WRF-Chem model, and thus, the performance of the ensembles is model-dependent. Changes in the model configuration (e.g., the spatial resolution or chemistry options) can cause differences in the DA system. In our study, the model resolution is 40 km, which might be too coarse for $SO_2$, as it's a relatively short-lifetime specie, and the localized characteristics might not be captured by the system. In addition, the reactions of $SO_2$ are only reflected in the WRF-Chem system and not in the EnKF process. Considering the reaction time of $SO_2$ in the ambient atmosphere, there might be some time lag in the hourly emission factors.

**Author contributions**

ZL and DC designed research; DC performed research; JB contributed towards development of DA system; MC provides funds; DC wrote the paper, with contributions from all co-authors.

**Acknowledgement**

This work was supported by the National Key R&D Program on Monitoring, Early Warning and Prevention of Major Natural Disasters under grant (2017YFC1501406), the National Natural Science Foundation of China (Grant No. 41807312) and Basic R&D special fund for central scientific research institutes (IUMKYSZHJ201701). NCAR is sponsored by US National Science Foundation. We would like to thank Zhen Peng for useful discussions and patient guide on DA system development.

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

**Tables and Figures**

**Table 1.** WRF-Chem model configuration.

**Table 2.** Experiments conducted in this study and the three groups of comparisons.

**Table 3.** Priori and analyzed January emissions and the changing ratios for 8 regions (units: $10^6$ kg per day)

**Table 4.** Overall statistics changes of the EMIS_DA_FCST experiment compared with the NO_DA_FCST
experiment

**Table 5.** Statistics of the EMIS_DA_FCST and NO_DA_FCST experiments in 8 regions (units: µg m$^{-3}$ for
BIAS and RMSE)

**Figure 1.** Flow chart of the data assimilation system that simultaneously optimizes the initial chemical
conditions and emissions.

**Figure 2.** Spatial distribution of the priori SO$_2$ emissions used in this study. Regions defined in red rectangles
are: a-NCP (the North China Plain), b-NEC (Northeastern China), c-EGT (the Energy Golden Triangle), d-XJ
(Xinjiang), e-SB (the Sichuan Basin), f-CC (Central China), g-YRD (the Yangtze River Delta), and h-PRD
(the Pearl River Delta). The color bars of the regional plots have the same meaning as those in the national
one. Units: mol km$^{-2}$ h$^{-1}$.

**Figure 3.** Observed and modeled monthly average SO$_2$ concentrations for January in 2015 (Left) and 2016
(right). (a) Observations, where the ranges of the different colors are the same as those of the color bars of
(b) and (c); (b) NO_DA, (c) CONC_DA, and (d) CONC_DA-NO_DA. Units: µg m$^{-3}$.

**Figure 4.** The spatial distribution of the statistics between the model simulations and the observations for
(a) January 2015 and (b) January 2016. Top: NO_DA vs. observation, bottom: CONC_DA vs. observation.
Units: µg m$^{-3}$ for BIAS and RMSE.

**Figure 5.** Observed and modeled SO$_2$ ambient concentration changes (January 2016 - January 2015). (a)
Observations, where the ranges of different colors are the same as those of the color bars of (b) and (c), (b)
NO_DA, and (c) CONC_DA. (Units: µg m$^{-3}$)

**Figure 6.** Regional averaged RMSE and the total spread for (a) January 2015 and (b) January 2016 in 8
regions. Time starts from 00UTC. (Units: µg m$^{-3}$).

**Figure 7.** Analyzed emissions for (a) January 2015 and (b) January 2016. (c) The differences of the 2015-
2010_prior and (d) ratios of the 2015/2010_priori. (e) The differences of the 2016-2015 and (f) ratios of
2016/2015. Units are mol km$^{-2}$ h$^{-1}$ for (a), (b), (c) and (e).

**Figure 8.** The differences between the analyzed 2015 January emissions and the 2010 priori emissions in 8
regions. Left panels are emission differences of 2015-2010 (units: mol km$^{-2}$ h$^{-1}$), and right panels are the ratios
of 2015/2010 in each region.

**Figure 9.** Same as Figure 8, but for the differences between the analyzed 2016 January emissions and the 2015 January emissions in 8 regions. Left panels are emission differences of 2016-2015 (units: mol km$^{-2}$ h$^{-1}$), and right panels are the ratios of 2016/2015 in each region.

**Figure 10.** Hourly factors in the priori emission inventory and those derived from the EMIS_DA experiment in 8 regions.

**Figure 11.** The spatial distribution of the error statistics between the model simulations and the observations for January 2015. (a) Statistics between NO_DA_FCST and the observations, with BIAS and RMSE in µg m$^{-3}$; (b)-(f) the statistics improvements from NO_DA_FCST to EMIS_DA_FCST for different groups of sites (classification in Table 4), with the BIAS and RMSE improvements in percentages. The color bars are all the same for (b)-(f) and are shown only in (c) and (f) to save space.

**Figure 12.** Same as Figure 11, but for January 2016.

**Figure 13.** Time series of the regional mean SO$_2$ concentrations from the observations and from model simulations with the priori and analyzed (posterior) emissions for (a) January 2015 and (b) January 2016 in 8 regions. Time starts from 00UTC. (Units: µg m$^{-3}$)

**Table 1.** WRF-Chem model configuration.

| | |
|---|---|
| Aerosol scheme | MOSAIC (4 bins) (Zaveri *et al.*, 2008) |
| Photolysis scheme | Fast-J (Wild *et al.*, 2000) |
| Gas phase chemistry | CBM-Z (Zavier *et al.*, 1999) |
| Cumulus parameterization | Grell 3D scheme |
| Short-wave radiation | Goddard Space Flight Center Shortwave radiation scheme (Chou and Suarez, 1994) |
| Long-wave radiation | RRTM (Mlawer *et al.*, 1997) |
| Microphysics | Single-Moment 6-class scheme (Grell and Devenyi, 2002) |
| Land-surface model | NOAH LSM   (Chen and Dudhia, 2001) |
| Lan use type | USGS 2m (kept the same for 2010-2015-216) |
| Boundary layer scheme | YSU   (Hong *et al.*, 2006) |
| Meteorology initial and boundary conditions | GFS analysis and forecast data at 6-hr frequency for control experiment, interpolated at 1-hr frequency for hourly assimilation experiments and forecast experiments |
| Initial condition for chemical species | 11-day spin-up |
| Boundary conditions for chemical species | averages of mid-latitude aircraft profiles (McKeen *et al.*, 2002) |
| Dust and sea salt Emissions | GOCART (Chin *et al.,* 2000, 2002) |

**Table 2.** Experiments conducted in this study and the three groups of comparisons.

| | | Design of the simulation | Purpose of the simulation | Purpose of the comparisons |
|---|---|---|---|---|
| Control experiments | NO_DA | 6-hr WRF-Chem cycling run with prior MEIC_2010 | Generate 2015-2016 concentration fields assuming the same emissions as in 2010 prior emissions | Concentrations from CONC_DA versus that from NO_DA: As the concentrations from CONC_DA is very close to the observations, the concentration differences between CONC_DA and NO_DA possibly indicated a model deficiency in reproducing the reality, which was mainly from the emissions changes from 2010 to 2015-2016. The assumption is that the GFS 6-hr analysis data provide good meteorological IC/BC values and that the model accurately simulated the meteorology conditions; thus, the emissions were the major deficiency in the model. |
| DA experiments | CONC_DA | WRF-Chem (with prior MEIC_2010) and GSI 3D-var hourly DA cycle<br><br>Hourly observations were assimilated and WRF-Chem concentration output were updated | Generate 2015-2016 concentration reanalysis fields integrating hourly observations | |
| | EMIS_DA | WRF-Chem (with prior MEIC_2010 at the beginning and later with forecast emissions) and EnSRF hourly DA cycle<br><br>Hourly observations were assimilated, and WRF-Chem concentration output and emissions were updated | Generate 2015-2016 analyzed emissions with hourly observations as constraints | Updated emissions from EMIS_DA versus prior emissions: The emissions differences between the 2015 analyzed emissions and the 2010 priori emissions not only reflected the changes from 2010 to 2015 but also included the deficiencies in the 2010 priori emission.<br><br>Updated emissions between different years: The differences between the 2016 analyzed emissions and the 2015 analyzed emissions reflected the pure emissions changes from 2015 to 2016, since the deficiencies of the 2010 priori emissions were offset in the subtraction<br><br>The emissions control policies are discussed to investigate whether the emissions changes are reasonable. |

| Forecast experiments | NO_DA_forecast | 24-hr WRF-Chem forecast with prior MEIC_2010, chemistry IC from CONC_DA at 00UTC | The simulation with only improved initial condition | Concentrations from EMIS_DA_forecast versus that from NO_DA_forecast:<br>The benefit by using updated emissions can be quantitatively assessed. |
|---|---|---|---|---|
| | EMIS_DA_forecast | 24-hr WRF-Chem forecast with updated 2015-2016 emissions, chemistry IC from CONC_DA at 00UTC | The simulation with both improved initial condition and updated emissions | |

**Table 3.** Priori and analyzed January emissions and the changing ratios for 8 regions (units: $10^6$ kg per day)

|  | 2010_prior | 2015_posterior | 2016_ posterior | (2015-2010) /2010 | (2016-2015) /2015 |
|---|---|---|---|---|---|
| NCP | 16.23 | 18.29 | 17.33 | 12.7% | -5.3% |
| NEC | 4.12 | 6.16 | 5.30 | 49.4% | -13.9% |
| ETR | 11.01 | 13.82 | 13.01 | 25.6% | -5.9% |
| XJ | 1.62 | 2.79 | 2.34 | 72.0% | -16.1% |
| SB | 17.12 | 15.33 | 13.43 | -10.5% | -12.4% |
| CC | 9.95 | 8.96 | 7.57 | -9.9% | -15.5% |
| YRD | 5.80 | 5.00 | 4.65 | -13.8% | -7.0% |
| PRD | 1.82 | 1.40 | 1.23 | -22.9% | -12.6% |

**Table 4.** Overall statistics changes of the EMIS_DA_FCST experiment compared with the NO_DA_FCST experiment

| Types | BIAS | RMSE | CORR | | |
|---|---|---|---|---|---|
| 2015 | | | | | |
| Better | 383 | 444 | 426 | | |
| Worse | 141 | 80 | 97 | | |
| 2016 | | | | | |
| Better | 375 | 444 | 456 | | |
| Worse | 148 | 79 | 67 | | |
| | | | | | |
| Groups | BIAS | RMSE | CORR | 2015 | 2016 |
| A.  Overall improved | decrease | decrease | increase | 300 | 321 |
| B.  Partially improved (BIAS, RMSE) | decrease | decrease | decrease | 61 | 43 |
| C.  Partially improved (RMSE, CORR) | increase | decrease | increase | 77 | 71 |
| D.  Not justified | | | | 70 | 77 |
| E.  Overall worse | increase | increase | decrease | 16 | 11 |

**Table 5.** Statistics of the EMIS_DA_FCST and NO_DA_FCST experiments in 8 regions (units: µg m$^{-3}$ for BIAS and RMSE)

| | N sites | N data | BIAS | | | RMSE | | | CORR | | |
|---|---|---|---|---|---|---|---|---|---|---|---|
| | | | NO_DA | EMIS_DA | Changes (%) | NO_DA | EMIS_DA | Changes (%) | NO_DA | EMIS_DA | Changes (%) |
| **2015** | | | | | | | | | | | |
| NCP | 67 | 46699 | -9.6 | -10.1 | 5.2% | 53.7 | 49.0 | -8.8% | 0.52 | 0.62 | 19.2% |
| NEC | 30 | 20910 | -29.3 | -22.6 | -22.9% | 61.8 | 57.1 | -7.6% | 0.52 | 0.56 | 7.7% |
| EGT | 45 | 31365 | -41.2 | -38.6 | -6.3% | 84.8 | 81.2 | -4.2% | 0.53 | 0.58 | 9.4% |
| XJ | 19 | 13243 | -12.6 | -10.3 | -18.3% | 36.8 | 33.7 | -8.4% | 0.06 | 0.28 | 366.7% |
| SB | 48 | 33456 | 9.7 | 2.7 | -72.2% | 45.1 | 32.5 | -27.9% | 0.20 | 0.29 | 45.0% |
| CC | 53 | 36941 | 6.1 | -1.4 | -77.0% | 49.7 | 34.6 | -30.4% | 0.32 | 0.39 | 21.9% |
| YRD | 34 | 23698 | 10.9 | 4.0 | -63.3% | 37.0 | 24.9 | -32.7% | 0.47 | 0.55 | 17.0% |
| PRD | 20 | 13940 | 8.7 | 1.9 | -78.2% | 24.7 | 15.0 | -39.3% | 0.42 | 0.49 | 16.7% |
| **2016** | | | | | | | | | | | |
| NCP | 67 | 46699 | 2.1 | -0.3 | -85.7% | 41.5 | 36.2 | -12.8% | 0.58 | 0.69 | 19.0% |
| NEC | 30 | 20910 | -16.8 | -14.7 | -12.5% | 41.2 | 36.9 | -10.4% | 0.50 | 0.58 | 16.0% |
| EGT | 45 | 31365 | -27.7 | -26.6 | -4.0% | 64.5 | 61.2 | -5.1% | 0.56 | 0.63 | 12.5% |
| XJ | 19 | 13243 | -5.8 | -6.0 | 3.4% | 30.5 | 26.9 | -11.8% | 0.23 | 0.47 | 104.3% |
| SB | 48 | 33456 | 14.5 | 5.2 | -64.1% | 38.9 | 23.1 | -40.6% | 0.17 | 0.25 | 47.1% |
| CC | 53 | 36941 | 11.2 | 2.6 | -76.8% | 38.0 | 22.2 | -41.6% | 0.28 | 0.37 | 32.1% |
| YRD | 34 | 23698 | 12.3 | 4.7 | -61.8% | 33.7 | 20.1 | -40.4% | 0.48 | 0.54 | 12.5% |
| PRD | 20 | 13940 | 9.8 | 2.4 | -75.5% | 20.9 | 10.0 | -52.2% | 0.30 | 0.39 | 30.0% |

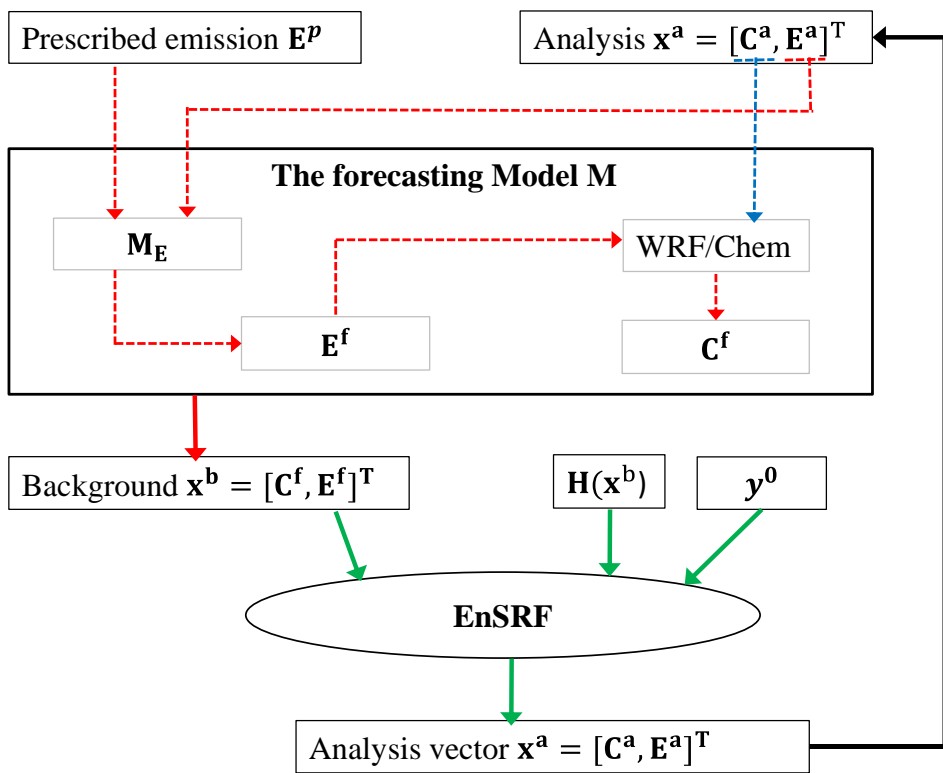

**Figure 1.** Flow chart of the data assimilation system that simultaneously optimizes the initial chemical conditions and emissions.

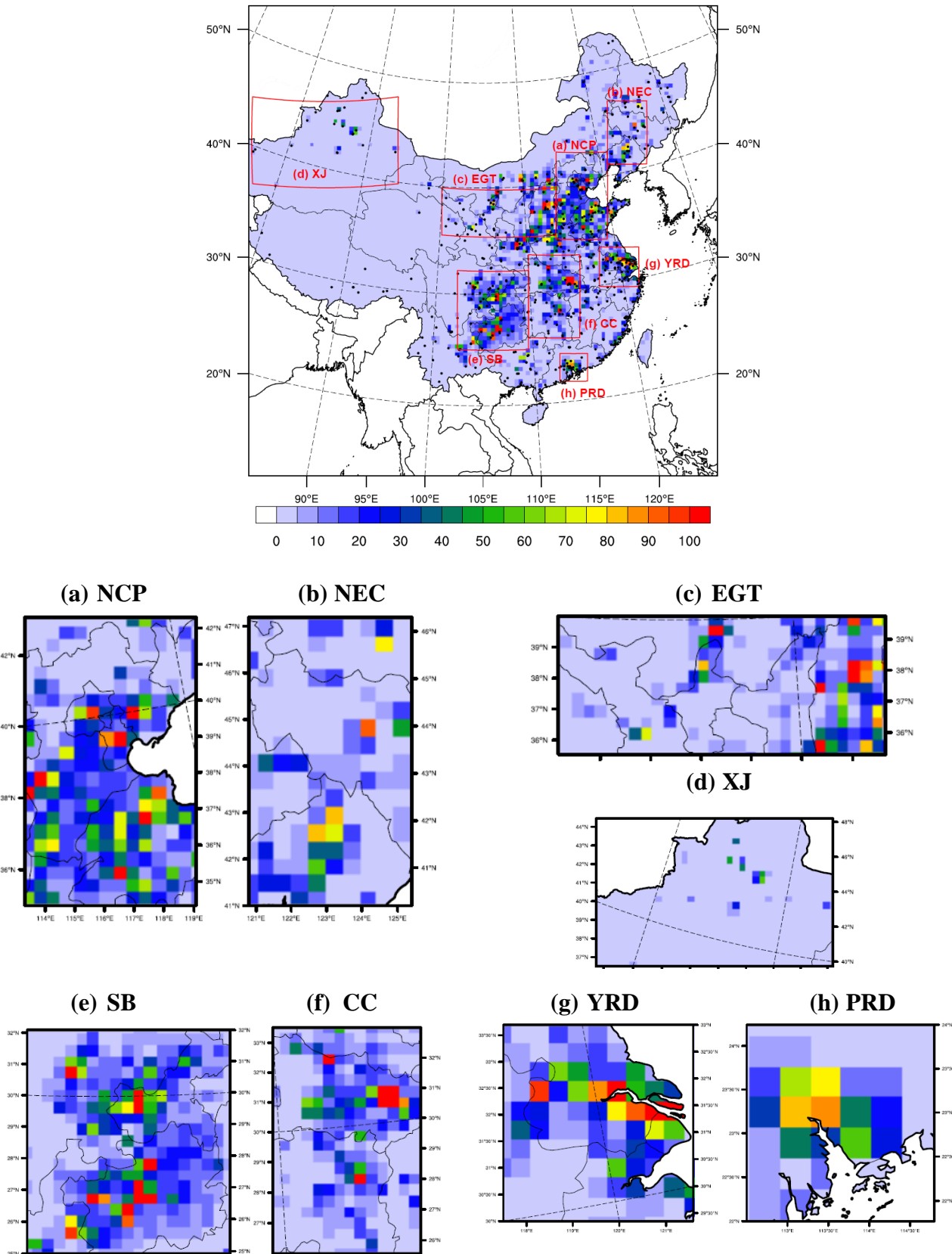

**Figure 2.** Spatial distribution of the priori SO₂ emissions used in this study. Regions defined in red rectangles are: a-NCP (the North China Plain), b-NEC (Northeastern China), c-EGT (the Energy Golden Triangle), d-XJ (Xinjiang), e-SB (the Sichuan Basin), f-CC (Central China), g-YRD (the Yangtze River Delta), and h-PRD (the Pearl River Delta). The color bars of the regional plots have the same meaning as those in the national one. Units: mol km⁻² h⁻¹.

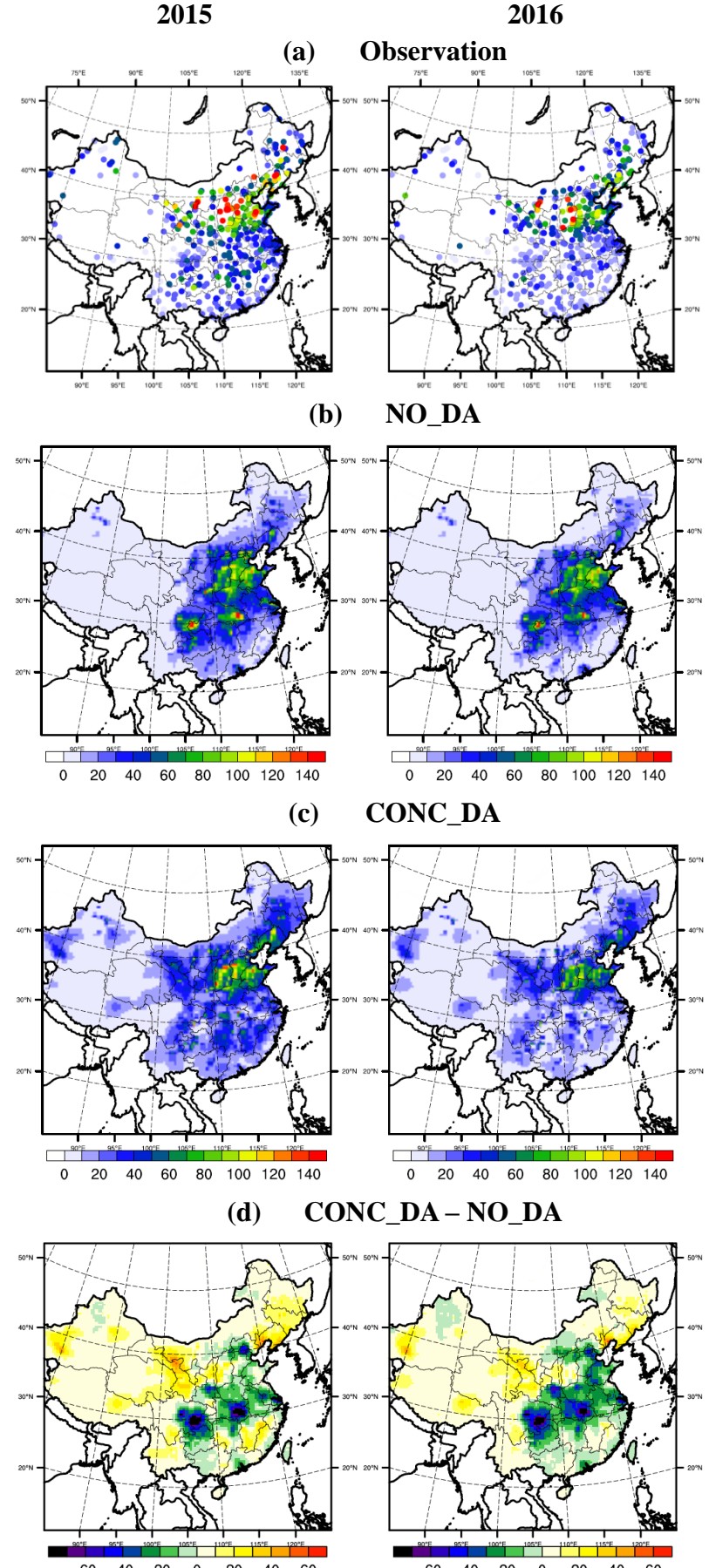

**Figure 3.** Observed and modeled monthly average $SO_2$ concentrations for January in 2015 (Left) and 2016 (right). (a) Observations, where the ranges of the different colors are the same as those of the color bars of (b) and (c); (b) NO_DA, (c) CONC_DA, and (d) CONC_DA-NO_DA. Units: $\mu g\ m^{-3}$.

**(a). 2015 - NO_DA (top) and CONC_DA (bottom)**

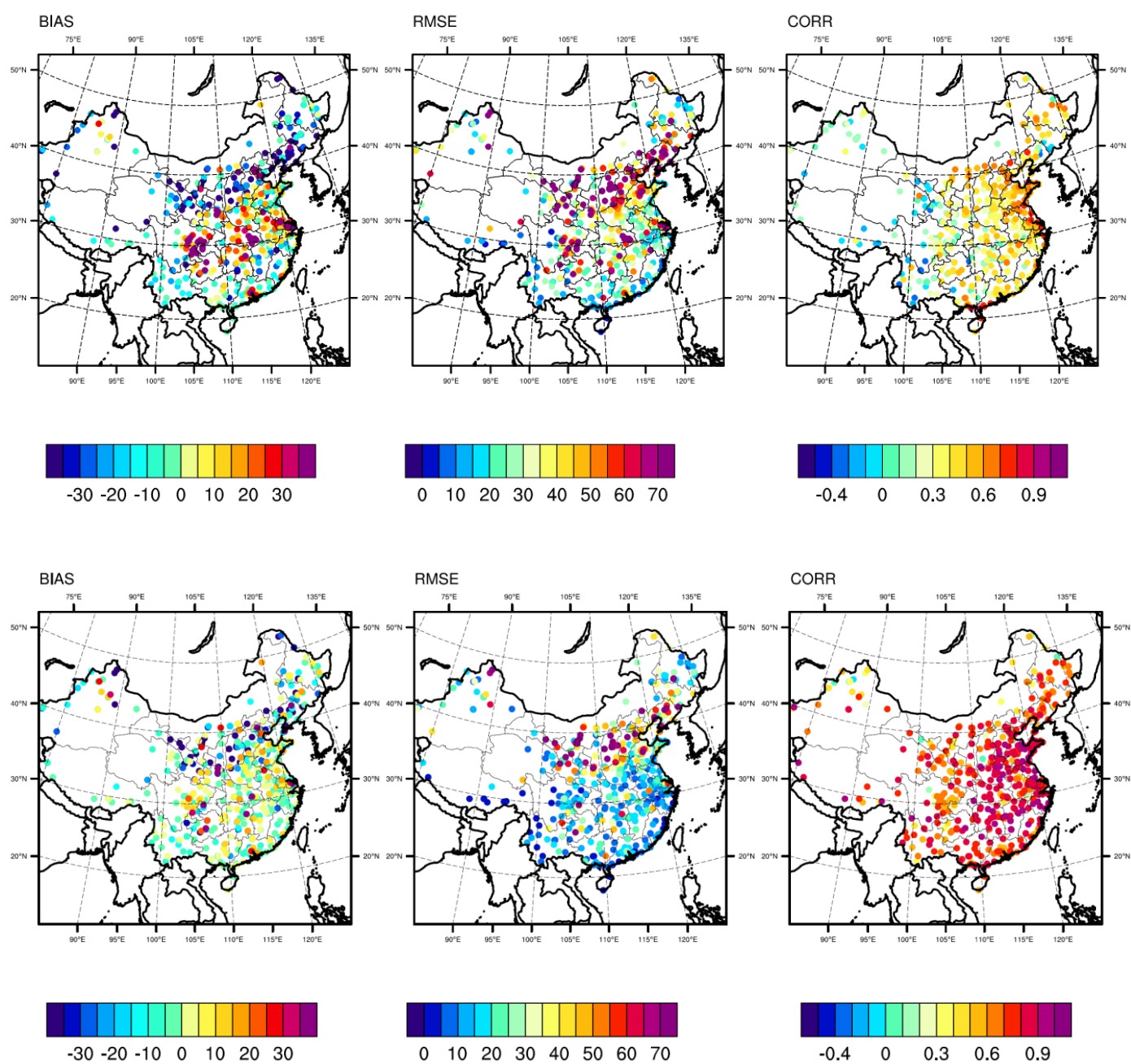

**Figure 4.** The spatial distribution of the statistics between the model simulations and the observations for (a) January 2015 and (b) January 2016. Top: NO_DA vs. observation, bottom: CONC_DA vs. observation. Units: μg m$^{-3}$ for BIAS and RMSE.

**(b). 2016 - NO_DA (top) and CONC_DA (bottom)**

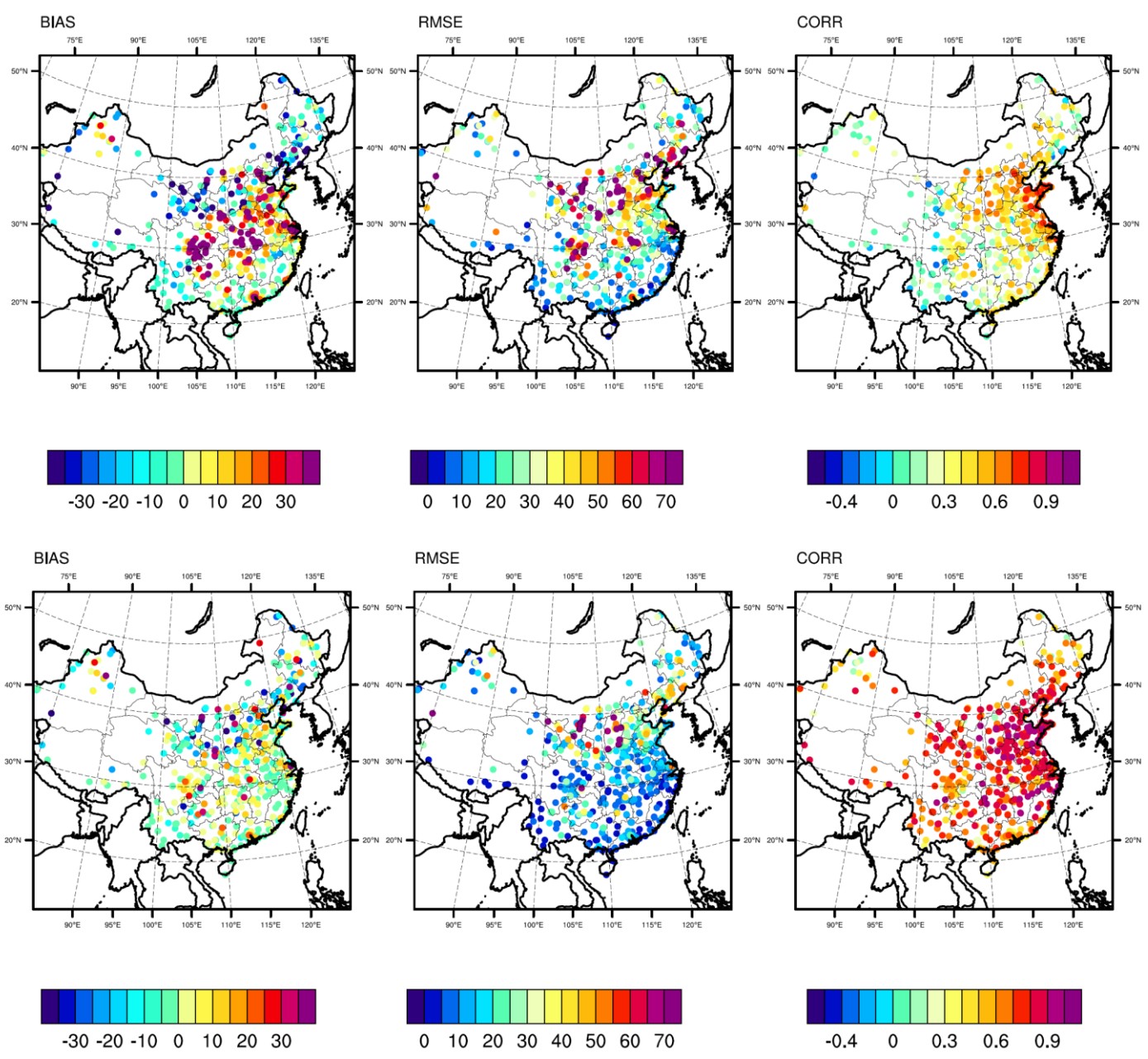

**Figure 4. (b)** Continue.

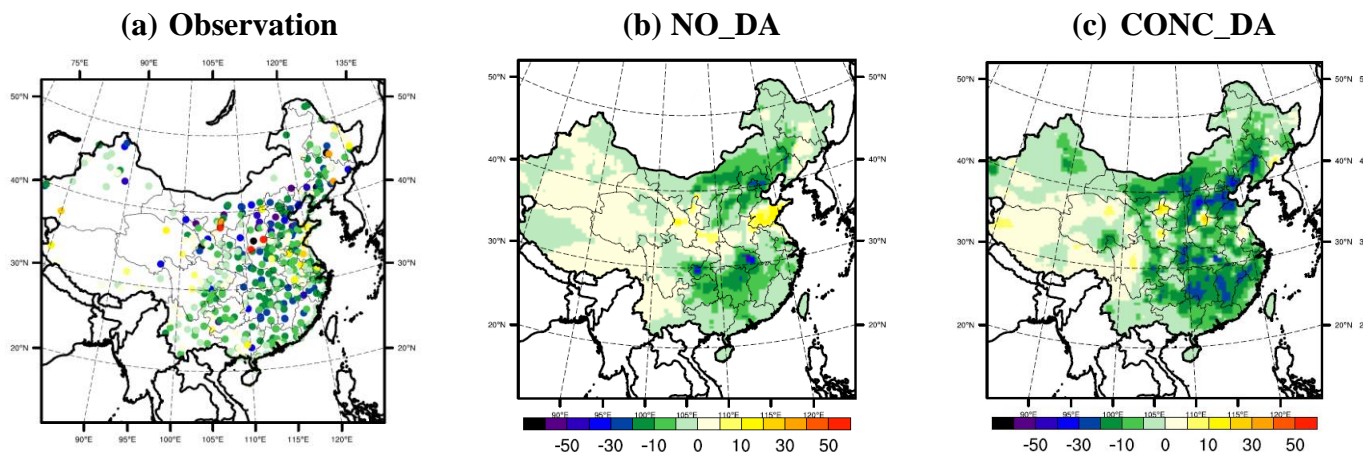

**Figure 5.** Observed and modeled SO$_2$ ambient concentration changes (January 2016 - January 2015). (a) Observations, where the ranges of different colors are the same as those of the color bars of (b) and (c), (b) NO_DA, and (c) CONC_DA. (Units: µg m$^{-3}$)

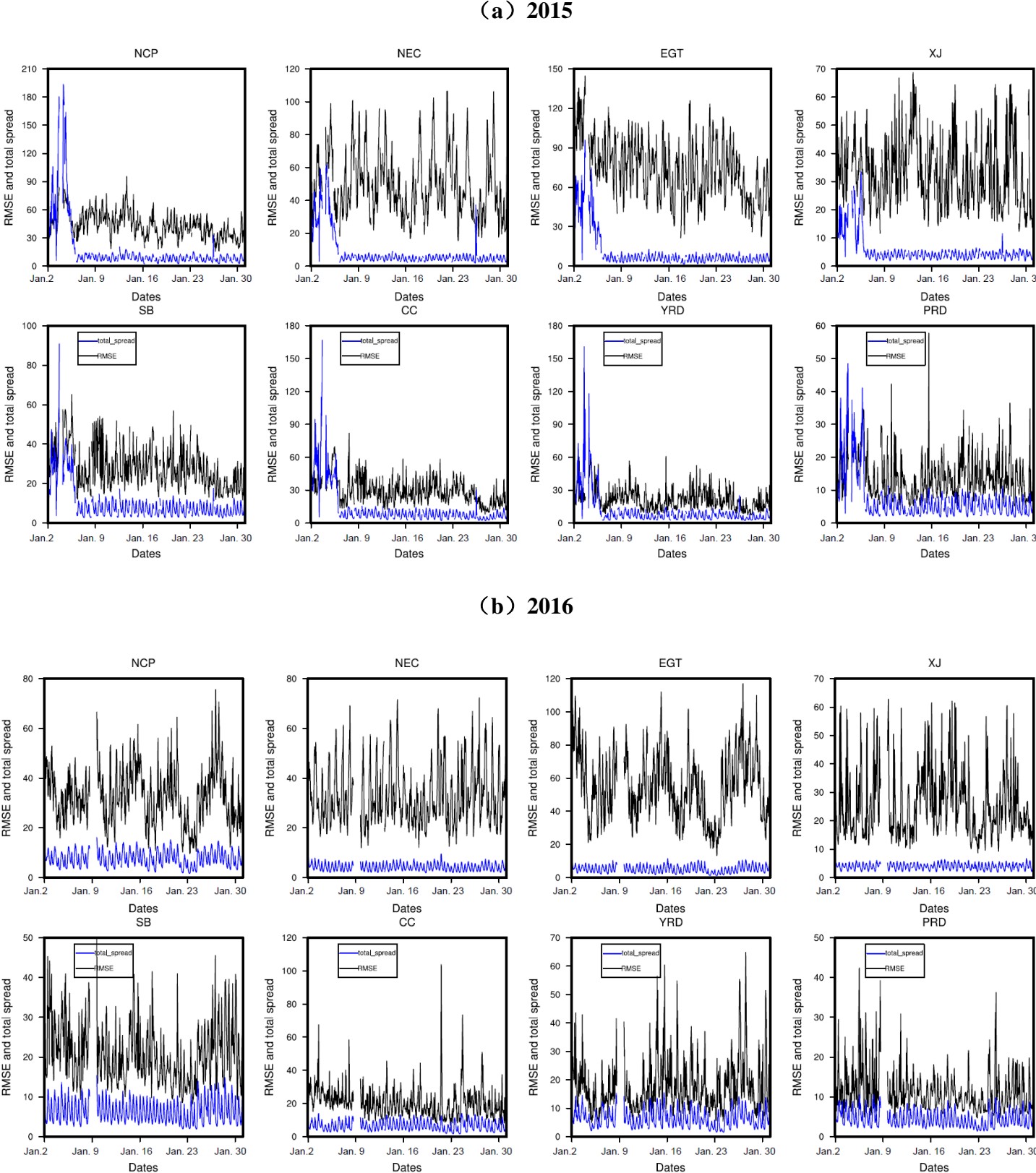

**Figure 6.** Regional averaged RMSE and the total spread for (a) January 2015 and (b) January 2016 in 8 regions. Time starts from 00UTC. (Units: $\mu g \, m^{-3}$).

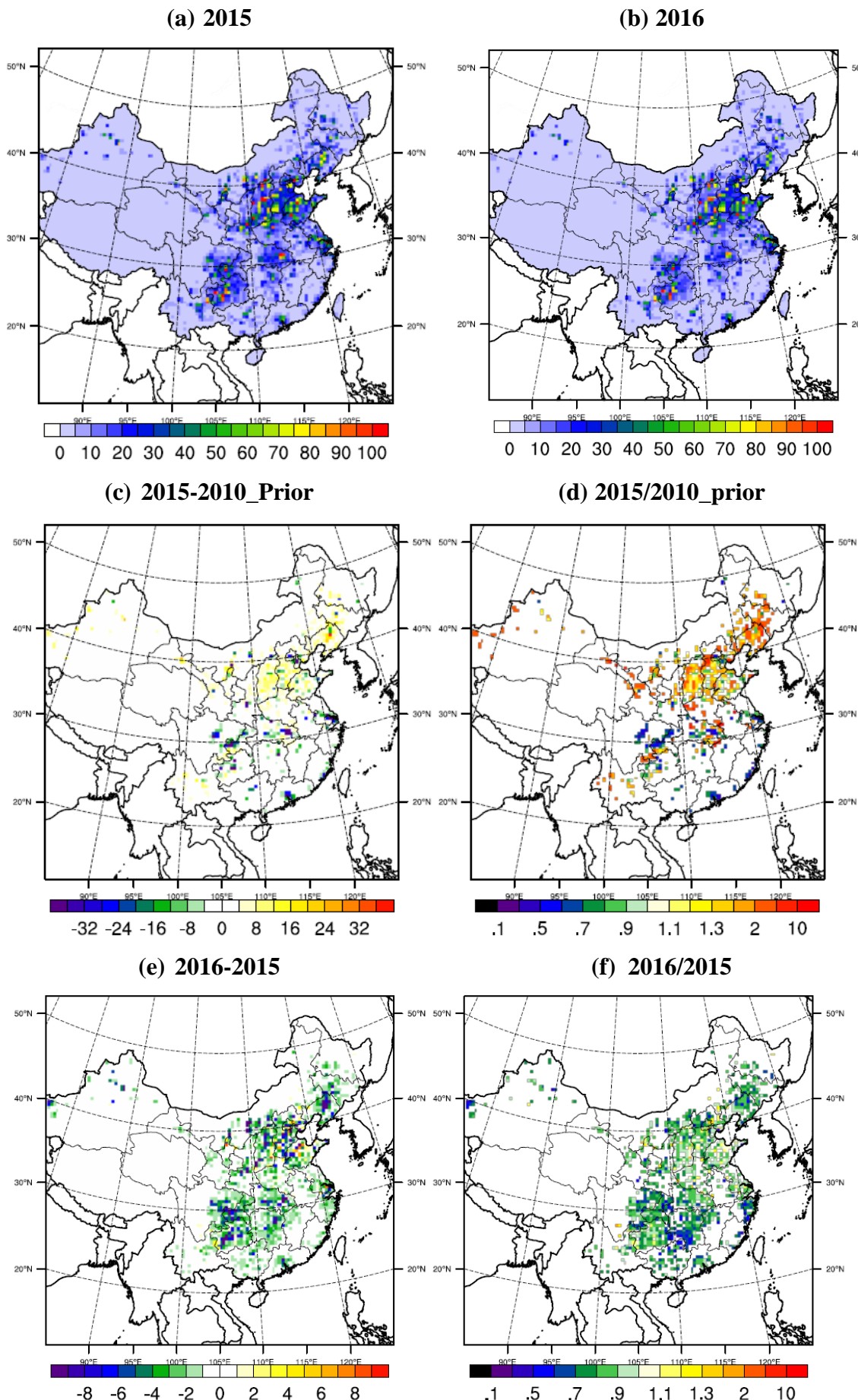

**Figure 7.** Analyzed emissions for (a) January 2015 and (b) January 2016. (c) The differences of the 2015-2010_prior and (d) ratios of the 2015/2010_priori. (e) The differences of the 2016-2015 and (f) ratios of 2016/2015. Units are mol km$^{-2}$ h$^{-1}$ for (a), (b), (c) and (e).

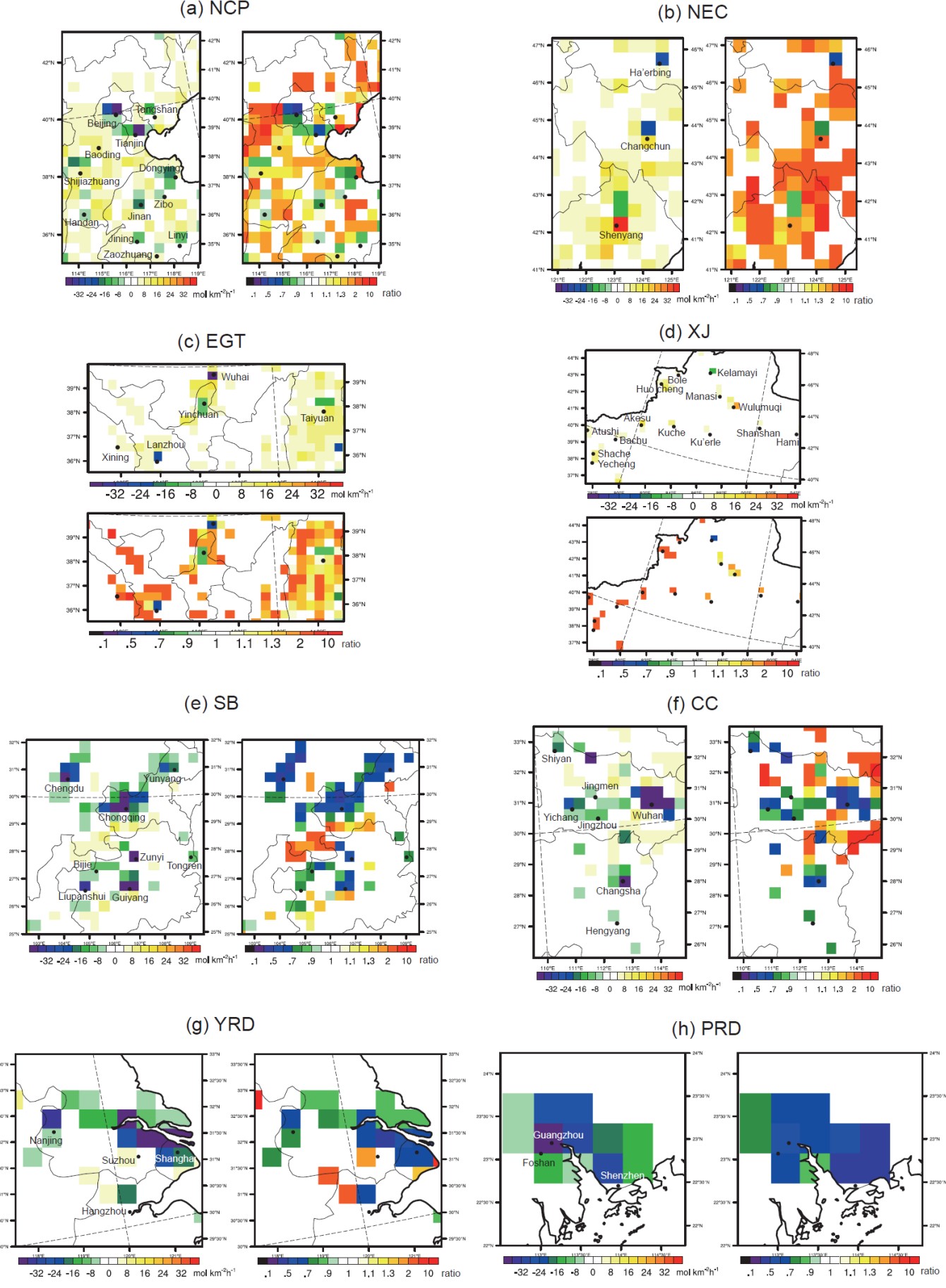

**Figure 8.** The differences between the analyzed 2015 January emissions and the 2010 priori emissions in 8 regions. Left panels are emission differences of 2015-2010 (units: mol km-2 h-1), and right panels are the ratios of 2015/2010 in each region.

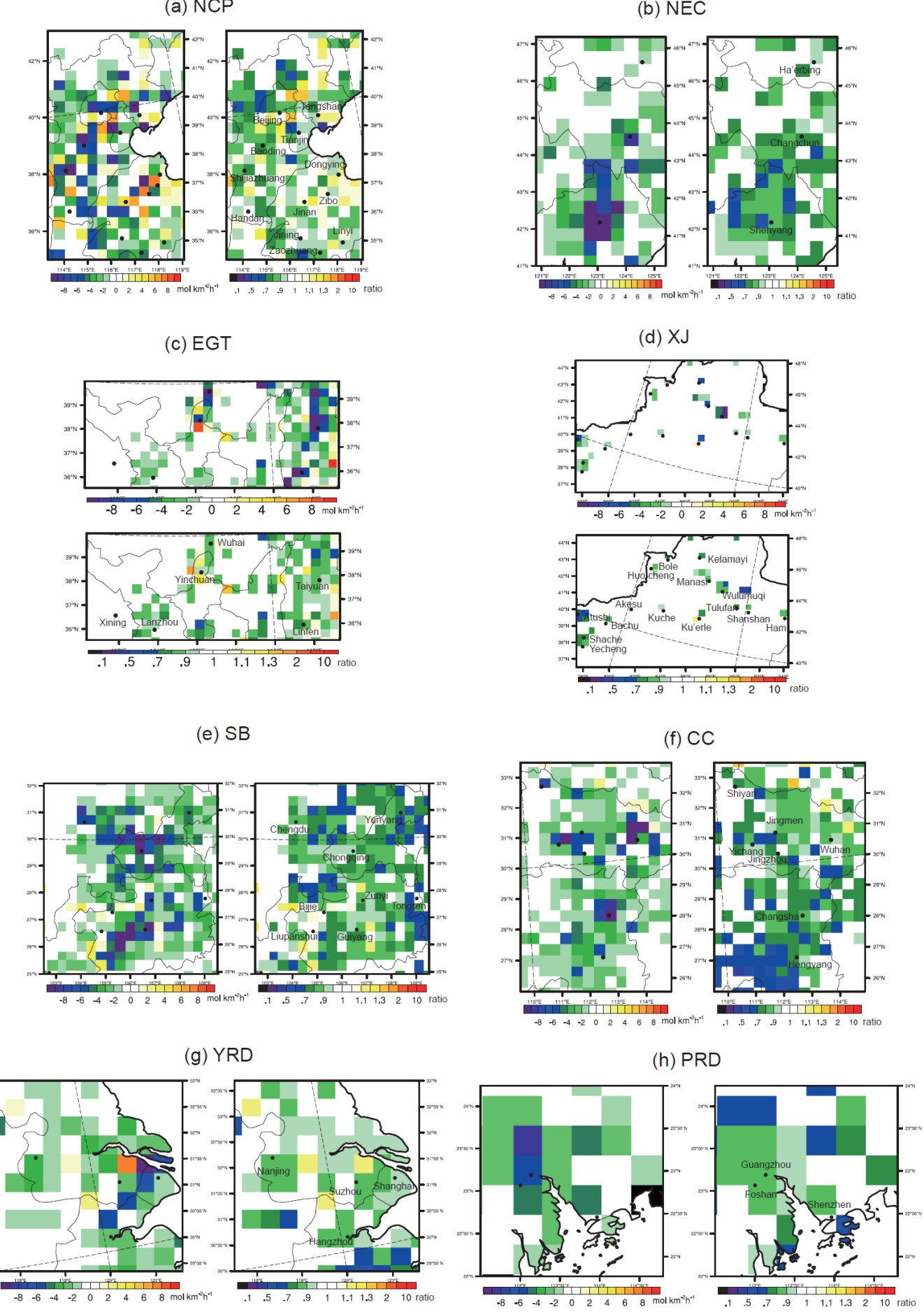

**Figure 9.** Same as Figure 8, but for the differences between the analyzed 2016 January emissions and the 2015 January emissions in 8 regions. Left panels are emission differences of 2016-2015 (units: mol km$^{-2}$ h$^{-1}$), and right panels are the ratios of 2016/2015 in each region.

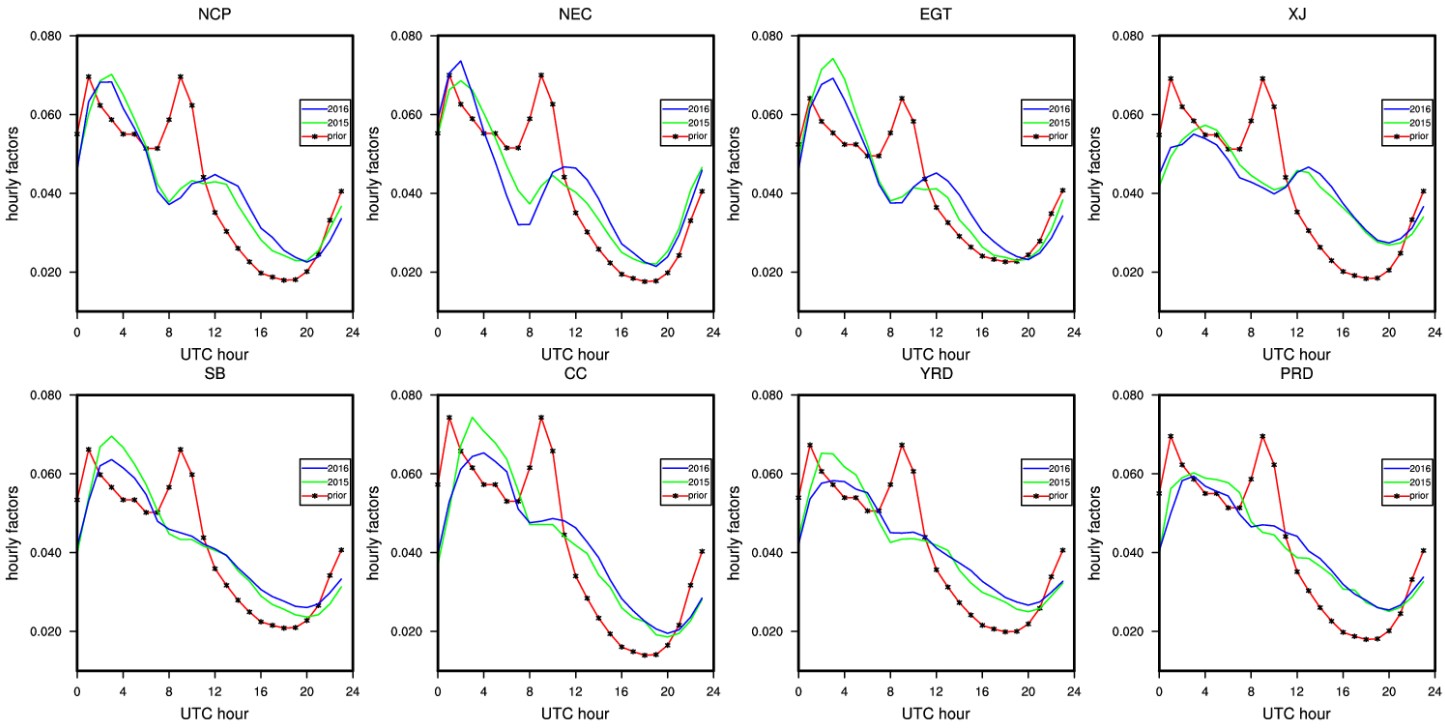

**Figure 10.** Hourly factors in the priori emission inventory and those derived from the EMIS_DA experiment in 8 regions.

# (a) NO_DA_FCST

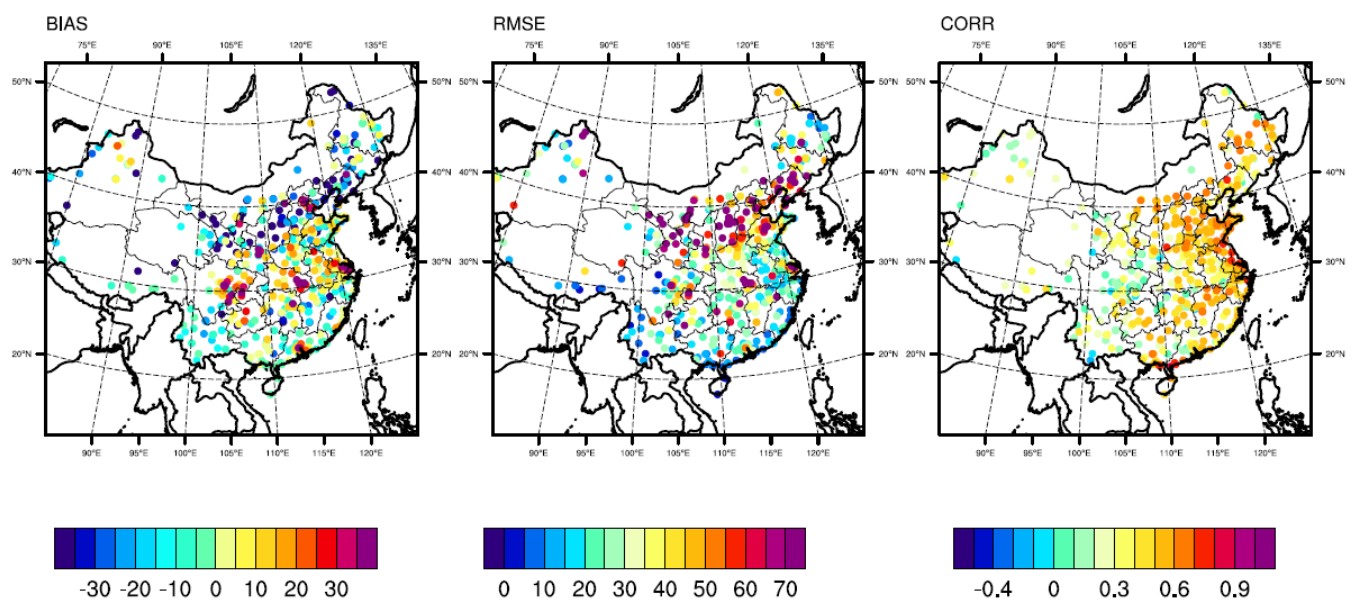

BIAS

RMSE

CORR

-30 -20 -10 0 10 20 30

0 10 20 30 40 50 60 70

-0.4 0 0.3 0.6 0.9

## (b) Differences (EMIS_DA_FCST - NO_DA_FCST) - Group A

**BIAS changes (%)**     **RMSE changes (%)**     **CORR changes**

## (c) Group B

-70 -40 -20 0 20 40 70

-70 -40 -20 0 20 40 70

-0.35 -0.15 0 0.15 0.35

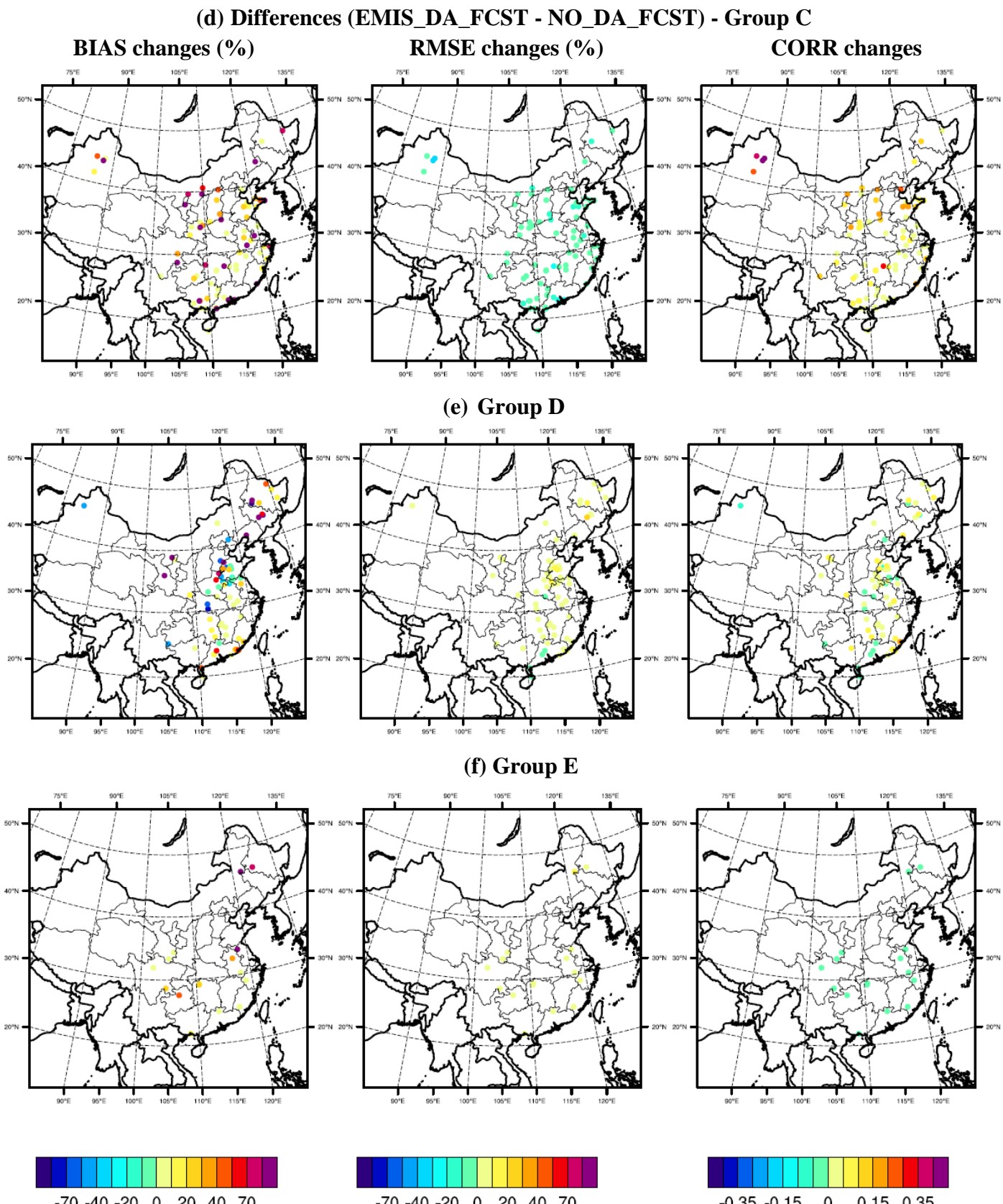

**Figure 11.** The spatial distribution of the error statistics between the model simulations and the observations for January 2015. (a) Statistics between NO_DA_FCST and the observations, with BIAS and RMSE in µg m$^{-3}$; (b)-(f) the statistics improvements from NO_DA_FCST to EMIS_DA_FCST for different groups of sites (classification in Table 4), with the BIAS and RMSE improvements in percentages. The color bars are all the same for (b)-(f) and are shown only in (c) and (f) to save space.

**(a) NO_DA_FCST**

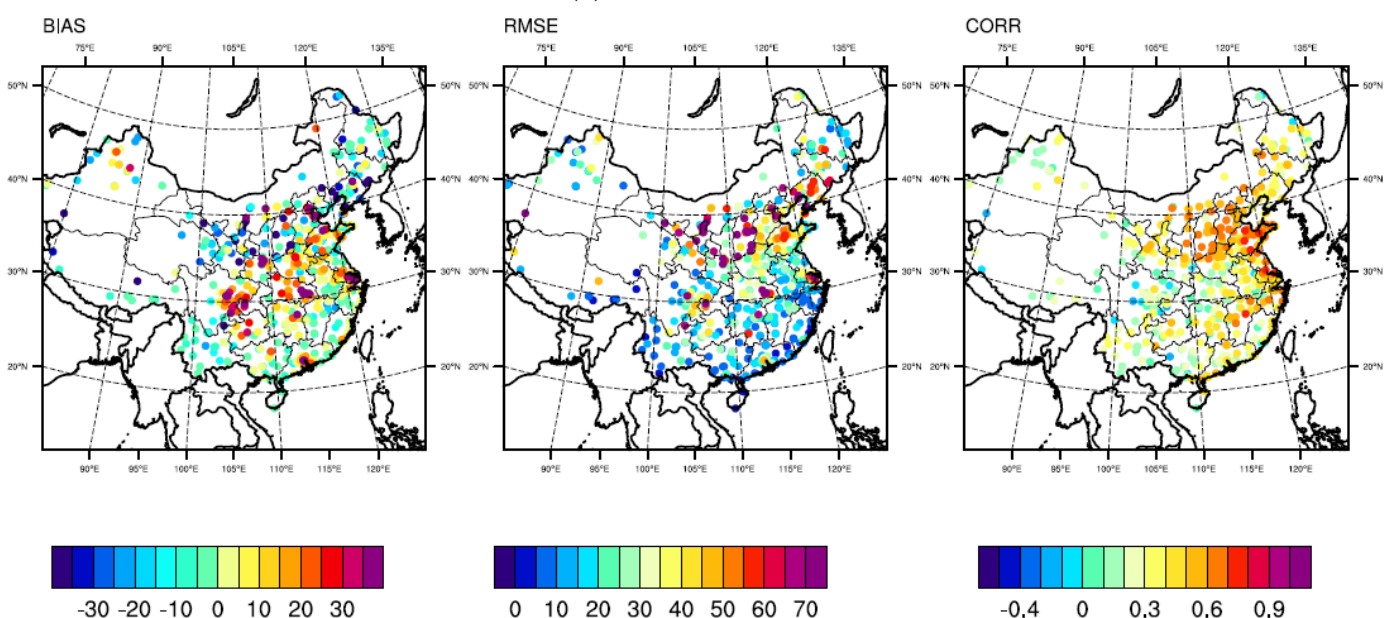

**(b) Differences (EMIS_DA_FCST - NO_DA_FCST) - Group A**

BIAS changes (%)     RMSE changes (%)     CORR changes

**(c) Group B**

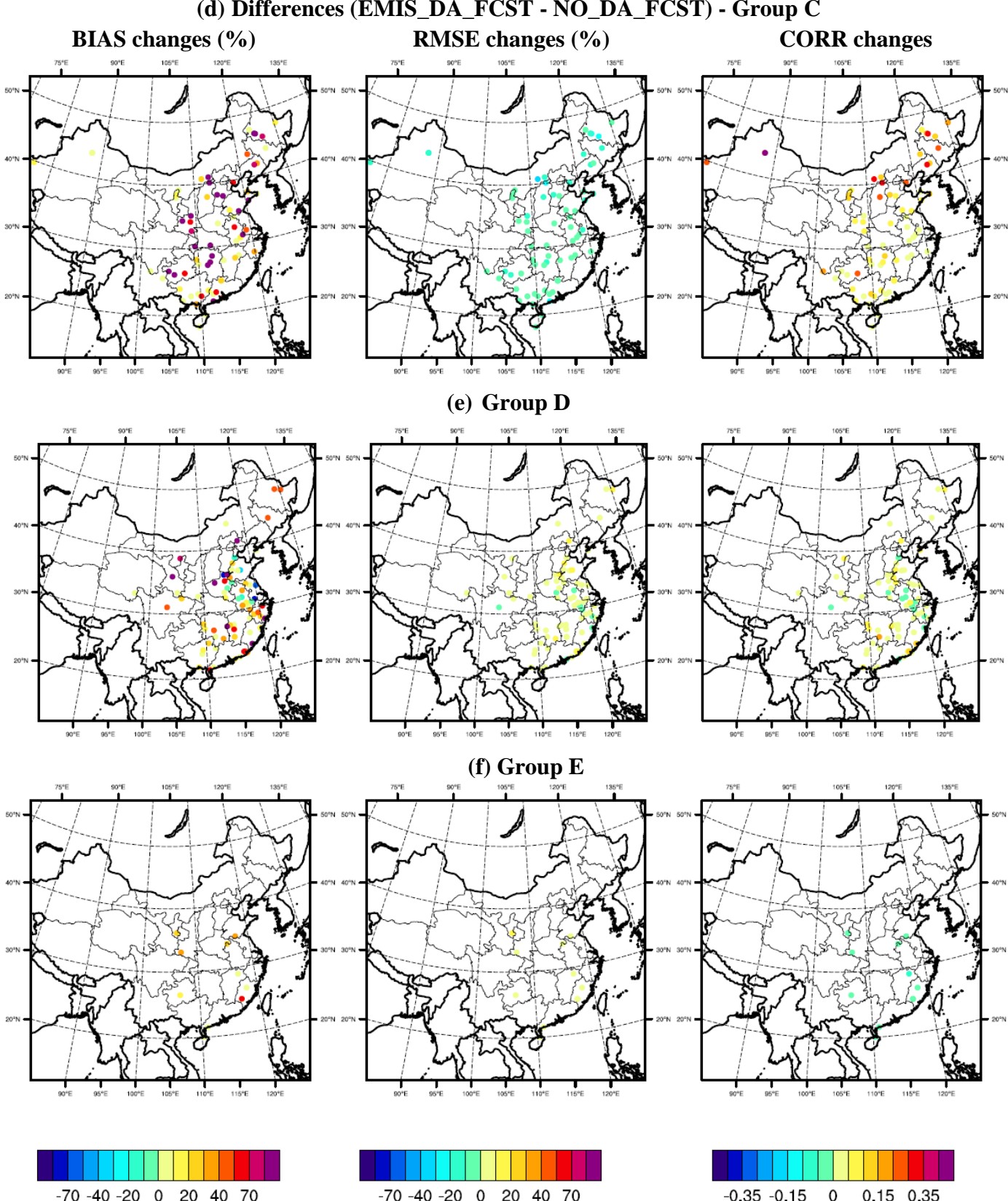

**Figure 12.** Same as Figure 11, but for January 2016.

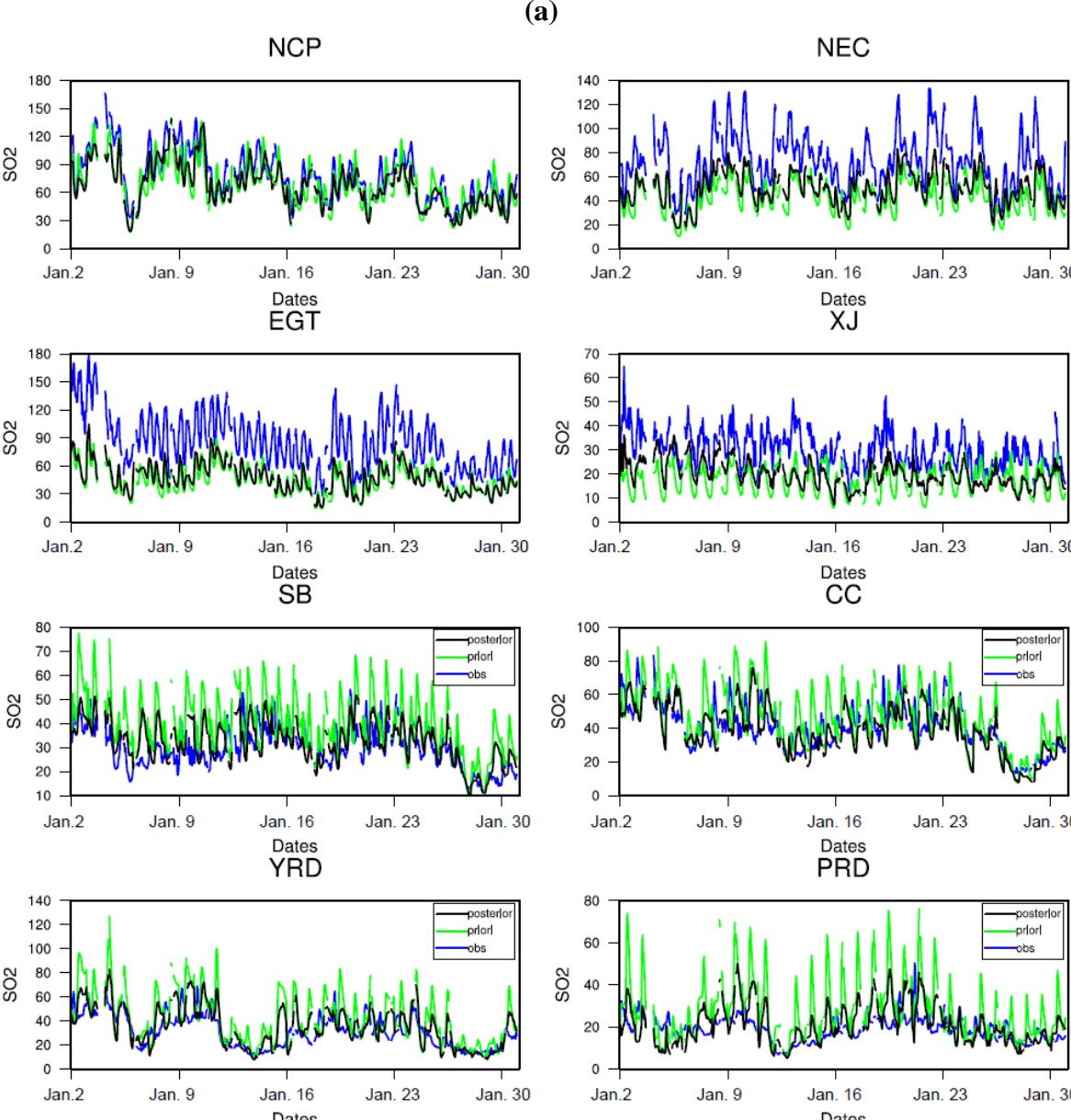

**Figure 13.** Time series of the regional mean SO₂ concentrations from the observations and from model simulations with the priori and analyzed (posterior) emissions for (a) January 2015 and (b) January 2016 in 8 regions. Time starts from 00UTC. (Units: µg m⁻³)

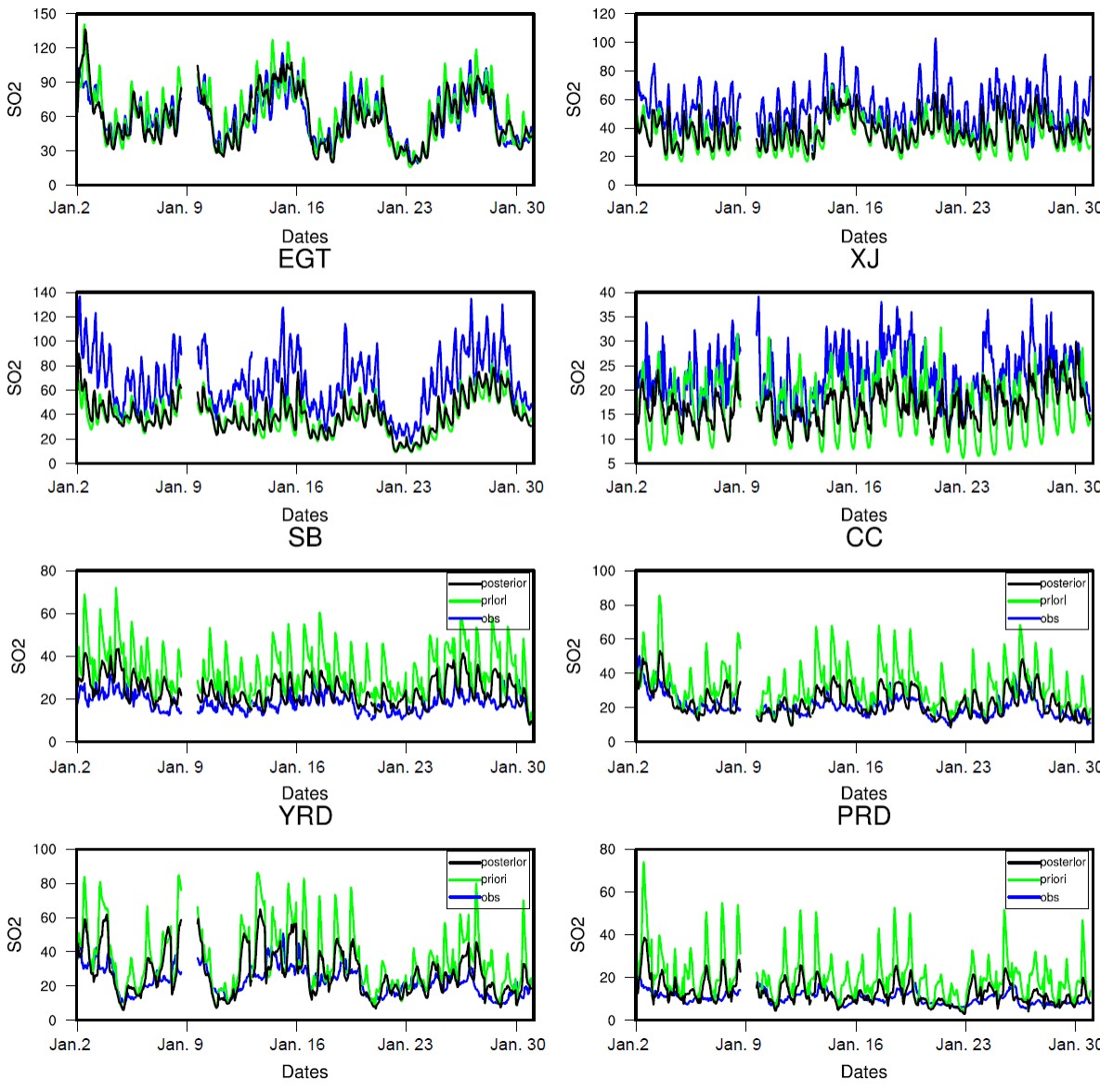

**Figure 13(b) Continue.**