# Peer review of "2015 and 2016 winter-time air pollution in China: SO₂ emission changes derived from a WRF-Chem/EnKF coupled data assimilation system"

_Atmospheric Chemistry and Physics, 2018_

## Referee Comment (RC1) · Anonymous Referee #1 · 8 Feb 2019

The paper with the title: '2015 and 2016 winter-time air pollution in China: SO2 emission changes derived from a WRF/Chem-EnKF coupled data assimilation system' analyzes the effect of data assimilation (DA) on the forecast of SO2 emissions for a Chinese domain comparing selected years. Doing this, the authors further aim to assess the impact of modified emissions (e.g. due to emission reduction strategies) from 2010 to 2015,2016 on SO2 levels for selected areas in China. The study per se is interesting and treats a relevant field also in the scope of this journal, due to its implication for a large area which is supposed to have regional to global impacts.

The partly poor language and structure however still need for significant revision before

[Figure]

I would recommend a publication in ACPD. Most importantly, the scope of the study has to be worked out more clearly. Starting from the abstract, which is missing from a clear description of the problem, the scope of the study and the presentation of the expected outcome. Continuing in the introduction section, it remains unclear what exactly is compared against, and what new development and improvements this study brings. In my point of view the study addresses two ways of data assimilation, correct? Is it the main task to evaluate this new development or is it rather the discussion of policy induced emission regulations. Please try to focus more clearly what are the main ideas behind this study and the intended outcome. Doing this, the introduction has to be re-structured and certain sections have to be revised due to language problems. The overall manuscript has to be double-checked for a clear presentation of firstly the purpose of the study, a clear description of the methodology and a precise presentation of the results and discussion of the outcome also in light of the presentation of the benefits of your approach for the research field.

With regard to the language, please check the entire manuscript for missing articles and filler words such as Page 4/ Line 5, Page 7/ Line2,Line 7.

According to methodology, some statements about the general WRF-Chem model setup have to be provided next to the explanation of the data assimilation framework. Is this adapted from another study? If so, please clarify. What is the difference of your approach to other DA configurations in WRF-Chem such as WRF-Chem/DART? Please discuss briefly.

In places, it is hard for the reader to distinguish between the different experiment-acronyms and one has to flip back and forth through the document. I would recommend to provide an overview table listing all the experiments used for the study with keyword/short description, purpose etc. Followed by a short description of what is aimed to be discovered in the single experiments, the scope of the study might become clearer for the reader as well. This information however is partly given at various places in the text already, but needs to be consolidated again in a revised version.
In this context it is not entirely clear what the difference between the experiments EMIS_DA and CONC_DA is and what emissions are used for the experiments. Maybe I am getting something wrong here, but to me it seems that 2010 emissions have been used for all simulations. As you are discussing emission reductions however, which dataset has been used for the years 2015, 2016? What approach did you follow to account for actual emission reductions? Besides that: Except from emissions and DA-technique, did you account for identical model configurations for all experiments? For instance, did you account for rapid urbanization in these areas in any dataset? What land use database is used and did the urban surface evolve over time? WRF offers different urban canopy models in its setup. Did you account for any of those?

Please find in the following further comments line by line:

Page 2 Line 2: this sentence is not quite correct. What you want to say something like: ambient concentration of various pollutants has been changed over the last couple of years.

L3: not clear what is meant by 'control studies' here and you how this is connected to the scope of your study. In general it is hard to follow the intention of you study within this sentence.

Please do re-write the abstract, particularly the introduction to it. This is crucial for getting the scope of your study and properly introduces the reader to the problem.

L11: unclear wording 'priori'

L17: how is this 'energy expansion strategy' manifested?

L21: unclear term: were corresponded

L22: bad sentence

Page3 Line 1: The analyzed emissions showed an improvement compared to what?

L2: BIAS and RMSE/correlation coefficients comparing what exactly? What does the

range mean?

L4: What is meant by 'limited due to a small spread'?

L9: Differences in lifetime relates to what process? Mixing, transport, boundary layer dynamics...?

L11: How are the emissions tracked by satellite observations?

L18: Source/citation missing for the 'national strategy'

L20: What is a power plant park?

L22: total amount has

L23: a bottom-up approach

Page4 Line2: unclear

L5: meaning unclear here: do you mean a wide spread analysis including several villages? I this context it is not clear what the Zhi (2017) paper really is about.

L8: Do the rural areas mentioned here refer to the studied villages?

L11: which models

L22: What models are you referring to here?

Page 5: Line 1: what exactly was perturbed

3-5: unclear structure and unclear meaning

7: capable of detecting

14: article missing

Page6: Line 10-21: fits better to the introduction

2: is used

Page7: Line 8: units missing

17: Peng et al: repetition

21: unclear

Page 8: Line 3,4: unclear about the details of the differences. What does the change from aerosols to SO2 infer? Why is that important?

22: unclear description of the cause for indirect relationships

Page 10: Line 18: convert

Section 2.3: Please specify the difference/benefits of your approach compared to the 'bottom up' inventories you mention here. What strategy is followed here? Please provide more details about the general methodology.

Page 11: Line2: What is the difference between priori emission year and focusing year?

L7: Source

L8: see above: unclear terminology

L19: optimal in what sense?

L21: unclear sentence

Page 12: Line 7: How often did you encounter these 650 ugm$^3$. Where does this threshold come from?

Page 13: Line 4: 50 member ensemble. As mentioned earlier, more information on the WRF-Chem model setup and general model configuration has to be provided here. Maybe a table is enough.

L14: Unclear meaning of 'assess the analyzed emission'. Basically you compare simulations with and without DA, correct? In general this Chapter needs more substance

in order to understand the differences and the purpose of the various experiments. As mentioned above, the different setups which have been used should be briefly summarized, e.g in a table.

Page 14: Line 13: 'Figure 13 shows' (check through the entire document)

L21: 'failure' bad wording here

L22: 'innovations exceeding' unclear

Page 15: Line10: 'The Golden Triangle'

Page 16: Line 4: What leads you to the conclusion that that meteorology is a minor factor here? Do you have proof for this, or for the other aspect respectively?

Page 18: Line 5: better: small/minor changes

Page 19: I was wondering what kind of land use dataset was used and how detailed the urban areas are represented. Did you account for an increase in urban land-cover/density from 2010 to 2016? Did you account for a urban canopy model included in WRF? As said before, a table highlighting the most important model setups will help.

Page 20: Line 7,8: unclear sentence

Page 21: Line1: Do colder temperatures in 2016 contribute to differences to 2015? Can you find different meteorological/dynamical patterns explaining this effect?

Chapter 5.1/5.2: Next to the presentation of the statistical measures (BIAS,RMSE) a more thorough discussion of their meaning and their connection to each other should be provided

General: The correct acronym should be WRF-Chem/EnKF.

Figures: Please check the legends and the axes for good readability.

Figure 13: What is the reason for the large spread in a priori $SO_2$? Please provide more details.

Figure 6: Please provide a better discussion on the large RMSE and -spread and qualitatively explain its numbers? Is it much/less or in the range of commonly found RMSEs for these approaches/ in this area, for this configuration? How is that related to the bias?

[Figure]

---

## Referee Comment (RC2) · Anonymous Referee #2 · 1 Mar 2019

Chen et al. applied a data assimilation system based on the WRF/Chem model and SO2 surface measurements to constrain hourly SO2 emissions over China for the January months 2015 and 2016. The two months were analyzed in order to (1) evaluate the Chinese SO2 emission reduction in recent years due to strict control strategies, and (2) test the ability of the WRF/Chem data assimilation to improve the emission estimates. The study is neatly conducted. An ensemble of model simulations with or without assimilation of the SO2 measurements are applied to quantify the emission changes in 2015 and 2016 relative to the prior emission estimates for the year 2010. This presents a good application of the data assimilation method to assess the Chinese emission and its changes that fits the journal scope well. I think the

authors shall address the following comments before publishing on ACP.

**Specific Comments:**
(1) Page 2, Line 7:
What does "larger spread" mean? Please explain "spread" here and in the text (Page 5, Line 1).

(2) Page 7, Line 20:
The study used the proxy of absolute emission values instead of emission scaling factors in the data assimilation system, as the authors explained that it allow the detection of new emission source. How is "new emission source" represented in the model? For a grid with zero emissions in the prior? If so, it is not clear in the text how this could be estimated and whether the system would generate negative emission estimates. If not, would the use of scaling factors in logarithm also work? This needs to be better described.

(3) Page 9, Section 2,2,2:
The section is difficult to understand for readers that did not read some papers in the reference list. What is the mathematical meaning of inflation factor? How to interpret the pointer symbol in Equations (2) and (3)? Please add more explanation.

(4) Page 12, Section 2.4:
Here the authors define SO2 measurements larger than 650 ug/m3 to be unrealistic, however, are these "unrealistic" measurements still used in the following comparisons (for example in Figure 3 and Figure 11). Please justify.

(5) Page 12, Line 11:
The phase "changing trend" is not proper here and throughout the text. The emission changes from 2015 to 2016 do not define a "trend", and the "trend" is not changing. I

suggest replace it by "emission changes".

(6) Page 13, Line 14:
Is there any difference between the experiments "NO_DA" and "NO_DA_forecast"? Generally, it is not very clear how many experiments were conducted and compared in this study. It would be helpful to add a table to summarize their information.

(7) Page 15, Line 15:
For "fixed hourly factors in the priori emissions", are the prior hourly factors provided by the MEIC inventory or defined by the model?

(8) Page 17, Section 4.2: A recent paper on ACP has analyzed the changes in Chinese anthropogenic emissions since 2010 based on the MEIC emission inventory. I suggest the auhtors compare their conclusions with the bottom-up estimates for additional evaluation.
Reference: Zheng, B., Tong, D., Li, M., Liu, F., Hong, C., Geng, G., Li, H., Li, X., Peng, L., Qi, J., Yan, L., Zhang, Y., Zhao, H., Zheng, Y., He, K., and Zhang, Q.: Trends in China's anthropogenic emissions since 2010 as the consequence of clean air actions, Atmos. Chem. Phys., 18, 14095-14111, https://doi.org/10.5194/acp-18-14095-2018, 2018.

(9) Page 21, Section 4.4: The study has presented monthly and hourly emission estimates. How about daily emission estimates? Are there significant daily variations in the constrained $SO_2$ emissions? Please clarify.

(10) Page 22, Line 6: Regarding the statement "the response time from emission to ambient concentration are simplified in the assimilation system", shouldn't the assimilation system consider physical and chemical transformations of $SO_2$ in the atmosphere? please explain why this is the case.

(11) Other corrections:

Page 7, Line 9 - "determined form" should be "determined from"

Page 7, Line 10 - defined the abbreviation "BECs"

Page 15, Line 4 - "emission decreasing" should be "emission decreases".

Page 17, Line 17 - "that from 2015 to 2016" should be "those from 2015 to 2016".

---

## Author Comment (AC1) · 28 Apr 2019

**COMMENTS TO THE AUTHOR(S)**

2015 and 2016 winter-time air pollution in China: SO2 emission changes derived from a WRF-Chem-EnKF coupled data assimilation system
Manuscript ID: acp-2018-1152
Authors: Chen, et al.

**Reviewer #1**

The paper with the title: '2015 and 2016 winter-time air pollution in China:  $SO_2$  emission changes derived from a WRF-Chem-EnKF coupled data assimilation system' analyzes the effect of data assimilation (DA) on the forecast of  $SO_2$  emissions for a Chinese domain comparing selected years. Doing this, the authors further aim to assess the impact of modified emissions (e.g. due to emission reduction strategies) from 2010to 2015,2016 on SO2 levels for selected areas in China. The study per se is interesting and treats a relevant field also in the scope of this journal, due to its implication for a large area which is supposed to have regional to global impacts.

The partly poor language and structure however still need for significant revision before I would recommend a publication in ACPD. Most importantly, the scope of the study has to be worked out more clearly. Starting from the abstract, which is missing from a clear description of the problem, the scope of the study and the presentation of the expected outcome. Continuing in the introduction section, it remains unclear what exactly is compared against, and what new development and improvements this study brings. In my point of view the study addresses two ways of data assimilation, correct? Is it the main task to evaluate this new development or is it rather the discussion of policy induced emission regulations. Please try to focus more clearly what are the main ideas behind this study and the intended outcome. Doing this, the introduction has to be re-structured and certain sections have to be revised due to language problems. The overall manuscript has to be double-checked for a clear presentation of firstly the purpose of the study, a clear description of the methodology and a precise presentation of the results and discussion of the outcome also in light of the presentation of the benefits of your approach for the research field.

With regard to the language, please check the entire manuscript for missing articles and filler words such as Page 4/ Line 5, Page 7/ Line2,Line 7.

According to methodology, some statements about the general WRF-Chem model setup have to be provided next to the explanation of the data assimilation frame work. Is this adapted from another study? If so, please clarify. What is the difference of your approach to other DA configurations in WRF-Chem such as WRF-Chem/DART? Please discuss briefly.

In places, it is hard for the reader to distinguish between the different experiment-acronyms and one has to flip back and forth through the document. I would recom-mend to provide an overview table listing all the experiments used for the study with key word/short description, purpose etc. Followed by a short description of what is aimed to be discovered in the single experiments, the scope of the study might be-come clearer for the reader as well. This information however is partly given at various places in the text already, but needs to be consolidated again in a revised version.

In this context it is not entirely clear what the difference between the experiments EMIS\_DA and CONC\_DA is and what emissions are used for the experiments. Maybe I am getting something wrong here, but to me it seems that 2010 emissions have been used for all simulations. As you are discussing emission reductions however, which dataset has been used for the years 2015, 2016? What approach did you follow to account for actual emission reductions? Besides that: Except from emissions and DA-technique, did you account for identical model configurations for all experiments? For instance, did you account for rapid urbanization in these areas in any dataset? What land use database is used and did the urban surface evolve over time? WRF offers different urban canopy models in its setup. Did you account for any of those?

**Response:**

Thanks for the valuable and insightful suggestions! We have made serval changes accordingly. Revised manuscript is after the response letter.

- 1. According to your suggestion, we hired the editing service from American Journal Experts (www.aje.com) and edited the manuscript thoroughly.
- 2. Rewrote the abstract and re-structured the introduction. Please see the updated abstract and introduction as below.

[revised manuscript text omitted]

5. Provided an overview table (Table 2 as below) listing all the experiments used for the study with key words, short descriptions, and purposes. Please find the details as below.

Identical model configurations were used for all the experiments (listed in Table 1). The same land use dataset (USGS) and land-surface model (NOAH LSM) were used. No urban canopy model was used in the study. We have added the discussion that "It should be noted that uncertainties might be produced in the analysis due to neglecting the rapid urbanization/land use changes from 2010 to 2015/2016".

[revised manuscript text omitted]

**Please find the itemized responses as below.**

Page 2 Line 2: this sentence is not quite correct. What you want to say something like: ambient concentration of various pollutants has been changed over the last couple of years.

L3: not clear what is meant by 'control studies' here and you how this is connected to the scope of your study. In general it is hard to follow the intention of you study within this sentence.

**Response:**

Thanks for pointing out this! Actually what we originally want to say is "To control pollution, a series of strict control strategies has been implemented by the government since 2010, including both long-term pollution control strategies and temporary emergency measures activated under different air pollution alerts, which has led to large spatial-temporal changes of emissions (factory mitigation from urban to rural regions, industries staggering peak production, etc.). These spatial-temporal emission changes are difficult to reflect in a timely manner in both bottom-up emissions inventories and air quality models, thus creating large uncertainties." As this sentences are too long in the abstract, we have removed and explained our point in the introduction.

Please do re-write the abstract, particularly the introduction to it. This is crucial forgetting the scope of your study and properly introduces the reader to the problem.

**Response:**

Yes, we did rewrite the abstract and the introduction.

**L11: unclear wording 'priori'**

**Response:**

Clarified in the text. "The 2010 January MEIC emission inventory was used as the priori (to generate background emission fields). The 2015 and 2016 January emissions were obtained by assimilating the hourly surface SO2 concentration observations for January 2015 and 2016."

L17: how is this 'energy expansion strategy' manifested?

**Response:**

Some studies have revealed the emission increase due to the energy expansion strategy in the northwestern China. The relevant policies were also added in the text.

Ling, Z. L., Huang, T., Zhao, Y., Li, J. X., Zhang, X. D., Wang, J. X., Lian, L. L., Mao, X. X., Gao, H., and Ma, J. M.: OMI-measured increasing SO2 emissions due to energy industry expansion and relocation in northwestern China, Atmos. Chem. Phys., 17, 9115-9131, 10.5194/acp-17-9115-2017, 2017.

Koukouli, M. E., Balis, D. S., van der A, R. J., Theys, N., Hedelt, P., Richter, A., Krotkov, N., Li, C., and Taylor, M.: Anthropogenic sulphur dioxide load over China as observed from different satellite sensors, Atmos. Environ., 145, 45-59, 10.1016/j.atmosenv.2016.09.007, 2016.

The Central Government of the People's Republic of China, The development of the western region in China: the twelfth five-year plan, 2012

The Central Government of the People's Republic of China, Strategic action Plan for Energy development (2014-2020), 2014

L21: unclear term: were corresponded

**Response:**

Clarified in the text. "These derived emissions changes coincided with the period of the energy development national strategy in northwestern China and the regulations for the reduction of SO2 emissions"

L22: bad sentence

**Response:**

**Removed in the abstract**

Page3 Line 1: The analyzed emissions showed an improvement compared to what?

L2: BIAS and RMSE/correlation coefficients comparing what exactly? What does the range mean?

**Response:**

Clarified in the text. "Forecast experiments using the priori and updated emissions were conducted. Comparisons showed improvements from using updated emissions. The improvements in southern China were much larger than those in northern and western China. For the Sichuan Basin, Central China, Yangtze River Delta, and Pearl River Delta, the BIAS decreased by 61.8%-78.2% (for different regions), the RMSE decreased by 27.9%-52.2%, and the correlation coefficients increased by 12.5%-47.1%."

L4: What is meant by 'limited due to a small spread'?

**Response:**

In a well-calibrated system, when compared to the observations, the prior ensemble mean rootmean square error (RMSE) would equal the prior "total spread" defined as the square root of the sum of the observation error variance and ensemble variance of simulated observations (Houtekamer *et al.*, 2005). For northern regions, the total spread (Figure 6) is relatively small compared to RMSE. It might indicate that the analyzed emissions converged gradually and the background emissions (calculated according to Eq. 4) of different members in the DA cycling were similar, thus leading to the small spread. As the spread is small, some observations might be rejected in the DA outlier check, thus may impact the DA performance.

The definition of RMSE and spread are as below.

For each time, an ensemble of *R* different forecasts are available, the ensemble mean is defined as,  $\bar{X} = \frac{1}{R} \sum_{r=1}^{R} X_r$ , for a region with *NS* observations (*Y*), the prior ensemble mean root-mean square error is defined as,  $RMSE = \sqrt{\frac{\sum_{is=1}^{NS} (\overline{X_{is}} - Y_{is})^2}{NS}}$ .

The ensemble variance at site *is* defined as,  $EV_{is} = \sqrt{\frac{\sum_{r=1}^{R} (X_{r,is} - \overline{X_{is}})^2}{R}}$ ; and the observation error at site *is* defined as,  $OE_{is} = \sqrt{\varepsilon_0^2 + \varepsilon_r^2}$  ( $\varepsilon_0$  is the measurement error and  $\varepsilon_r$  is the representativeness error); for a region with *NS* observations (*Y*), the total spread is defined as,  $total spread = \sqrt{\frac{\sum_{is=1}^{NS} (OE_{is})^2}{NS} + \frac{\sum_{is=1}^{NS} (EV_{is})^2}{NS}}$ .

L9: Differences in lifetime relates to what process? Mixing, transport, boundary layer dynamics...? **Response:**

Actually what we want to say is " $SO_2$  is a chemical reactive short-lived atmospheric trace gas compared to  $CO_2$ , CO etc." We rewrote the abstract and removed the sentence.

L11: How are the emissions tracked by satellite observations?

**Response:**

Basically the "top-down" emissions were generated by assimilating satellite vertical concentration columns combining air quality models (as the linkage of emission to ambient concentration). The results represented the best compromise between the assimilated observations and the available prior emissions. Please find the typical references as below.

Lee, C., Martin, R. V., van Donkelaar, A., Lee, H., Dickerson, R. R., Hains, J. C., Krotkov, N., Richter, A., Vinnikov, K., and Schwab, J. J.: SO2 emissions and lifetimes: Estimates from inverse modeling using in situ and global, space-based (SCIAMACHY and OMI) observations, J. Geophys. Res.-Atmos., 116, Artn D06304, 10.1029/2010jd014758, 2011.

Koukouli, M. E., Balis, D. S., van der A, R. J., Theys, N., Hedelt, P., Richter, A., Krotkov, N., Li, C., and Taylor, M.: Anthropogenic sulphur dioxide load over China as observed from different satellite sensors, Atmos. Environ., 145, 45-59, 10.1016/j.atmosenv.2016.09.007, 2016.

Ling, Z. L., Huang, T., Zhao, Y., Li, J. X., Zhang, X. D., Wang, J. X., Lian, L. L., Mao, X. X., Gao, H., and Ma, J. M.: OMI-measured increasing SO2 emissions due to energy industry expansion and relocation in northwestern China, Atmos. Chem. Phys., 17, 9115-9131, 10.5194/acp-17-9115-2017, 2017.

L18: Source/citation missing for the 'national strategy'

**Response:**

The national strategy can be referred to following policies.

The Central Government of the People's Republic of China, The development of the western region in China: the twelfth five-year plan, 2012

The Central Government of the People's Republic of China, Strategic action Plan for Energy development (2014-2020), 2014

The statement had been removed in the abstract and added in the main text.

L20: What is a power plant park?

**Response:**

Power Plant Park means the energy base with a few large power plants.

L22: total amount has

L23: a bottom-up approach

**Response:**

The sentences have been removed in the introduction. The similar errors at other places have been corrected.

**Page4 Line2: unclear**

**Response:**

"Unlike in other countries, the national emissions inventories in China (e.g., NEI05-08-11-14-17) are provided in a timely manner and updated for the public. It was previously stated that "there are no official data about how much air pollutants are emitted by China every year. The inventories developed by researchers often lag several years behind the present" (Zheng et al., 2018). MEIC is the only publicly available emissions inventory dataset released by the Tsinghua University research community. In the MEIC, the total amount of sectoral emissions at the national and provincial levels has generally been estimated based on the "bottom-up" approach, which relied on available statistical information concerning activities (energy, industrial production, vehicles, etc.), emissions factors and end-pipe control levels. Due to the large burden of work and the availability of statistical data, the MEIC emissions inventory is not updated annually (e.g., the public versions are MEIC-2010-2012); thus, there are always a few years of time-lag when applying MEIC EI for research studies. In addition, to drive the regional air quality models, the annual/monthly total amounts of emissions at the national and provincial level are allocated spatially and temporally to generate hourly gridded emissions input for the model. Concerning temporal allocation, as many emissions sources have large diurnal and weekly variability that is not fully represented, arbitrary hourly/weekly factors were used in preparing the hourly gridded emissions for the air quality models. Thus, the uncertainties of the statistical information and the spatial-temporal allocation could both cause inaccurate representation of the hourly gridded emissions input and affect the performance of the model application."

The statement had been removed in the introduction and added in section 2.2.

L5: meaning unclear here: do you mean a wide spread analysis including several villages? I this context it is not clear what the Zhi (2017) paper really is about.

L8: Do the rural areas mentioned here refer to the studied villages?

**Response:**

Zhi et al. (2017) conducted village energy survey for the rural areas of two cities, Beijing and

Baoding. The same group had done wide spread analysis for several villages in Hebei province. The references had been added in the text.

[revised manuscript text omitted]

**for this case."**

Page 8: Line 3,4: unclear about the details of the differences. What does the change from aerosols to SO2 infer? Why is that important?

**Response:**

"In air quality models, deficiencies of concentration simulations come from various aspects, including the initial condition, emissions, meteorology, chemistry, transport, etc. Especially for  $PM_{2.5}$  and  $PM_{10}$  simulations in China, the significant differences between the models and observations possibly come from the deficiency of the chemistry representation in the model, including missing paths of secondary organic aerosols (e.g., Chen *et al.*, 2017) and heterogeneous reactions (e.g., Zheng *et al.*, 2015), in addition to emissions. However, to reduce the error variance, emissions adjustments may compensate for the model error, which leads to unrealistic/excessive emissions adjustment. Because the purpose was to improve the forecasting of chemical species, evaluations of emission changes were not conducted in the previous two studies."

".....we updated the EnSRF system to evaluate SO2 emissions changes, for which the chemistry is better understood and represented in the model."

Zheng, B., Zhang, Q., Zhang, Y., He, K. B., Wang, K., Zheng, G. J., Duan, F. K., Ma, Y. L., and Kimoto, T.: Heterogeneous chemistry: a mechanism missing in current models to explain secondary inorganic aerosol formation during the January 2013 haze episode in North China, Atmos. Chem. Phys., 15, 2031-2049, 10.5194/acp-15-2031-2015, 2015.

Qi Chen, Tzung-May Fu, Jianlin Hu, Qi Ying, Lin Zhang, Modelling secondary organic aerosols in China, National Science Review, Volume 4, Issue 6, November 2017, Pages 806–809, https://doi.org/10.1093/nsr/nwx143

22: unclear description of the cause for indirect relation ships

**Response:**

**Clarified in the text.**

"Building upon Miyazaki *et al.* (2012), we used a similar approach to address the indirect relationships between  $SO_2$  concentrations and  $SO_2$  emissions caused by chemical and transport processes. The chemical processes include several paths of  $SO_2$  oxidation, such as gas-phase reactions with the hydroxyl radical (OH), aqueous-phase reactions with  $O_3$  or hydrogen peroxide

(H2O2), and heterogeneous reactions in high-RH environments (e.g., Li *et al.*, 2011; Wang *et al.*, 2012b)"

Page 10: Line 18: convert

**Response:**

**Corrected.**

Section 2.3: Please specify the difference/benefits of your approach compared to the 'bottom up' inventories you mention here. What strategy is followed here? Please provide more details about the general methodology.

**Response:**

The methodology of the bottom-up emission inventory was described in the same session (originally in section 2.3 and moved to section 2.2) and the benefits of our approach was also introduced.

"Unlike in other countries, the national emissions inventories in China (e.g., NEI05-08-11-14-17) are provided in a timely manner and updated for the public. It was previously stated that "there are no official data about how much air pollutants are emitted by China every year. The inventories developed by researchers often lag several years behind the present" (Zheng et al., 2018). MEIC is the only publicly available emissions inventory dataset released by the Tsinghua University research community. In the MEIC, the total amount of sectoral emissions at the national and provincial levels has generally been estimated based on the "bottom-up" approach, which relied on available statistical information concerning activities (energy, industrial production, vehicles, etc.), emissions factors and end-pipe control levels. Due to the large burden of work and the availability of statistical data, the MEIC emissions inventory is not updated annually (e.g., the public versions are MEIC-2010-2012); thus, there are always a few years of time-lag when applying MEIC EI for research studies. In addition, to drive the regional air quality models, the annual/monthly total amounts of emissions at the national and provincial level are allocated spatially and temporally to generate hourly gridded emissions input for the model. Concerning temporal allocation, as many emissions sources have large diurnal and weekly variability that is not fully represented, arbitrary hourly/weekly factors were used in preparing the hourly gridded emissions for the air quality models. Thus, the uncertainties of the statistical information and the spatial-temporal allocation could both cause inaccurate representation of the hourly gridded emissions input and affect the performance of

the model application."

The methodology of optimizing the emissions is described in section 2.4.

Page 11: Line2: What is the difference between priori emission year and focusing year?

**Response:**

For our case, MEIC-2010 was used for the simulations of 2015-2016. As MEIC-2010 used the statistical information of the year 2010, significant changes were expected to occur from 2010 to 2015-2016.

"There are several different driving factors in different regions that may lead to inhomogeneous changing trends during the examined years (especially from 2010 to 2015). As the Chinese government has implemented desulfurization legislation (since 2005-2006 but with stricter control of the actual use of installations since 2008-2009) and strict control strategies to ensure the air quality during winter seasons since 2013, significant SO2 emissions reductions are expected to have occurred since 2010. However, there are converse results for certain regions. For example, Cheng et al. (2017) and Zhi et al. (2017) conducted a village energy survey for the rural areas in Hebei Province and revealed a huge amount of missing rural raw coal for winter heating. For Beijing and Baoding, rural emissions from raw coal in winter were higher than those from the industrial and urban household sectors in the two cities in 2013. Considering the living habit of residents in northern China, this may imply an extreme underestimation of rural household coal consumptions by the China Energy Statistical Yearbooks. Additionally, multisatellite data (Ozone Monitoring Instrument-OMI, the SCanning Imaging Absorption spectroMeter for Atmospheric CartograpHY-SCIAMACHY) revealed increasing SO2 emissions due to energy industry expansion in northwestern China (the central government of the People's Republic of China, 2012, 2014), especially new power plant installations in Xinjiang and Shaanxi, e.g., in Shen et al. (2016) and Koukouli et al. (2016)."

**L7: Source**

L8: see above: unclear terminology

**Response:**

Clarified. "Additionally, multisatellite data (Ozone Monitoring Instrument-OMI, the SCanning Imaging Absorption spectroMeter for Atmospheric CartograpHY-SCIAMACHY) revealed increasing SO2 emissions due to energy industry expansion in northwestern China (the central government of the People's Republic of China, 2012, 2014), especially new power plant installations in Xinjiang and Shaanxi, e.g., in Shen *et al.* (2016) and Koukouli *et al.* (2016)"

L19: optimal in what sense?

**Response:**

"Actually, different sectors/regions may have different hourly emissions factors. For example, the transportation sector may emit peak emissions during rush hours, and the industry sector may emit emissions during the production period. Thus, it is not optimal to use the same hourly emissions factors for all sectors. For regions in different time zones, different hourly emissions factors are also expected. The diurnal variability is not publicly released and is highly uncertain, which brings large uncertainty in the simulations. We also want to investigate whether the DA system can capture the diurnal variations of SO2 emissions by using the hourly surface SO2 concentrations as constraints"

L21: unclear sentence

**Response:**

Clarified in the text.

"The diurnal variability is not publicly released and is highly uncertain, which brings large uncertainty in the simulations. We also want to investigate whether the DA system can capture the diurnal variations of SO2 emissions by using the hourly surface SO2 concentrations as constraints"

Page 12: Line 7: How often did you encounter these 650 ugm3. Where does this threshold come from?

**Response:**

According to the Air Quality Index in China, when the hourly SO2 is ranged at 650-800  $\mu$ g m-3, the SO2 pollution would be classified as the moderate pollution level; when the hourly SO2 is higher than 800  $\mu$ g m-3, the SO2 index would be calculated according to the 24-hour averages, instead of hourly averages. To ensure the accuracy of the hourly data and avoid the artificial adjustment of the data, we have chosen to treat the data larger than 650  $\mu$ g m-3 as unrealistic. Actually the ratio of those unrealistic data is very small, around 0.051% (only 1513 over 2938479 valid raw hourly data, at 1600+ sites for the three months).

Page 13: Line 4: 50 member ensemble. As mentioned earlier, more information on the WRF-Chem model setup and general model configuration has to be provided here. Maybe a table is enough.

**Response:**

The WRF-Chem model setup and general model configurations had been provided in Table 1 and section 2.1. The procedure of the DA cycling had been provided in section 2.4.4.

L14: Unclear meaning of 'assess the analyzed emission'. Basically you compare simulations with and without DA, correct? In general this Chapter needs more substance in order to understand the differences and the purpose of the various experiments. As mentioned above, the different setups which have been used should be briefly summarized, e.g in a table.

**Response:**

Yes, it's difficult to assess the analyzed emission. The statement had been modified. "To investigate the impact of using analyzed emissions from the EnKF DA system, two forecast experiments (NO\_DA\_forecast and EMIS\_DA\_forecast) were conducted for the same period. Twenty-four-hour forecasts were performed at 00UTC of each day from 1-31 January for 2015 and 2016. The original priori emissions and the updated analyzed emissions were used, respectively, in the NO\_DA\_forecast and EMIS\_DA\_forecast experiments."

Experiments conducted in this study and three groups of comparisons had been summarized in Table 2.

Page 14: Line 13: 'Figure 13 shows' (check through the entire document)

**Response:**

Corrected.

L21: 'failure' bad wording here

L22: 'innovations exceeding' unclear

**Response:**

The statement had been modified. "The reason why the improvements at those locations are not significant may be the data filtering process, in which  $SO_2$  data with either the observed values larger than 650 µg m-3 or innovations/deviations (observations minus the model-simulated values determined from the first-guess fields) exceeding 100 µg m-3 were rejected."

Page 15: Line10: 'The Golden Triangle'

**Response:**

Corrected

Page 16: Line 4: What leads you to the conclusion that that meteorology is a minor factor here? Do you have proof for this, or for the other aspect respectively?

**Response:**

It's based our assumption and we had described it in section 2.6. "The assumption is that the GFS 6-hr analysis data provide good meteorological IC/BC values and that the model accurately simulated the meteorology conditions; thus, the emissions were the major deficiency in the model."

"As the concentrations from CONC\_DA is very close to the observations, the concentration differences between CONC\_DA and NO\_DA possibly indicated a model deficiency in reproducing the reality, which was mainly from the emissions changes from 2010 to 2015-2016."

Page 18: Line 5: better: small/minor changes

**Response:**

Corrected. "These increases are relatively small in absolute value (shown in a light-yellow color in Fig. 7c), but the 2015/2010 ratios can reach large numbers (shown in orange to red colors in Fig. 7d), as the priori emissions in those regions are very small (Fig. 2); thus, minor changes lead to large ratios."

Page 19: I was wondering what kind of land use dataset was used and how detailed the urban areas are represented. Did you account for an increase in urban land-cover/density from 2010 to 2016? Did you account for a urban canopy model included in WRF? As said before, a table highlighting the most important model setups will help.

**Response:**

Thanks for pointing out this! The USGS land use 2m dataset was used in WRF-Chem simulation. We are not quite sure which dataset had been used when Tsinghua group conducted the spatial allocation in the MEIC-2010 emission inventory. The increase in urban land cover/density from 2010 to 2016 was not taken into account. No urban canopy model was used in WRF. The details are listed in Table 1. We have added the discussion as below. "It should be noted that uncertainties might be produced in the analysis due to neglecting the rapid urbanization/land use changes from 2010 to 2015/2016."

Page 20: Line 7,8: unclear sentence

**Response:**

The statement had been modified. "Koukouli *et al.* (2016) and Ling et al. (2017) used multisatellite data to investigate the SO2 load changes from 2004-2014/2005-2015 and identified locations with increases (including U'rumqi in Xinjiang and cities in northwestern China). They reported that "These belong to provinces with emerging economies which are in haste to install power plants and are possibly viewed leniently by the authorities, in favor of growth." Our findings are also consistent with those of these two studies."

Page 21: Line1: Do colder temperatures in 2016 contribute to differences to 2015?Can you find different meteorological/dynamical patterns explaining this effect?

**Response:**

The figures below (from Chen *et al.*, 2018) show the January meteorology differences from 2015 to 2016 (from model simulations with GFS 6-hr analysis data as IC/BC). Large differences are shown for planetary boundary layer height (PBLH), Surface pressure (PSFC), 2-meter Temperature (T2), 2-meter Relative Humidity (RH) and 10-meter wind speed.

---

## Author Response (AR1)

**Dear ACP Editor:**

**We have addressed all the comments raised by both reviewers, and incorporated them in the revised manuscript. Please find below our itemized responses to the reviewer's comments.**

**Thank you very much for your consideration.**

**Sincerely,**

**Dan Chen, et al.**
* * *
**COMMENTS TO THE AUTHOR(S)**

2015 and 2016 winter-time air pollution in China: $SO_2$ emission changes derived from a WRF-Chem-EnKF coupled data assimilation system

Manuscript ID:     acp-2018-1152

Authors: Chen, et al.

**Reviewer #1**

The paper with the title: '2015 and 2016 winter-time air pollution in China: $SO_2$ emission changes derived from a WRF-Chem-EnKF coupled data assimilation system' analyzes the effect of data assimilation (DA) on the forecast of $SO_2$ emissions for a Chinese domain comparing selected years. Doing this, the authors further aim to assess the impact of modified emissions (e.g. due to emission reduction strategies) from 2010 to 2015,2016 on SO2 levels for selected areas in China. The study per se is interesting and treats a relevant field also in the scope of this journal, due to its implication for a large area which is supposed to have regional to global impacts.

The partly poor language and structure however still need for significant revision before I would recommend a publication in ACPD. Most importantly, the scope of the study has to be worked out more clearly. Starting from the abstract, which is missing from a clear description of the problem, the scope of the study and the presentation of the expected outcome. Continuing in the introduction section, it remains unclear what exactly is compared against, and what new development and improvements this study brings. In my point of view the study addresses two ways of data

assimilation, correct? Is it the main task to evaluate this new development or is it rather the discussion of policy induced emission regulations. Please try to focus more clearly what are the main ideas behind this study and the intended outcome. Doing this, the introduction has to be re-structured and certain sections have to be revised due to language problems. The overall manuscript has to be double-checked for a clear presentation of firstly the purpose of the study, a clear description of the methodology and a precise presentation of the results and discussion of the outcome also in light of the presentation of the benefits of your approach for the research field.

With regard to the language, please check the entire manuscript for missing articles and filler words such as Page 4/ Line 5, Page 7/ Line2,Line 7.

According to methodology, some statements about the general WRF-Chem model setup have to be provided next to the explanation of the data assimilation frame work. Is this adapted from another study? If so, please clarify. What is the difference of your approach to other DA configurations in WRF-Chem such as WRF-Chem/DART? Please discuss briefly.

In places, it is hard for the reader to distinguish between the different experiment-acronyms and one has to flip back and forth through the document. I would recom-mend to provide an overview table listing all the experiments used for the study with key word/short description, purpose etc. Followed by a short description of what is aimed to be discovered in the single experiments, the scope of the study might be-come clearer for the reader as well. This information however is partly given at various places in the text already, but needs to be consolidated again in a revised version.

In this context it is not entirely clear what the difference between the experiments EMIS_DA and CONC_DA is and what emissions are used for the experiments. Maybe I am getting something wrong here, but to me it seems that 2010 emissions have been used for all simulations. As you are discussing emission reductions however, which dataset has been used for the years 2015, 2016? What approach did you follow to account for actual emission reductions? Besides that: Except from emissions and DA-technique, did you account for identical model configurations for all experiments? For instance, did you account for rapid urbanization in these areas in any dataset? What land use database is used and did the urban surface evolve over time? WRF offers different urban canopy models in its setup. Did you account for any of those?

**Response:**

Thanks for the valuable and insightful suggestions! We have made serval changes accordingly. Revised manuscript is after the response letter.

1.  According to your suggestion, we hired the editing service from American Journal Experts

(www.aje.com) and edited the manuscript thoroughly.

2. Rewrote the abstract and re-structured the introduction. Please see the updated abstract and introduction as below.

[revised manuscript text omitted]

3. Checked the entire manuscript for missing articles and also added details of the important articles, such as Page 4/ Line 5, Page 7/ Line2, Line 7.

4. Added the configurations of WRF-Chem model in section 2.1 and also added brief discussion of the differences compared with WRF-Chem/DART.

**Table 1.** WRF-Chem model configuration.

| | |
|---|---|
| Aerosol scheme | MOSAIC (4 bins) (Zaveri *et al.*, 2008) |
| Photolysis scheme | Fast-J (Wild *et al.*, 2000) |
| Gas phase chemistry | CBM-Z (Zavier *et al.*, 1999) |
| Cumulus parameterization | Grell 3D scheme |
| Short-wave radiation | Goddard Space Flight Center Shortwave radiation scheme (Chou and Suarez, 1994) |
| Long-wave radiation | RRTM (Mlawer *et al.*, 1997) |
| Microphysics | Single-Moment 6-class scheme (Grell and Devenyi, 2002) |
| Land-surface model | NOAH LSM (Chen and Dudhia, 2001) |
| Lan use type | USGS 2m (kept the same for 2010-2015-216) |
| Boundary layer scheme | YSU (Hong *et al.*, 2006) |
| Meteorology initial and boundary conditions | GFS analysis and forecast data at 6-hr frequency for control experiment, interpolated at 1-hr frequency for hourly assimilation experiments and forecast |

|  |  |
|---|---|
|  | experiments |
| Initial condition for chemical species | 11-day spin-up |
| Boundary conditions for chemical species | averages of mid-latitude aircraft profiles (McKeen *et al.*, 2002) |
| Dust and sea salt Emissions | GOCART |

"Similar to other ensemble DA configurations in WRF-Chem, such as WRF-Chem/Dart (Mizzi *et al.,* 2016), WRF-Chem is used to propagate the initial ensemble forward in time. The EnSRF is used to assimilate the observations and update the meteorological conditions, chemical initial conditions and/or emissions. The differences relative to WRF-Chem/Dart are mainly in the assimilation engine (the ensemble adjustment Kalman filter is used in WRF-Chem/Dart, while the ensemble square root filter is used in our study), the structure of the state variables (meteorological and chemical initial/conditions are used in WRF-Chem/Dart, while chemical initial condition/emissions are used in our study), the cycling procedures, etc."

Mizzi, A. P., Arellano, A. F., Edwards, D. P., Anderson, J. L., and Pfister, G. G.: Assimilating compact phase space retrievals of atmospheric composition with WRF-Chem/DART: a regional chemical transport/ensemble Kalman filter data assimilation system. Geoscientific Model Development, 9(3), 965- 740, 2016

5. Provided an overview table (Table 2 as below) listing all the experiments used for the study with key words, short descriptions, and purposes. Please find the details as below.

Identical model configurations were used for all the experiments (listed in Table 1). The same land use dataset (USGS) and land-surface model (NOAH LSM) were used. No urban canopy model was used in the study. We have added the discussion that "It should be noted that uncertainties might be produced in the analysis due to neglecting the rapid urbanization/land use changes from 2010 to 2015/2016".

In addition, it has always been challenging to verify optimized "top-down" emissions from the inverse approach due to the uncertainty of the "bottom-up" emissions inventory and the lack of sufficient independent observational data (not used in the DA process). Herein, we designed three groups of comparisons to address this issue. The details are also included in the table.

Table 2. Experiments conducted in this study and the three groups of comparisons.

| | | Design of the simulation | Purpose of the simulation | Purpose of the comparisons |
|---|---|---|---|---|
| Control experiments | NO_DA | 6-hr WRF-Chem cycling run with prior MEIC_2010 | Generate 2015-2016 concentration fields assuming the same emissions as in 2010 prior emissions | Concentrations from CONC_DA versus that from NO_DA:  As the concentrations from CONC_DA is very close to the observations, the concentration differences between CONC_DA and NO_DA possibly indicated a model deficiency in reproducing the reality, which was mainly from the emissions changes from 2010 to 2015-2016. The assumption is that the GFS 6-hr analysis data provide good meteorological IC/BC values and that the model accurately simulated the meteorology conditions; thus, the emissions were the major deficiency in the model. |
| DA experiments | CONC_DA | WRF-Chem (with prior MEIC_2010) and GSI 3D-var hourly DA cycle  Hourly observations were assimilated and WRF-Chem concentration output were updated | Generate 2015-2016 concentration reanalysis fields integrating hourly observations | |
| | EMIS_DA | WRF-Chem (with prior MEIC_2010 at the beginning and later with forecast emissions) and EnSRF hourly DA cycle  Hourly observations were assimilated, and WRF-Chem | Generate 2015-2016 analyzed emissions with hourly observations as constraints | Updated emissions from EMIS_DA versus prior emissions: The emissions differences between the 2015 analyzed emissions and the 2010 priori emissions not only reflected the changes from 2010 to 2015 but also included the deficiencies in the 2010 priori emission.  Updated emissions between different years: |

| | | concentration output and emissions were updated | | The differences between the 2016 analyzed emissions and the 2015 analyzed emissions reflected the pure emissions changes from 2015 to 2016, since the deficiencies of the 2010 priori emissions were offset in the subtraction

The emissions control policies are discussed to investigate whether the emissions changes are reasonable. |
|---|---|---|---|---|
| Forecast experiments | NO _DA_forecast | 24-hr WRF-Chem forecast with prior MEIC_2010, chemistry IC from CONC_DA at 00UTC | The simulation with only improved initial condition | Concentrations from EMIS_DA_forecast versus that from NO_DA_forecast:
The benefit by using updated emissions can be quantitatively assessed. |
| | EMIS_DA_fore cast | 24-hr WRF-Chem forecast with updated 2015-2016 emissions, chemistry IC from CONC_DA at 00UTC | The simulation with both improved initial condition and updated emissions | |

Please find the itemized responses as below.

Page 2 Line 2: this sentence is not quite correct. What you want to say something like: ambient concentration of various pollutants has been changed over the last couple of years.

L3: not clear what is meant by 'control studies' here and you how this is connected to the scope of your study. In general it is hard to follow the intention of you study within this sentence.

**Response:**

Thanks for pointing out this! Actually what we originally want to say is "To control pollution, a series of strict control strategies has been implemented by the government since 2010, including both long-term pollution control strategies and temporary emergency measures activated under different air pollution alerts, which has led to large spatial-temporal changes of emissions (factory mitigation from urban to rural regions, industries staggering peak production, etc.). These spatial-temporal emission changes are difficult to reflect in a timely manner in both bottom-up emissions inventories and air quality models, thus creating large uncertainties." As this sentences are too long in the abstract, we have removed and explained our point in the introduction.

Please do re-write the abstract, particularly the introduction to it. This is crucial forgetting the scope of your study and properly introduces the reader to the problem.

**Response:**

Yes, we did rewrite the abstract and the introduction.

**Response:**

Clarified in the text. "The 2010 January MEIC emission inventory was used as the priori (to generate background emission fields). The 2015 and 2016 January emissions were obtained by assimilating the hourly surface $SO_2$ concentration observations for January 2015 and 2016."

L17: how is this 'energy expansion strategy' manifested?

**Response:**

Some studies have revealed the emission increase due to the energy expansion strategy in the northwestern China. The relevant policies were also added in the text.

Ling, Z. L., Huang, T., Zhao, Y., Li, J. X., Zhang, X. D., Wang, J. X., Lian, L. L., Mao, X. X., Gao, H., and Ma, J. M.: OMI-measured increasing SO2 emissions due to energy industry expansion and relocation in northwestern China, Atmos. Chem. Phys., 17, 9115-9131, 10.5194/acp-17-9115-2017, 2017.

Koukouli, M. E., Balis, D. S., van der A, R. J., Theys, N., Hedelt, P., Richter, A., Krotkov, N., Li, C., and Taylor, M.: Anthropogenic sulphur dioxide load over China as observed from different satellite sensors, Atmos. Environ., 145, 45-59, 10.1016/j.atmosenv.2016.09.007, 2016.

*The Central Government of the People's Republic of China, The development of the western region in China: the twelfth five-year plan, 2012*

*The Central Government of the People's Republic of China, Strategic action Plan for Energy development (2014-2020), 2014*

L21: unclear term: were corresponded

**Response:**

Clarified in the text. "These derived emissions changes coincided with the period of the energy development national strategy in northwestern China and the regulations for the reduction of $SO_2$ emissions"

L22: bad sentence

**Response:**

Removed in the abstract

Page3 Line 1: The analyzed emissions showed an improvement compared to what?

L2: BIAS and RMSE/correlation coefficients comparing what exactly? What does the range mean?

**Response:**

Clarified in the text. "Forecast experiments using the priori and updated emissions were conducted. Comparisons showed improvements from using updated emissions. The improvements in southern China were much larger than those in northern and western China. For the Sichuan Basin,

Central China, Yangtze River Delta, and Pearl River Delta, the BIAS decreased by 61.8%-78.2% (for different regions), the RMSE decreased by 27.9%-52.2%, and the correlation coefficients increased by 12.5%-47.1%."

L4: What is meant by 'limited due to a small spread'?

**Response:**

In a well-calibrated system, when compared to the observations, the prior ensemble mean root-mean square error (RMSE) would equal the prior "total spread" defined as the square root of the sum of the observation error variance and ensemble variance of simulated observations (Houtekamer *et al.*, 2005). For northern regions, the total spread (Figure 6) is relatively small compared to RMSE. It might indicate that the analyzed emissions converged gradually and the background emissions (calculated according to Eq. 4) of different members in the DA cycling were similar, thus leading to the small spread. As the spread is small, some observations might be rejected in the DA outlier check, thus may impact the DA performance.

The definition of RMSE and spread are as below.

For each time, an ensemble of $R$ different forecasts are available, the ensemble mean is defined as, $\bar{X} = \frac{1}{R}\sum_{r=1}^{R} X_r$, for a region with $NS$ observations ($Y$), the prior ensemble mean root-mean square error is defined as, $RMSE = \sqrt{\frac{\sum_{is=1}^{NS}(\overline{X_{is}} - Y_{is})^2}{NS}}$ .

The ensemble variance at site $is$ defined as, $EV_{is} = \sqrt{\frac{\sum_{r=1}^{R}(X_{r,is} - \overline{X_{is}})^2}{R}}$; and the observation error at site $is$ defined as, $OE_{is} = \sqrt{\varepsilon_0^2 + \varepsilon_r^2}$ ($\varepsilon_0$ is the measurement error and $\varepsilon_r$ is the representativeness error); for a region with $NS$ observations ($Y$), the total spread is defined as, $total\ spread = \sqrt{\frac{\sum_{is=1}^{NS}(OE_{is})^2}{NS} + \frac{\sum_{is=1}^{NS}(EV_{is})^2}{NS}}$.

L9: Differences in lifetime relates to what process? Mixing, transport, boundary layer dynamics...?

**Response:**

Actually what we want to say is "$SO_2$ is a chemical reactive short-lived atmospheric trace gas compared to $CO_2$, CO etc." We rewrote the abstract and removed the sentence.

L11: How are the emissions tracked by satellite observations?

**Response:**

Basically the "top-down" emissions were generated by assimilating satellite vertical concentration columns combining air quality models (as the linkage of emission to ambient concentration). The results represented the best compromise between the assimilated observations and the available prior emissions. Please find the typical references as below.

Lee, C., Martin, R. V., van Donkelaar, A., Lee, H., Dickerson, R. R., Hains, J. C., Krotkov, N., Richter, A., Vinnikov, K., and Schwab, J. J.: $SO_2$ emissions and lifetimes: Estimates from inverse modeling using in situ and global, space-based (SCIAMACHY and OMI) observations, J. Geophys. Res.-Atmos., 116, Artn D06304, 10.1029/2010jd014758, 2011.

Koukouli, M. E., Balis, D. S., van der A, R. J., Theys, N., Hedelt, P., Richter, A., Krotkov, N., Li, C., and Taylor, M.: Anthropogenic sulphur dioxide load over China as observed from different satellite sensors, Atmos. Environ., 145, 45-59, 10.1016/j.atmosenv.2016.09.007, 2016.

Ling, Z. L., Huang, T., Zhao, Y., Li, J. X., Zhang, X. D., Wang, J. X., Lian, L. L., Mao, X. X., Gao, H., and Ma, J. M.: OMI-measured increasing $SO_2$ emissions due to energy industry expansion and relocation in northwestern China, Atmos. Chem. Phys., 17, 9115-9131, 10.5194/acp-17-9115-2017, 2017.

L18: Source/citation missing for the 'national strategy'

**Response:**

The national strategy can be referred to following policies.
*The Central Government of the People's Republic of China, The development of the western region in China: the twelfth five-year plan, 2012*
*The Central Government of the People's Republic of China, Strategic action Plan for Energy development (2014-2020), 2014*
The statement had been removed in the abstract and added in the main text.

L20: What is a power plant park?

**Response:**

Power Plant Park means the energy base with a few large power plants.

L22: total amount has

L23: a bottom-up approach

**Response:**

The sentences have been removed in the introduction. The similar errors at other places have been corrected.

**Response:**

"Unlike in other countries, the national emissions inventories in China (e.g., NEI05-08-11-14-17) are provided in a timely manner and updated for the public. It was previously stated that "there are no official data about how much air pollutants are emitted by China every year. The inventories developed by researchers often lag several years behind the present" (Zheng *et al.,* 2018). MEIC is the only publicly available emissions inventory dataset released by the Tsinghua University research community. In the MEIC, the total amount of sectoral emissions at the national and provincial levels has generally been estimated based on the "bottom-up" approach, which relied on available statistical information concerning activities (energy, industrial production, vehicles, etc.), emissions factors and end-pipe control levels. Due to the large burden of work and the availability of statistical data, the MEIC emissions inventory is not updated annually (e.g., the public versions are MEIC-2010-2012); thus, there are always a few years of time-lag when applying MEIC EI for research studies. In addition, to drive the regional air quality models, the annual/monthly total amounts of emissions at the national and provincial level are allocated spatially and temporally to generate hourly gridded emissions input for the model. Concerning temporal allocation, as many emissions sources have large diurnal and weekly variability that is not fully represented, arbitrary hourly/weekly factors were used in preparing the hourly gridded emissions for the air quality models. Thus, the uncertainties of the statistical information and the spatial-temporal allocation could both cause inaccurate representation of the hourly gridded emissions input and affect the performance of the model application."

The statement had been removed in the introduction and added in section 2.2.

L5: meaning unclear here: do you mean a wide spread analysis including several villages? I this context it is not clear what the Zhi (2017) paper really is about.

L8: Do the rural areas mentioned here refer to the studied villages?

**Response:**

Zhi *et al.* (2017) conducted village energy survey for the rural areas of two cities, Beijing and

Baoding. The same group had done wide spread analysis for several villages in Hebei province. The references had been added in the text.

[revised manuscript text omitted]

**Response:**

Corrected.

Section 2.3: Please specify the difference/benefits of your approach compared to the 'bottom up' inventories you mention here. What strategy is followed here? Please provide more details about the general methodology.

**Response:**

The methodology of the bottom-up emission inventory was described in the same session (originally in section 2.3 and moved to section 2.2) and the benefits of our approach was also introduced.

"Unlike in other countries, the national emissions inventories in China (e.g., NEI05-08-11-14-17) are provided in a timely manner and updated for the public. It was previously stated that "there are no official data about how much air pollutants are emitted by China every year. The inventories developed by researchers often lag several years behind the present" (Zheng *et al.,* 2018). MEIC is the only publicly available emissions inventory dataset released by the Tsinghua University research community. In the MEIC, the total amount of sectoral emissions at the national and provincial levels has generally been estimated based on the "bottom-up" approach, which relied on available statistical information concerning activities (energy, industrial production, vehicles, etc.), emissions factors and end-pipe control levels. Due to the large burden of work and the availability of statistical data, the MEIC emissions inventory is not updated annually (e.g., the public versions are MEIC-2010-2012); thus, there are always a few years of time-lag when applying MEIC EI for research studies. In addition, to drive the regional air quality models, the annual/monthly total amounts of emissions at the national and provincial level are allocated spatially and temporally to generate hourly gridded emissions input for the model. Concerning temporal allocation, as many emissions sources have large diurnal and weekly variability that is not fully represented, arbitrary hourly/weekly factors were used in preparing the hourly gridded emissions for the air quality models. Thus, the uncertainties of the statistical information and the spatial-temporal allocation could both cause inaccurate representation of the hourly gridded emissions input and affect the performance of

the model application."

The methodology of optimizing the emissions is described in section 2.4.

Page 11: Line2: What is the difference between priori emission year and focusing year?

**Response:**

For our case, MEIC-2010 was used for the simulations of 2015-2016. As MEIC-2010 used the statistical information of the year 2010, significant changes were expected to occur from 2010 to 2015-2016.

"There are several different driving factors in different regions that may lead to inhomogeneous changing trends during the examined years (especially from 2010 to 2015). As the Chinese government has implemented desulfurization legislation (since 2005-2006 but with stricter control of the actual use of installations since 2008-2009) and strict control strategies to ensure the air quality during winter seasons since 2013, significant $SO_2$ emissions reductions are expected to have occurred since 2010. However, there are converse results for certain regions. For example, Cheng *et al.* (2017) and Zhi *et al.* (2017) conducted a village energy survey for the rural areas in Hebei Province and revealed a huge amount of missing rural raw coal for winter heating. For Beijing and Baoding, rural emissions from raw coal in winter were higher than those from the industrial and urban household sectors in the two cities in 2013. Considering the living habit of residents in northern China, this may imply an extreme underestimation of rural household coal consumptions by the China Energy Statistical Yearbooks. Additionally, multisatellite data (Ozone Monitoring Instrument-OMI, the SCanning Imaging Absorption spectroMeter for Atmospheric CartograpHY-SCIAMACHY) revealed increasing $SO_2$ emissions due to energy industry expansion in northwestern China (the central government of the People's Republic of China, 2012, 2014), especially new power plant installations in Xinjiang and Shaanxi, e.g., in Shen *et al.* (2016) and Koukouli *et al.* (2016)."

**Response:**

Clarified. "Additionally, multisatellite data (Ozone Monitoring Instrument-OMI, the SCanning Imaging Absorption spectroMeter for Atmospheric CartograpHY-SCIAMACHY) revealed increasing $SO_2$ emissions due to energy industry expansion in northwestern China (the central

government of the People's Republic of China, 2012, 2014), especially new power plant installations in Xinjiang and Shaanxi, e.g., in Shen *et al.* (2016) and Koukouli *et al.* (2016)"

L19: optimal in what sense?

**Response:**

"Actually, different sectors/regions may have different hourly emissions factors. For example, the transportation sector may emit peak emissions during rush hours, and the industry sector may emit emissions during the production period. Thus, it is not optimal to use the same hourly emissions factors for all sectors. For regions in different time zones, different hourly emissions factors are also expected. The diurnal variability is not publicly released and is highly uncertain, which brings large uncertainty in the simulations. We also want to investigate whether the DA system can capture the diurnal variations of $SO_2$ emissions by using the hourly surface $SO_2$ concentrations as constraints"

L21: unclear sentence

**Response:**

Clarified in the text.

"The diurnal variability is not publicly released and is highly uncertain, which brings large uncertainty in the simulations. We also want to investigate whether the DA system can capture the diurnal variations of $SO_2$ emissions by using the hourly surface $SO_2$ concentrations as constraints"

Page 12: Line 7: How often did you encounter these 650 ugm3. Where does this threshold come from?

**Response:**

According to the Air Quality Index in China, when the hourly $SO_2$ is ranged at 650-800 $\mu g\ m^{-3}$, the $SO_2$ pollution would be classified as the moderate pollution level; when the hourly $SO_2$ is higher than 800 $\mu g\ m^{-3}$, the $SO_2$ index would be calculated according to the 24-hour averages, instead of hourly averages. To ensure the accuracy of the hourly data and avoid the artificial adjustment of the data, we have chosen to treat the data larger than 650 $\mu g\ m^{-3}$ as unrealistic. Actually the ratio of those unrealistic data is very small, around 0.051% (only 1513 over 2938479 valid raw hourly data, at 1600+ sites for the three months).

Page 13: Line 4: 50 member ensemble. As mentioned earlier, more information on the WRF-Chem model setup and general model configuration has to be provided here. Maybe a table is enough.

**Response:**

The WRF-Chem model setup and general model configurations had been provided in Table 1 and section 2.1. The procedure of the DA cycling had been provided in section 2.4.4.

L14: Unclear meaning of 'assess the analyzed emission'. Basically you compare simulations with and without DA, correct? In general this Chapter needs more substance in order to understand the differences and the purpose of the various experiments. As mentioned above, the different setups which have been used should be briefly summarized, e.g in a table.

**Response:**

Yes, it's difficult to assess the analyzed emission. The statement had been modified. "To investigate the impact of using analyzed emissions from the EnKF DA system, two forecast experiments (NO_DA_forecast and EMIS_DA_forecast) were conducted for the same period. Twenty-four-hour forecasts were performed at 00UTC of each day from 1-31 January for 2015 and 2016. The original priori emissions and the updated analyzed emissions were used, respectively, in the NO_DA_forecast and EMIS_DA_forecast experiments."

Experiments conducted in this study and three groups of comparisons had been summarized in Table 2.

Page 14: Line 13: 'Figure 13 shows' (check through the entire document)

**Response:**

Corrected.

L21: 'failure' bad wording here

L22: 'innovations exceeding' unclear

**Response:**

The statement had been modified. "The reason why the improvements at those locations are not significant may be the data filtering process, in which $SO_2$ data with either the observed values larger than 650 μg m$^{-3}$ or innovations/deviations (observations minus the model-simulated values determined from the first-guess fields) exceeding 100 μg m$^{-3}$ were rejected."

Page 15: Line10: 'The Golden Triangle'

**Response:**

Corrected

Page 16: Line 4: What leads you to the conclusion that that meteorology is a minor factor here? Do you have proof for this, or for the other aspect respectively?

**Response:**

It's based our assumption and we had described it in section 2.6. "The assumption is that the GFS 6-hr analysis data provide good meteorological IC/BC values and that the model accurately simulated the meteorology conditions; thus, the emissions were the major deficiency in the model."

"As the concentrations from CONC_DA is very close to the observations, the concentration differences between CONC_DA and NO_DA possibly indicated a model deficiency in reproducing the reality, which was mainly from the emissions changes from 2010 to 2015-2016."

Page 18: Line 5: better: small/minor changes

**Response:**

Corrected. "These increases are relatively small in absolute value (shown in a light-yellow color in Fig. 7c), but the 2015/2010 ratios can reach large numbers (shown in orange to red colors in Fig. 7d), as the priori emissions in those regions are very small (Fig. 2); thus, minor changes lead to large ratios."

Page 19: I was wondering what kind of land use dataset was used and how detailed the urban areas are represented. Did you account for an increase in urban land-cover/density from 2010 to 2016? Did you account for a urban canopy model included in WRF? As said before, a table highlighting the most important model setups will help.

**Response:**

Thanks for pointing out this! The USGS land use 2m dataset was used in WRF-Chem simulation. We are not quite sure which dataset had been used when Tsinghua group conducted the spatial allocation in the MEIC-2010 emission inventory. The increase in urban land cover/density from 2010 to 2016 was not taken into account. No urban canopy model was used in WRF. The details are listed in Table 1. We have added the discussion as below. "It should be noted that uncertainties might be produced in the analysis due to neglecting the rapid urbanization/land use changes from 2010 to 2015/2016."

Page 20: Line 7,8: unclear sentence

**Response:**

The statement had been modified. "Koukouli *et al.* (2016) and Ling et al. (2017) used multisatellite data to investigate the $SO_2$ load changes from 2004-2014/2005-2015 and identified locations with increases (including U'rumqi in Xinjiang and cities in northwestern China). They reported that "These belong to provinces with emerging economies which are in haste to install power plants and are possibly viewed leniently by the authorities, in favor of growth." Our findings are also consistent with those of these two studies."

Page 21: Line1: Do colder temperatures in 2016 contribute to differences to 2015?Can you find different meteorological/dynamical patterns explaining this effect?

**Response:**

The figures below (from Chen *et al.*, 2018) show the January meteorology differences from 2015 to 2016 (from model simulations with GFS 6-hr analysis data as IC/BC). Large differences are shown for planetary boundary layer height (PBLH), Surface pressure (PSFC), 2-meter Temperature (T2), 2-meter Relative Humidity (RH) and 10-meter wind speed.

[Figure]

Figure 8. Modeled meteorology condition changes (January 2016 – January 2015).

(a) PBLH, (b) PSFC, (c) T2, (d) RH2 and (e) 10-m wind speed. (Source: Chen *et al.,* 2018)

To clearly show the ambient response to different meteorology conditions from 2015-2016, the differences of the NO_DA experiments for the two months are shown in Figure 5b. As the same emission inventory (2010-MEIC) were used for the two months, the differences (Fig. 5b) mainly reflect the ambient response to the different meteorology conditions. It seemed that differences of meteorology conditions partially explain the ambient concentration differences, but it also indicated that factors other than meteorology also played important roles and caused changes in different directions. For example, due to the larger wind speed and higher PBLH in January 2016, lower concentrations were shown in most regions in Figure 5b. However, observations (Fig 5a) still showed grids with positive changes.

[Figure]

**Figure 5.** Observed and modeled SO$_2$ ambient concentration changes (January 2016 - January 2015). (a) Observations, (b) NO_DA, (c) CONC_DA.    (Unit: μg m$^{-3}$)

Chapter 5.1/5.2: Next to the presentation of the statistical measures (BIAS,RMSE) amore thorough discussion of their meaning and their connection to each other should be provided

**Response:**

The following discussions were added in the text.

"Statistics, including the BIAS (bias, equal to the difference between the modeled value and the observational value, representing the overall model tendency), RMSE (root mean square error/root mean square deviation, equal to the square root of the second moment of the differences between the model values and the observational values, reflecting both model biases and error variances) and CORR (correlation coefficient, equal to the linear relationship between the modeled values and the observational values), were chosen to evaluate the two forecast experiments with priori emissions and analyzed emissions, respectively. For a single site, the three statistics (BIAS, RMSE and CORR) may change in two directions—for example, the BIAS (bias of the absolute emission amount) may get worse, but the RMSE (error variance) and CORR (in terms of the diurnal or day-to-day emission changes) may get better."

General: The correct acronym should be WRF-Chem/EnKF.

**Response:**

Corrected.

Figures: Please check the legends and the axes for good readability.

**Response:**

Checked the legends and axes for the figures, and also added clarifications of the color bars etc. Would also like to make necessary changes in the future if needed.

Figure 13: What is the reason for the large spread in a priori $SO_2$? Please provide more details

**Response:**

In both NO_DA_forecast and EMIS_DA_forecast, the 24-hr forecast were conducted every day at 00UTC using the $SO_2$ concentration results from CONC_DA as initial conditions. It means for both NO_DA_forecast and EMIS_DA_forecast, the accumulated concentration biases were almost "corrected" at 00UTC every day; thus the differences of variations during a single day mainly come from the emission differences. In NO_DA_forecast using priori emissions, the absolute emission values were too high (for southern regions) and the hourly allocations were inappropriate, the two aspects caused that the simulated variations of $SO_2$ were just too large compared with observations.

Figure 6: Please provide a better discussion on the large RMSE and -spread and qualitatively explain its numbers? Is it much/less or in the range of commonly found RMSEs for these approaches/ in this area, for this configuration? How is that related to the bias?

**Response:**

For each time, an ensemble of $R$ different forecasts are available, the ensemble mean is defined as, $\bar{X} = \frac{1}{R}\sum_{r=1}^{R} X_r$, for a region with $NS$ observations ($Y$), the prior ensemble mean root-mean square error is defined as, $RMSE = \sqrt{\frac{\sum_{is=1}^{NS}(\overline{X_{is}}-Y_{is})^2}{NS}}$ .

The ensemble variance at site $is$ defined as, $EV_{is} = \sqrt{\frac{\sum_{r=1}^{R}(X_{r,is}-\overline{X_{is}})^2}{R}}$; and the observation error at site $is$ defined as, $OE_{is} = \sqrt{\varepsilon_0^2 + \varepsilon_r^2}$ ($\varepsilon_0$ is the measurement error and $\varepsilon_r$ is the representativeness error); for a region with $NS$ observations ($Y$), the total spread is defined as,

$$total\ spread = \sqrt{\frac{\sum_{is=1}^{NS}(OE_{is})^2}{NS} + \frac{\sum_{is=1}^{NS}(EV_{is})^2}{NS}}$$

The following discussions have been added in the text.

"Typically, the statistics at a single site for a certain period reflect the model biases

and variances of errors at that site for the whole period. Differently, herein, the statistics for the 8 regions were determined for all sites within the region at a 1-hr frequency, which means that the statistics actually reflect the biases and error variances of the model simulations for those sites at every hour. As the emissions and meteorology conditions could be very different at sites in the same region, the RMSE for that region could be large. Due to the spatial-temporal inhomogeneity of emissions and meteorological conditions in different regions, the model shows different performances in terms of the differences in the RMSE. The "total-spread" reflects the ensemble variances of the model-simulated values."

"For the northern regions, the total spread (Fig. 6) was relatively small compared to the RMSE. This might indicate that the analyzed emissions converged gradually and that the background emissions (calculated according to Eq. 4) of different members in the DA cycling were similar, thus leading to the small spread. As the spread is small, some observations might be rejected in the DA outlier check, which may impact the DA performance. The distinction of the comparisons among different regions (the North China Plain vs. the Yangtze River Delta/Pearl River Delta) indicated the deficiencies of the perturbation procedure in the DA system when applied to northern regions. Further investigations should be conducted to generate larger spreads for northern regions in future studies"

**Reviewer #2**

Chen *et al.* applied a data assimilation system based on the WRF-Chem model and SO$_2$ surface measurements to constrain hourly SO$_2$ emissions over China for the January months 2015 and 2016. The two months were analyzed in order to (1)evaluate the Chinese SO$_2$ emission reduction in recent years due to strict control strategies, and (2) test the ability of the WRF-Chem data assimilation to improve the emission estimates. The study is neatly conducted. An ensemble of model simulations with or without assimilation of the SO$_2$ measurements are applied to quantify the emission changes in 2015 and 2016 relative to the prior emission estimates for the year 2010. This presents a good application of the data assimilation method to assess the Chinese emission and its changes that fits the journal scope well. I think the authors shall address the following comments before publishing on ACP.

**Response:**

Thanks for the valuable and insightful suggestions! We have made serval changes accordingly. Please see the itemized responses as below. Revised manuscript is after the response letter.

Specific Comments:

(1) Page 2, Line 7:What does "larger spread" mean? Please explain "spread" here and in the text (Page5, Line 1).

**Response:**

Actually spread means the variance of ensemble members. In an ensemble system with $R$ members, the ensemble mean $\bar{X} = \frac{1}{R}\sum_{r=1}^{R} X_r$ and the spread (also as the ensemble variance) is defined as $\sqrt{\frac{\sum_{r=1}^{R} (X_r - \bar{\bar{X}})^2}{R}}$.

When the meteorology conditions and emissions were both perturbed, larger ensemble variances are easily achieved (in our previous study). However to investigate the exact emission-concentration relationship and investigate the DA system capability of updating emission changes, only emissions were perturbed in this study.

The abstract and introduction were rewritten, and the word "spread" were not in the two parts anymore. We have added the definition in the later text.

(2) Page 7, Line 20:The study used the proxy of absolute emission values instead of emission scaling factors in the data assimilation system, as the authors explained that it allow the detection of new emission source. How is "new emission source" represented in the model? For a grid with zero emissions in the prior? If so, it is not clear in the text how this could be estimated and whether the system would generate negative emission estimates. If not, would the use of scaling factors in logarithm also work? This needs to be better described.

**Response:**

Thanks for the suggestion, the scaling factors in logarithm should also work.

"For a grid with zero emissions in the priori emissions, the absolute emissions values would be added into the DA analysis to reflect the new emissions sources. Negative emissions estimates were not permitted in the system due to mandatory setting of the minimum values (a small positive value close to zero)"

(3) Page 9, Section 2,2,2:The section is difficult to understand for readers that did not read some papers in the reference list. What is the mathematical meaning of inflation factor? How to interpret the pointer symbol in Equations (2) and (3)? Please add more explanation.

**Response:**

"During the analysis process, the analyzed emissions of different members converge gradually, and the background emissions (calculated according to Eq. 4) of different members in the DA cycling become similar, thus leading to a small ensemble spread (variance). To maintain the spread level, an artificial inflation process (the original perturbations time the inflation factor larger than 1) was added to increase the perturbations."

The pointer symbol means the perturbations (standard deviations) were inflated and substituted the original ones.

(4) Page 12, Section 2.4:Here the authors define $SO_2$ measurements larger than 650 ug/m3 to be unrealistic, however, are these "unrealistic" measurements still used in the following comparisons(for example in Figure 3 and Figure 11). Please justify.

**Response:**

According to the Air Quality Index in China, when the hourly $SO_2$ is ranged at 650-800 μg m$^{-3}$, the $SO_2$ index would be classified as the moderate pollution level; when the hourly $SO_2$ is higher than 800 μg m$^{-3}$, the $SO_2$ index would be calculated according to the 24-hour average, instead of hourly averages. To ensure the accuracy of the hourly data and avoid the artificial adjustment of the data, we have chosen to treat the data larger than 650 μg m$^{-3}$ as unrealistic. Actually the ratio of those unrealistic data is very small 0.051% (only 1513 over 2938479 valid raw hourly data, at 1600+ sites for the three months).

Those unrealistic measurements (larger than 650 μg m$^{-3}$) were not used in the following comparisons (in Figure 3 and Figure 11). However, some high values (e.g. values of 649 μg m$^{-3}$) were still kept in the comparisons. As those values might possibly be rejected in the DA system (due to large innovations) but were still used in the comparisons, large discrepancies between the observation and assimilation results would still occur.

It's hard to determine the threshold of unrealistic values and also the rejection criteria in the DA system. To obtain the overall optimization and also balance the system computation stability/efficiently, the filter processes are necessary. It should be carefully tuned for different research purposes in the future.

(5) Page 12, Line 11:The phase "changing trend" is not proper here and throughout the text. The emission changes from 2015 to 2016 do not define a "trend", and the "trend" is not changing. I suggest replace it by "emission changes".

**Response:**

Thanks for the suggestion! Corrected in the text.

(6) Page 13, Line 14:Is there any difference between the experiments "NO_DA" and "NO_DA_forecast"? Generally, it is not very clear how many experiments were conducted and compared in this study. It would be helpful to add a table to summarize their information.

**Response:**

Thanks for the suggestion! The experiments and the descriptions are summarized in Table 2.

**Table 2.** Experiments conducted in this study and three groups of comparisons.

| | | Design of the simulation | Purpose of the simulation | Purpose of the comparisons |
|---|---|---|---|---|
| Control experiments | NO_DA | 6-hr WRF-Chem cycling run with prior MEIC_2010 | Generate 2015-2016 concentration fields assuming the same emissions as in 2010 prior emissions | Concentrations from CONC_DA versus that from NO_DA: As the concentrations from CONC_DA is very close to the observations, the concentration differences between CONC_DA and NO_DA possibly indicated a model deficiency in reproducing the reality, which was mainly from the emissions changes from 2010 to 2015-2016. The assumption is that the GFS 6-hr analysis data provide good meteorological IC/BC values and that the model accurately simulated the meteorology conditions; thus, the emissions were the major deficiency in the model. |
| DA experiments | CONC_DA | WRF-Chem (with prior MEIC_2010) and GSI 3D-var hourly DA cycle  Hourly observations were assimilated and WRF-Chem concentration output were updated | Generate 2015-2016 concentration reanalysis fields integrating hourly observations | |
| | EMIS_DA | WRF-Chem (with prior MEIC_2010 at the beginning and later with forecast emissions) and EnSRF hourly DA cycle  Hourly observations were assimilated, and WRF-Chem | Generate 2015-2016 analyzed emissions with hourly observations as constraints | Updated emissions from EMIS_DA versus prior emissions: The emissions differences between the 2015 analyzed emissions and the 2010 priori emissions not only reflected the changes from 2010 to 2015 but also included the deficiencies in the 2010 priori emission.  Updated emissions between different years: |

| | | concentration output and emissions were updated | | The differences between the 2016 analyzed emissions and the 2015 analyzed emissions reflected the pure emissions changes from 2015 to 2016, since the deficiencies of the 2010 priori emissions were offset in the subtraction

The emissions control policies are discussed to investigate whether the emissions changes are reasonable. |
|---|---|---|---|---|
| Forecast experiments | NO_DA_forecast | 24-hr WRF-Chem forecast with prior MEIC_2010, chemistry IC from CONC_DA at 00UTC | The simulation with only improved initial condition | Concentrations from EMIS_DA_forecast versus that from NO_DA_forecast:
The benefit by using updated emissions can be quantitatively assessed. |
| | EMIS_DA_forecast | 24-hr WRF-Chem forecast with updated 2015-2016 emissions, chemistry IC from CONC_DA at 00UTC | The simulation with both improved initial condition and updated emissions | |

(7) Page 15, Line 15:For "fixed hourly factors in the priori emissions", are the prior hourly factors provided by the MEIC inventory or defined by the model?

**Response:**

"fixed hourly factors in the priori emissions" were not provided by MEIC inventory but artificially preset by the model.

(8) Page 17, Section 4.2: A recent paper on ACP has analyzed the changes in Chinese anthropogenic emissions since 2010 based on the MEIC emission inventory. I suggest the auhtors compare their conclusions with the bottom-up estimates for additional evaluation. Reference: Zheng, B., Tong, D., Li, M., Liu, F., Hong, C., Geng, G., Li, H., Li, X., Peng,L., Qi, J., Yan, L., Zhang, Y., Zhao, H., Zheng, Y., He, K., and Zhang, Q.: Trends in China's anthropogenic emissions since 2010 as the consequence of clean air actions, Atmos. Chem. Phys., 18, 14095-14111, https://doi.org/10.5194/acp-18-14095-2018,2018.

**Response:**

Thanks for the suggestion! Discussions compared with the reference had been added in the text.

"In the recent study by Zheng *et al.* (2018), the 2010-2017 trends of anthropogenic emissions in China were investigated. According to the "bottom-up" approach, the annual total amounts of $SO_2$ emissions were calculated to be 27.8, 16.9 and 13.4 Tg for the years 2010, 2015 and 2016, respectively. The 2010 to 2015-2016 decreases were mostly attributed to the power and industry sectors due to the strict pollution control measures implemented for these two sectors. The sectoral distribution of emissions changed significantly during the recent years, and emissions other than those from power and industry have occupied larger portions, especially for the residential sector, as the current control policies have limited effects on reducing emissions from the residential sector. According to Zheng et al. (2018), the national total $SO_2$ emissions decreased by 20.8% from 2015 to 2016. Our derived changes ratios for the month of January in most of the regions (NEC, XJ, SB, CC, PRD) are comparable (13.9%, 16.1%, 12.4%, 15.5%, 12.6%, respectively, see Table 3), but the change ratios for NCP, ETR and YRD are relatively smaller. As discussed in Zheng *et al.* (2018), "bottom-up" emissions estimates are uncertain due to incomplete knowledge of the underlying data, and uncertainties are larger when emissions are contributed by scattered emissions sources. Especially for the residential sector, the effectiveness of the measures (e.g., phasing out of small

high-emissions stoves, banning of coal heating) is difficult to validate due to the lack of inspections; thus, higher uncertainties may arise for regions in which residential emissions are relatively important. "

(9) Page 21, Section 4.4: The study has presented monthly and hourly emission estimates. How about daily emission estimates? Are there significant daily variations in the constrained $SO_2$ emissions? Please clarify.

**Response:**

Yes, there are different daily variation patterns for different regions. For northern regions, the daily variation ratios (daily emissions/monthly mean emissions) ranges from 0.75-1.3. For southern regions, the ratios' ranges are relatively smaller. As it is difficult to verify the daily emission changes from reality in current stage, we chose to not discuss this issue in this paper.

(10) Page 22, Line 6: Regarding the statement "the response time from emis-sion to ambient concentration are simplified in the assimilation system", shouldn't the assimilation system consider physical and chemical transformations of $SO_2$ in the atmosphere? please explain why this is the case.

**Response:**

Yes, the assimilation system consider physical and chemical transformation of $SO_2$ in the atmosphere through the WRF-Chem forecast. However, in the EnSRF assimilation step, by using the observations at the current hour, the state variables including both emission for the last hour and concentrations at the current hour were both updated. It means, the reactions of $SO_2$ is only reflected in the WRF-Chem system but not in the EnKF process; considering the reaction time of $SO_2$ in the ambient, there might be some time lag of the analyzed emission. Thus ensemble algorithm taking into account of time series of observations should be more promising, although it may cause more expensive computing cost.

The description of initialization and the procedure of the DA system is added in the text to help the readers get the point.

"The WRF-Chem/EnKF assimilation system framework is shown in Fig. 1, and the workflow is briefly introduced here. The initialization and spin-up procedures of the 50-member ensemble were conducted using 72-hr ensemble forecasts ahead of the focused period through the same method used in Peng *et al.* (2016, 2018). For the 50 members, the lateral boundary conditions and initial condition of meteorology from GFS were perturbed by adding Gaussian random noise with

a zero mean and statistic background error covariances to the meteorological parameters. The emissions of the 50 members were generated by adding random noise to the priori emissions, similar to the method in Schwartz *et al.* (2014) and Peng *et al.* (2016). After the 72-hr forecasts, 50-memble ensemble SO$_2$ forecasts were generated, which were used as part of the background ($C_i^b$ in Eq. 1) in the first EMIS_DA cycle. The other part of the background ($E_i^b$ in Eq. 1) was the perturbed emissions of the last time step. In the EnSRF assimilation step, the state variables, including both the emissions for the last hour and the concentrations at the current hour, were updated. In the new 1-hr cycle, the background field of emissions is forecast through Eq. (4), and the background concentration is from the WRF-Chem 1-hr forecast using the updated chemical fields of the previous assimilation cycle as the ICs. With hourly cycling, the hourly analyzed emissions were obtained. "

(11) Other corrections:

Page 7, Line 9 - "determined form" should be "determined from"

Page 7, Line 10 - defined the abbreviation "BECs"

Page 15, Line 4 - "emission decreasing" should be "emission decreases".

Page 17, Line 17 - "that from 2015 to 2016" should be "those from 2015 to 2016".

**Response:**

Thanks! Corrected.

[revised manuscript text omitted]

**Figure 13(b) Continue.**

---

## Author Response (AR2)

**Dear ACP Editor:**

**We have addressed the comments raised by the reviewer, and incorporated them in the revised manuscript. Please find below our itemized responses to the reviewer's comments.**

**Thank you very much for your consideration.**

**Sincerely,**

**Dan Chen, et al.**
* * *
**COMMENTS TO THE AUTHOR(S)**

2015 and 2016 winter-time air pollution in China: $SO_2$ emission changes derived from a WRF-Chem-EnKF coupled data assimilation system

Manuscript ID:      acp-2018-1152

Authors: Chen, et al.

**Reviewer #1**

Dear authors,

Thank you for providing a detailed revision and meaningful answers to most of the points addressed. Find below some minor points which should be accounted for before publication:

a) Abstract: Line 10: still unclear what is meant by that line. Please specify what is meant by 'newly added sources' here. Please double-check if the abstract can be understood without knowing the manuscript and meets the criteria as listed at the ACP page.

b) With 'missing articles' I literally meant the missing of 'filling words' such as 'a', 'the' etc. please check again

c) With 'USGS 2m' you actually refer to minutes, correct? please clarify

**Response:**

Dear reviewer,

  Many thanks for your valuable and insightful suggestions that helped to improve our manuscript!

We have made serval changes accordingly. Revised manuscript is after the response letter.

1.  Actually 'newly added sources' means '"new" (emerging) emission sources that not considered in the priori emission inventory'. In our study, some emerging emission sources occurred in Xinjiang and the Energy Golden Triangle regions from 2010 to 2015, due to the energy expansion strategy. We have changed the statement accordingly in the abstract. We also asked the colleagues to read the abstract to make sure it meets the criteria as listed at the ACP page.

2.  Thanks and sorry for the confusing! Actually we hired the editing service from American Journal Experts (www.aje.com) and edited the manuscript thoroughly. Those missing articles had been checked and corrected carefully.

3.  Thanks! Yes, we are using USGS 2 minute data. We have clarified in the text.